# On the Expressive Power of GNNs to Solve Linear SDPs

**Chendi Qian** [1]   **Christopher Morris** [1]

## Abstract

Semidefinite programs (SDPs) are a powerful framework for convex optimization and for constructing strong relaxations of hard combinatorial problems. However, solving large SDPs can be computationally expensive, motivating the use of machine learning models as fast computational surrogates. Graph neural networks (GNNs) are a natural candidate in this setting due to their sparsity-awareness and ability to model variable-constraint interactions. In this work, we study what expressive power is sufficient to recover optimal SDP solutions. We first prove negative results showing that standard GNN architectures fail on recovering linear SDP solutions. We then identify a more expressive architecture that captures the key structure of SDPs and can, in particular, emulate the updates of a standard first-order solver. Empirically, on both synthetic and SDPLIB benchmarks of various classes of SDPs, this more expressive architecture achieves consistently lower prediction error and objective gap than theoretically weaker baselines. Finally, using the learned high-quality predictions to warm-start the first-order solver yields practical speedups of up to 80%.

## 1. Introduction

*Semidefinite programming* (SDP) is a cornerstone of modern optimization, serving as a fundamental tool in both combinatorial and convex optimization (Boyd & Vandenberghe, 2004; Nocedal & Wright, 1999). In particular, it provides tight relaxations for NP-hard *combinatorial optimization* (CO) problems, such as max-cut (Goemans & Williamson, 1995), max clique (Galli & Letchford, 2017), and graph coloring (Charikar, 2002); and it subsumes important problem classes including *second-order cone programming* (SOCP),

*quadratically constrained quadratic programming* (QCQP), and *linear programming* (LP) (Dattorro, 2010), see Appendix D for more details. Despite their theoretical utility, solving large-scale SDPs remains computationally prohibitive; state-of-the-art *interior point method* (IPM) solvers typically scale super-cubically with the problem size, e.g., $\mathcal{O}(n^3)$ or even $\mathcal{O}(n^6)$ per-iteration, depending on the concrete implementation (Helmberg et al., 1996; Vandenberghe & Boyd, 1996; Jiang et al., 2020).

To address these scalability challenges, *learning-to-optimize* (L2O) has emerged as a promising approach that aims to replace computationally intensive optimization routines with lightweight machine learning proxies. In this context, *graph neural networks* (GNNs) (Scarselli et al., 2008; Gilmer et al., 2017) have shown strong potential, particularly for learning to solve *mixed-integer linear programs* (MILPs) by representing them as variable-constraint bipartite graphs (Gasse et al., 2019; Chen et al., 2023b). Recent works have further aligned GNNs with classical algorithms, such as the IPM or *primal-dual hybrid gradient* (PDHG) algorithm (Qian et al., 2024; Li et al., 2024a).

However, extending this success to SDPs poses a fundamental challenge, and there is currently a lack of neural architectures capable of effectively representing even *linear SDPs*. Unlike LPs, where variables are vector entries, SDP variables are matrix entries, where the underlying matrix is constrained to the *positive semidefinite* (PSD) cone. This introduces unique structural symmetries, such as equivariance under simultaneous row and column permutations, that standard bipartite representations fail to capture. While the expressivity of GNNs and their connection to the Weisfeiler–Leman (WL) hierarchy of graph isomorphism tests (Cai et al., 1992) are well-studied (Xu et al., 2019; Morris et al., 2019; 2020; Maron et al., 2019a), the specific expressive power required to represent the mapping from a linear SDP instance to its optimal matrix solution remains unclear.

**Present work** In this work, we bridge GNN expressivity theory and an important class of SDP. That is, we formally characterize the inductive bias needed to represent linear SDPs. Our analysis reveals that 1- and 2-WL-like methods, which we term VC-WL and VC-2-WL, are insufficient. In contrast, we show that a *folklore 2-dimensional Weisfeiler–Leman* (2-FWL) equivalent architecture, which we denote

---

[1]Faculty of Computer Science, RWTH Aachen University, Germany. Correspondence to: Chendi Qian <chendi.qian@log.rwth-aachen.de>.

*Proceedings of the $43^{rd}$ International Conference on Machine Learning*, Seoul, South Korea. PMLR 306, 2026. Copyright 2026 by the author(s).

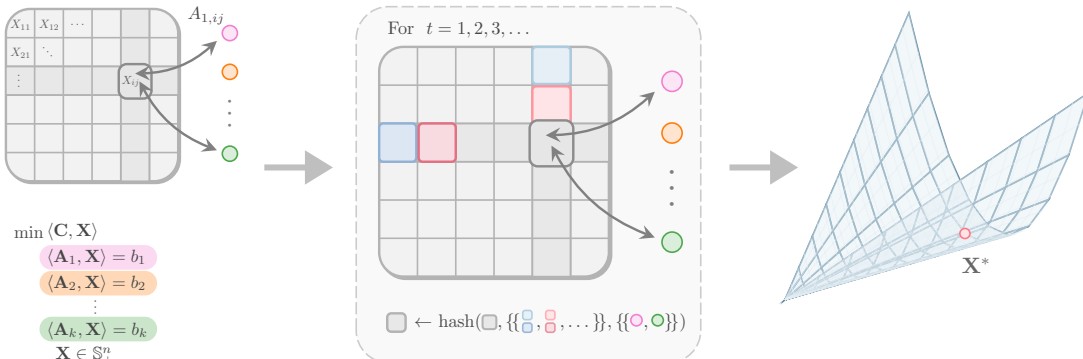

*Figure 1.* Our framework initializes variable and constraint embeddings directly from the SDP problem data. At each step $t$, the model iteratively updates these representations by aggregating messages from variable and constraint neighbors via permutation-equivariant functions. Finally, the variable embeddings are decoded to predict the optimal solution $\boldsymbol{X}^*$.

as VC-2-FWL, is capable of capturing linear SDPs.

Concretely, we contribute the following.

1. **Impossibility results** We prove that standard constraint-variable message passing (VC-WL) fails to represent SDPs. Furthermore, we show that the higher-order architecture VC-2-WL is also insufficient to capture SDPs.

2. **Theoretical sufficiency for solution** We establish that the expressivity of VC-2-FWL is sufficient to represent SDP solutions. Specifically, we prove that the algorithmic operations of the PDHG solver (Chambolle & Pock, 2016; Wang et al., 2024) are subsumed by the stable coloring of VC-2-FWL. This implies that VC-2-FWL possesses the expressivity to simulate the solver's trajectory and approximate the solution.

3. **Empirical validation** We validate our theory on synthetic and real-world benchmarks, showing that VC-2-FWL-expressive architectures consistently achieve the lowest loss and objective gap. Moreover, we demonstrate that utilizing these high-quality predictions to warm-start a standard solver significantly reduces convergence time.

*In summary, we formally establish the expressivity level required to solve linear SDPs, offering the first rigorous theoretical framework to ground the design of neural SDP solvers. By validating that expressive power directly translates to solution quality, our work establishes a principled blueprint for future research in learning-based optimization.*

**Conflict of Interest Disclosure** We declare that we have no specific financial conflicts of interest to disclose related to this work.

### 1.1. Related work

In the following, we discuss related work.

**MPNNs and L2O** *Message-passing neural networks* (MPNNs) (Gilmer et al., 2017; Scarselli et al., 2008) have been extensively studied and are broadly categorized into spatial and spectral variants. Spatial MPNNs (Bresson & Laurent, 2017; Duvenaud et al., 2015; Hamilton et al., 2017; Velickovic et al., 2018; Xu et al., 2019) follow the message-passing paradigm introduced by Gilmer et al. (2017). MPNNs have shown strong potential in L2O. A widely adopted approach represents MILPs using constraint-variable bipartite graphs (Gasse et al., 2019; Ding et al., 2020; Chen et al., 2023a; Khalil et al., 2022; Qian et al., 2024). Recent work has also aligned MPNNs with various optimization algorithms, including IPMs (Qian et al., 2024; Qian & Morris, 2025b), PDHG (Li et al., 2024a), and some distributed algorithms (Li et al., 2024b; 2025). From a theoretical perspective, several studies have analyzed the expressivity of MPNNs in approximating solutions to LP, QP, and more general SOCP (Chen et al., 2023a;b; 2024; 2025; Qian et al., 2024; Wu et al., 2025; Qian & Morris, 2025a; Li et al., 2026). In contrast, there has been relatively little work on applying machine learning to SDP. Early approaches leveraged the relation between KKT conditions and dynamical system equilibria to model SDP solving via ODEs (Jiang & Wang, 1999; Gao, 2004; Nikseresht & Nazemi, 2019; Kriváchy et al., 2021). However, these methods are empirically limited to small-scale instances with few variables. Closely related is Yau et al. (2024), which aligns GNNs with low-rank SDP relaxations for maximum constraint satisfaction problems (Max-CSP). However, a crucial distinction lies in the problem objective: Yau et al. (2024) analyzes GNNs as approximation algorithms for CO problems, using the algorithms of their SDP relaxations as a tool. In contrast, our work treats general linear SDPs as the primary object of

study, focusing on the GNN's ability to recover the optimal SDP solution itself.

**Expressivity of GNNs**  MPNNs are limited by the expressivity of the 1-WL test (Xu et al., 2019; Morris et al., 2019). To address this, various approaches have been proposed, including, among others, higher-order GNNs (Morris et al., 2019; 2020; Maron et al., 2019a; Morris et al., 2022), subgraph-based methods (Bevilacqua et al., 2022; Papp et al., 2021; Qian et al., 2022; Bevilacqua et al., 2024; Frasca et al., 2022; Bar-Shalom et al., 2024; Zhang et al., 2023; Southern et al., 2025), feature augmentation (Sato et al., 2021; You et al., 2021; Brasoveanu et al., 2023; Eliasof et al., 2023), and techniques probing local structure (Zhang & Li, 2021; Chen et al., 2022; Huang et al., 2023; Zhao et al., 2022). Graph transformers (Dwivedi & Bresson, 2020; Rampášek et al., 2022; Kim et al., 2022; Müller et al., 2024) have also been analyzed through the lens of the WL hierarchy (Müller & Morris, 2024; Müller et al., 2024). Notably, Hu et al. (2022) study 2-WL and 2-FWL for link prediction using second-order features, similar in form to our approach. However, our work differs in principle and focuses on semidefinite programming. For a survey on graph expressivity, see Morris et al. (2023).

## 1.2. Background

Here, we provide the necessary background; see Appendix B for an extended background.

**SDP**  Let $\mathbb{S}^n \subset \mathbb{R}^{n \times n}$ denote the set of symmetric matrices, and $\mathbb{S}^n_+ \subset \mathbb{S}^n$ denote the cone of PSD matrices. We consider the standard *linear SDP* problem with primal variable $\boldsymbol{X} \in \mathbb{S}^n_+$ of the form:[1]

$$\min_{\boldsymbol{X} \in \mathbb{S}^n_+} \quad \langle \boldsymbol{C}, \boldsymbol{X} \rangle$$
$$\text{s.t.} \quad \langle \boldsymbol{A}_k, \boldsymbol{X} \rangle = b_k, \quad k \in [m]. \tag{1}$$

Here, $\boldsymbol{A}_k \in \mathbb{S}^n$ represents the $k$-th constraint matrix, whose entry indexed at $(i, j)$ is $A_{k,ij}$, and we denote the stack of them as a tensor $\mathbf{A} \in \mathbb{R}^{m \times n \times n}$. Here, the dot product on matrices is defined as $\langle \boldsymbol{C}, \boldsymbol{X} \rangle := \sum_{i=1}^n \sum_{j=1}^n C_{ij} X_{ij}$. Further, we define the linear operator

$$\mathcal{A} \colon \mathbb{S}^n \to \mathbb{R}^m, \ \mathcal{A}(\boldsymbol{X}) = \left( \langle \boldsymbol{A}_1, \boldsymbol{X} \rangle, \ldots, \langle \boldsymbol{A}_m, \boldsymbol{X} \rangle \right)^\mathsf{T},$$

and its adjoint operator

$$\mathcal{A}^* \colon \mathbb{R}^m \to \mathbb{S}^n, \quad \mathcal{A}^*(\boldsymbol{y}) = \sum_{i=k}^m y_k \boldsymbol{A}_k$$

for brevity. While the primal variable $\boldsymbol{X}$ contains $n^2$ entries, the symmetry requirement restricts the problem to $\frac{n(n+1)}{2}$

free variables. We assume the constraints are linearly independent; otherwise, some of them can be reduced, or the problem is infeasible, thus $m$ is bounded by $\mathcal{O}(n^2)$. Without loss of generality, we assume the coefficient matrices $\boldsymbol{C}$ and $\{\boldsymbol{A}_k\}_{k=1}^m$ are symmetric.[2] SDP problems may admit multiple optimal solutions with various ranks (Han et al., 2025), making the solution set diverse and intractable to characterize. To ensure a well-defined learning target, we follow Chen et al. (2023a) in the LP case and focus on the unique optimal solution with the *minimum Frobenius norm*.

**Proposition 1.1.** *The SDP problem defined in Equation* (1) *has a unique primal solution* $\boldsymbol{X}^*$ *with the minimum Frobenius norm* $\|\boldsymbol{X}^*\|_F^2$.

**WL hierarchy**  The WL hierarchy is one of the heuristics for the graph isomorphism problem (Cai et al., 1992), and a measurement for the expressive power of GNNs (Morris et al., 2023). Let $[n] := \{1, \ldots, n\}$ and let $\{\!\{\ldots\}\!\}$ denote multisets. A graph $G := (V, E)$ consists of a finite set of *nodes* $V$ and *edges* $E \subseteq \{(u, v) \mid u, v \in V\}$.[3][4] The 1-WL (Weisfeiler & Leman, 1968) (or *color refinement*) iteratively updates the color $\boldsymbol{c}_v^t$, for iteration $t > 0$, of a node $v$ based on the colors of its neighbors $N(v) := \{u \in V \mid (u, v) \in E\}$. The node colors are initialized with labeling function $l \colon V \to \mathbb{N}$ as $\boldsymbol{c}_v^0 := l(v)$, and updated with

$$\boldsymbol{c}_v^t := \mathsf{hash}\Big(\boldsymbol{c}_v^{t-1}, \big\{\!\!\big\{ \boldsymbol{c}_u^{t-1} \mid u \in N(v) \big\}\!\!\big\}\Big),$$

for $t > 0$. To distinguish more graphs, the hierarchy extends to $k$-tuples of nodes ($k \geq 2$), leading to the $k$-WL. In this work, we focus on the case $k = 2$, where colors are assigned to node pairs $(u, v) \in V^2$. The initialization $\boldsymbol{c}_{uv}^0 := \mathsf{atp}(u, v)$ is the *atomic type*, which encodes the isomorphism type of the pair, i.e., whether $u = v$ and whether $(u, v) \in E$. The *folklore* (2-FWL) and *standard* (2-WL) variants differ in how they aggregate information from a third node $w \in V$. That is, the 2-FWL aggregates the *joint* configuration of $w$ with respect to both $u$ and $v$, i.e.,

$$\boldsymbol{c}_{uv}^t := \mathsf{hash}\Big(\boldsymbol{c}_{uv}^{t-1}, \big\{\!\!\big\{ (\boldsymbol{c}_{wv}^{t-1}, \boldsymbol{c}_{uw}^{t-1}) \mid w \in V \big\}\!\!\big\}\Big).$$

In contrast, 2-WL aggregates the interaction of $w$ with each node independently:

$$\mathsf{hash}\Big(\boldsymbol{c}_{uv}^{t-1}, \big(\big\{\!\!\big\{ \boldsymbol{c}_{wv}^{t-1} \mid w \in V \big\}\!\!\big\}, \big\{\!\!\big\{ \boldsymbol{c}_{uw}^{t-1} \mid w \in V \big\}\!\!\big\}\big)\Big).$$

Crucially, 2-FWL is strictly more expressive than 2-WL (Grohe, 2017). Both algorithms run until stabilization and

---

[1]In some literature, the $\min$ is replaced by $\inf$, as the minimum may not be reached. We exclude this case.

[2]If $\boldsymbol{C}$ is not symmetric, it can be replaced by $(\boldsymbol{C} + \boldsymbol{C}^\mathsf{T})/2$ without affecting the objective value, as $\boldsymbol{X}$ is symmetric.

[3]Sometimes denoted as $V(G)$ and $E(G)$ for specific $G$.

[4]For notational convenience, we usually write $(u, v)$ for the undirected edge $\{u, v\}$.

yield a unique *stable coloring* $\boldsymbol{c}^\infty$. Finally, we say algorithm $\mathcal{A}$ *refines* $\mathcal{B}$, denoted $\mathcal{A} \sqsubseteq \mathcal{B}$, if $\mathcal{A}$ leads to a finer partitioning of (higher-order) nodes than $\mathcal{B}$. The corresponding strict relation is denoted by $\sqsubset$.

**MPNNs** MPNNs learn a $d$-dimensional real-valued vector of each node in a graph by aggregating information from neighboring nodes. Following Gilmer et al. (2017), let $G$ be an attributed, edge-weighted graph with edge-weights $w\colon E(G) \to \mathbb{R}$ with initial node-feature $\boldsymbol{h}_v^0 \in \mathbb{R}^{d_0}$, $d_0 \in \mathbb{N}$, for $v \in V(G)$. An *MPNN architecture* consists of a composition of $L$ neural network layers for some $L > 0$. In each *layer*, $t \in \mathbb{N}$, we compute a node feature

$$\boldsymbol{m}_v^t = \mathsf{AGG}^t\left(\left\{\!\!\left\{\left(\boldsymbol{h}_v^{t-1}, \boldsymbol{h}_u^{t-1}, w_{vu}\right) \mid u \in N(v)\right\}\!\!\right\}\right)$$
$$\boldsymbol{h}_v^t = \mathsf{UPD}^t\left(\boldsymbol{h}_v^{t-1}, \boldsymbol{m}_v^t\right) \in \mathbb{R}^{d_t}$$

where $\mathsf{UPD}^t$ and $\mathsf{AGG}^t$ may be parameterized functions, e.g., neural networks. In the case of graph-level tasks, one uses a READOUT function

$$\boldsymbol{h}_G \coloneqq \mathsf{READOUT}\left(\left\{\!\!\left\{\boldsymbol{h}_v^L \mid v \in V(G)\right\}\!\!\right\}\right) \in \mathbb{R}^d,$$

to compute a single vectorial representation based on learned node features after iteration $L$. Again, READOUT may be a parameterized function.

# 2. Representing SDP instances

Here, we outline our graph representation for faithfully encoding SDP problems.

## 2.1. From LP to SDP

We first revisit the standard graph representation used for LP. Given an LP instance $I\colon \min_{\boldsymbol{x} \geq \boldsymbol{0}} \ \boldsymbol{c}^\intercal\boldsymbol{x}$ s.t. $\boldsymbol{A}\boldsymbol{x} = \boldsymbol{b}$, we can construct a bipartite *variable-constraint* (V-C) graph with constraint nodes $C(I)$ and variable nodes $V(I)$ (Chen et al., 2023a). Edges between $C(I)$ and $V(I)$ are defined by nonzero entries of $\boldsymbol{A}$ with weights $A_{cv}$, for $v \in V(I), c \in C(I)$. Since LP is a special case of SDP where all matrices are diagonal, it is natural to extend this representation to the general SDP setting. Specifically, we treat the matrix-shaped variable $\boldsymbol{X}$ as a collection of $n^2$ individual variables, initializing node features with coefficients from the objective matrix $\boldsymbol{C}$; and create $m$ constraint nodes initialized with $\boldsymbol{b}$. A variable node indexed by $(i, j)$ is connected to a constraint node $k$ if $A_{k,ij} \neq 0$, with edge weight $A_{k,ij}$. We hereby define the neighbors of variable nodes $N(ij) \coloneqq \{k \in [m] \mid A_{k,ij} \neq 0\}$, corresponding to the variable $X_{ij}$, and of constraint nodes $N(k) \coloneqq \{(i,j) \in [n]^2 \mid A_{k,ij} \neq 0\}$, corresponding to the constraint indexed $k$.

To study the expressive power of MPNNs operating on V-C graphs for SDPs, we define a variant VC-WL of the 1-WL.

To that, let $\boldsymbol{v}_{ij}^t$ and $\boldsymbol{c}_k^t$ denote the colors of variable node $(i, j)$ and constraint node $k$ at iteration $t$, respectively. We define the initialization and color update as follows,

$$\begin{aligned} \boldsymbol{v}_{ij}^0 &\coloneqq \mathsf{init}_{\mathsf{v}}(C_{ij}, \mathbb{I}_{i=j}), \text{ for } i, j \in [n], \\ \boldsymbol{c}_k^0 &\coloneqq \mathsf{init}_{\mathsf{c}}(b_k), \text{ for } k \in [m], \qquad (2) \\ \boldsymbol{v}_{ij}^t &\coloneqq \mathsf{hash}\big(\boldsymbol{v}_{ij}^{t-1}, \{\!\!\{(A_{k,ij}, \boldsymbol{c}_k^{t-1}) \mid k \in N(ij)\}\!\!\}\big), \\ \boldsymbol{c}_k^t &\coloneqq \mathsf{hash}\big(\boldsymbol{c}_k^{t-1}, \{\!\!\{(A_{k,ij}, \boldsymbol{v}_{ij}^{t-1}) \mid (i,j) \in N(k)\}\!\!\}\big), \end{aligned}$$

where init and hash are injective functions, $\mathbb{I}_{i=j}$ is a diagonal indicator which takes 1 if $i = j$ otherwise 0. While this representation captures the sparsity of the problem and inherits the invariance and equivariance of MPNNs, we show it fundamentally lacks the expressivity required for SDPs.

**Proposition 2.1.** *The* VC-WL *fails to represent linear SDP solutions. That is, there exist instances where the stable coloring of the* VC-WL *satisfies* $\boldsymbol{v}_{ij}^\infty = \boldsymbol{v}_{pq}^\infty$ *for distinct variable indices* $(i, j)$ *and* $(p, q)$, *yet the entries in the unique optimal solution differ, i.e.,* $X_{ij}^* \neq X_{pq}^*$.

Hence, in the following, we investigate more powerful algorithms.

## 2.2. Leveraging higher-order information

The limitation of standard bipartite graph representation stems from treating SDP variables as independent entities, effectively flattening the problem and discarding the crucial intrinsic matrix geometry. Unlike standard LPs, the primal variables $X_{ij}$ of an SDP form a PSD matrix, characterized by spectral properties and global correlations. Consequently, an effective representation must move beyond bipartite graph representation and instead explicitly model the global interactions among $X_{ij}$.

We seek a neural architecture $\mathsf{NN}\colon \mathbb{S}^n \times (\mathbb{S}^n)^m \times \mathbb{R}^m \to \mathbb{S}^n$ that maps an SDP instance to a solution $\boldsymbol{X}$ while strictly respecting the problem's underlying symmetries. Specifically, we enforce three design principles.

1. **Symmetry** Since the solution resides in $\mathbb{S}_+^n$, for any symmetric inputs $\boldsymbol{C}$ and $\mathbf{A}$, the solution $\boldsymbol{X} \coloneqq \mathsf{NN}(\boldsymbol{C}, \mathbf{A}, \boldsymbol{b})$ must satisfy $\boldsymbol{X} = \boldsymbol{X}^\intercal$.

2. **Equivariance** For any permutation matrix $\boldsymbol{P} \in \{0, 1\}^{n \times n}$, transforming the inputs via $\boldsymbol{P}\boldsymbol{C}\boldsymbol{P}^\intercal$ and $\boldsymbol{P}\boldsymbol{A}_k\boldsymbol{P}^\intercal$ for all $k$ must result in the equivalently permuted output $\boldsymbol{P}\boldsymbol{X}\boldsymbol{P}^\intercal$.

3. **Invariance** For any permutation matrix $\boldsymbol{Q} \in \{0, 1\}^{m \times m}$ applied to the constraints, the output must remain unchanged: $\mathsf{NN}(\boldsymbol{C}, \boldsymbol{Q}\mathbf{A}, \boldsymbol{Q}\boldsymbol{b}) = \mathsf{NN}(\boldsymbol{C}, \mathbf{A}, \boldsymbol{b})$.

Since the primal variables are indexed by pairs $(i, j)$, they are structurally analogous to the colored 2-tuples in a higher-order graph. Guided by these principles, 2-WL and 2-FWL

emerge not merely as heuristics, but also as the natural inductive bias for this domain. Consequently, we can adapt the iterative color refinement to distinguish the variables of SDP instances. We formally introduce two variants, the VC-2-WL and VC-2-FWL, by extending the standard 2-WL and 2-FWL with slight modifications and the incorporation of constraint nodes. We define both variants because of the established hierarchy that 2-FWL $\sqsubset$ 2-WL in graph theory. This comparative approach allows us to theoretically *pinpoint* the exact level of expressivity required, i.e., whether the VC-2-WL suffices or whether joint structural modeling of VC-2-FWL is necessary.

Both variants initialize variable and constraint features according to Equation (2). The VC-2-WL update rule extends the standard 2-WL by integrating constraint node information alongside the structural aggregation, i.e.,

$$v_{ij}^t := \mathsf{hash}\Big(v_{ij}^{t-1}, \{\!\{v_{uj}^{t-1} \mid u \in [n]\}\!\}, \{\!\{v_{iu}^{t-1} \mid u \in [n]\}\!\},$$
$$\{\!\{(A_{k,ij}, c_k^{t-1}) \mid k \in N(ij)\}\!\}\Big) \qquad (3)$$
$$c_k^t := \mathsf{hash}\Big(c_k^{t-1}, \{\!\{(A_{k,ij}, v_{ij}^{t-1}) \mid (i,j) \in N(k)\}\!\}\Big).$$

The variable update aggregates row and column neighborhoods independently, reflecting standard 2-WL. However, this creates an asymmetry, as the hash function treats the row- and column-neighbor multisets as an *ordered* tuple, meaning the updated $v_{ij}$ and $v_{ji}$ are not guaranteed to be identical. To strictly enforce symmetry in output, we explicitly symmetrize the states: $v_{ji}^t \leftarrow v_{ij}^t$ for all $i < j$.

In contrast, the VC-2-FWL update couples the neighbor indices to capture joint interactions, i.e.,

$$v_{ij}^t := \mathsf{hash}\Big(v_{ij}^{t-1}, \{\!\{\{\!\{v_{uj}^{t-1}, v_{iu}^{t-1}\}\!\} \mid u \in [n]\}\!\},$$
$$\{\!\{(A_{k,ij}, c_k^{t-1}) \mid k \in N(ij)\}\!\}\Big) \qquad (4)$$
$$c_k^t := \mathsf{hash}\Big(c_k^{t-1}, \{\!\{(A_{k,ij}, v_{ij}^{t-1}) \mid (i,j) \in N(k)\}\!\}\Big).$$

Unlike standard 2-FWL, which aggregates a multiset of *ordered tuples*, our design aggregates a multiset of *unordered multisets* $\{\!\{v_{uj}^t, v_{iu}^t\}\!\}$. This modification ensures the updated $v_{ij}^t = v_{ji}^t$ without the need for manual symmetrization. For theoretical comparison, we define a tuple-based version, VC-2-FWL+, in Appendix E.1. We establish the following strict hierarchy of expressivity, see Appendix C.2 for the detailed proof, i.e.,

VC-2-FWL+ $\sqsubset$ {VC-2-FWL $\not\equiv$ VC-2-WL} $\sqsubset$ VC-WL.

That is, VC-2-FWL+ strictly refines VC-2-FWL and VC-2-WL, while VC-2-FWL and VC-2-WL are incomparable, and they both strictly refine VC-WL. Guided by this hierarchy, we first examine the limitations of the intermediate variant, VC-2-WL. Despite strictly refining the VC-WL, we demonstrate that it remains insufficient.

**Proposition 2.2.** *The* VC-2-WL *fails to represent linear SDP solutions. That is, there exist instances where the stable colors under* VC-2-WL *satisfy* $v_{ij}^\infty = v_{pq}^\infty$ *for distinct indices* $(i,j)$ *and* $(p,q)$, *yet the unique optimal solution entries differ, i.e.,* $X_{ij}^* \neq X_{pq}^*$.

However, we argue that VC-2-FWL+ is *overly expressive* by proving in Appendix C.4 that VC-2-FWL is sufficiently expressive for this domain. Formally,

**Theorem 2.3.** *Let* $X^* \in \mathbb{S}_+^n$ *be the primal optimal solution to a given SDP instance and given indices* $(i,j), (p,q)$. *If the stable colorings of* VC-2-FWL *satisfy* $v_{ij}^\infty = v_{pq}^\infty$, *then the solution values satisfy* $X_{ij}^* = X_{pq}^*$.

In practice, excessive expressivity can introduce unnecessarily too many colors or features, potentially causing the neural network to overfit. On the contrary, by discarding unnecessary directional information, our design yields a smaller set of colors which facilitates faster convergence, and encourages the neural network to learn more robust and generalizable embeddings.

In summary, both VC-2-WL and VC-2-FWL satisfy our three design principles: symmetric outputs, permutation equivariance regarding $C, \mathbf{A}$, and invariance to constraint ordering. Crucially, VC-2-FWL provides the expressivity level required to solve the problem.

### 2.3. Complexity

We analyze the computational complexity of VC-2-FWL by mapping it to the standard 1-WL operating on an implicit auxiliary graph. By leveraging tight complexity bounds from Berkholz et al. (2017), we have the following proposition (see Appendix C.5 for proof):

**Proposition 2.4.** *Given an SDP of* $n \times n$ *variables and* $m$ *constraints,* $nnz(\mathbf{A})$ *denotes the number of non-zero entries in* $\mathbf{A}$, *then* VC-2-FWL *converges in time* $\mathcal{O}\left((n^3 + nnz(\mathbf{A})) \log n\right)$.

The derived complexity is highly competitive for practical applications. In many applications, such as max-cut, the constraint tensor is *sparse*, rendering the nnz($\mathbf{A}$) term negligible. Notably, our total complexity is comparable to the fastest implementation of IPM for SDPs (Jiang et al., 2020), which has complexity $\mathcal{O}(n^{3.5} \log \frac{1}{\epsilon})$. Besides, the per-iteration of our VC-2-FWL is $\mathcal{O}(n^3 + nnz(\mathbf{A}))$, which is comparable to any SDP solver per iteration (Wen et al., 2010; Jiang et al., 2020; Wang et al., 2024), as they all require spectral decomposition of complexity $\mathcal{O}(n^3)$. A critical advantage of our approach, however, is that the neuralized VC-2-FWL being highly compatible for parallelization on GPUs.

## 2.4. Expressive neural architecture

Based on the initialization functions of Equation (2) and the update functions of Equation (4), we can neuralize the VC-2-FWL using multi-layer perceptrons (MLPs) and instantiate a neural network, which we name VC-2-FMPNN.

The feature initialization is

$$
\begin{aligned}
\boldsymbol{h}_k^0 &:= \mathsf{INIT}_{\mathrm{c}}(b_k) \in \mathbb{R}^d \\
\boldsymbol{h}_{ij}^0 &:= \mathsf{INIT}_{\mathrm{v}}(C_{ij}, \mathbb{I}_{i=j}) \in \mathbb{R}^d
\end{aligned}
\tag{5}
$$

and the update function

$$
\begin{aligned}
\boldsymbol{m}_{ij,\mathrm{v}\to\mathrm{v}}^t &:= \sum_{u\in[n]} \mathsf{MSG}_{\mathrm{v}\to\mathrm{v}}^t \left( \mathsf{MAP}^t\left(\boldsymbol{h}_{uj}^{t-1}\right) + \mathsf{MAP}^t\left(\boldsymbol{h}_{iu}^{t-1}\right)\right) \\
\boldsymbol{m}_{ij,\mathrm{c}\to\mathrm{v}}^t &:= \sum_{k\in N(ij)} \mathsf{MSG}_{\mathrm{c}\to\mathrm{v}}^t \left(A_{k,ij}, \boldsymbol{h}_k^{t-1}\right) \\
\boldsymbol{m}_{k,\mathrm{v}\to\mathrm{c}}^t &:= \sum_{(i,j)\in N(k)} \mathsf{MSG}_{\mathrm{v}\to\mathrm{c}}^t \left(A_{k,ij}, \boldsymbol{h}_{ij}^{t-1}\right) \\
\boldsymbol{h}_{ij}^t &:= \mathsf{UPD}_{\mathrm{v}}^t \left(\boldsymbol{h}_{ij}^{t-1}, \boldsymbol{m}_{ij,\mathrm{v}\to\mathrm{v}}^t, \boldsymbol{m}_{ij,\mathrm{c}\to\mathrm{v}}^t\right) \\
\boldsymbol{h}_k^t &:= \mathsf{UPD}_{\mathrm{c}}^t \left(\boldsymbol{h}_k^{t-1}, \boldsymbol{m}_{k,\mathrm{v}\to\mathrm{c}}^t\right)
\end{aligned}
\tag{6}
$$

where INIT, MSG, MAP, UPD are MLPs. We now formally establish that this neural architecture possesses the capacity to fully simulate the VC-2-FWL algorithm, thereby inheriting its theoretical expressivity.

**Proposition 2.5.** *There exists a set of parameters for the functions* INIT, UPD, MSG, MAP, *such that* VC-*2*-FMPNN *has maximal expressivity equal to the* VC-*2*-FWL.

See to Appendix C.6 for a proof and how we also neuralize VC-WL and VC-2-WL to VC-MPNN and VC-2-MPNN.

# 3. Empirical results

In the following, we aim to assess the extent to which our theoretical results translate to practice. Concretely, we answer the following questions.

**Q1 Expressivity** Do the theoretical hierarchies of expressivity translate into approximation performance on SDP?

**Q2 Efficiency** Can the proposed neural architecture accelerate the solving process for state-of-the-art solvers?

The implementation of our neural methods and baselines can be accessed at https://github.com/chendiqian/GNN4SDP.

## 3.1. Experimental protocol

Here, we outline the architectural configuration and details on training and evaluation. Before presenting our empirical

findings, it is important to clarify the scope of our evaluation. The primary objective of this work is to develop neural architectures capable of approximating solutions to general linear SDPs, rather than serving as an end-to-end solver for CO problems. Consequently, our main experiments evaluate the quality of the predicted SDP continuous solutions against exact SDP solvers. While mapping these continuous embeddings back to discrete integer solutions is beyond our core focus, we have included downstream evaluations comparing our approximated SDP solutions to exact CO targets in Appendix X for completeness.

**Neural architecture** We evaluate the approximation performance of VC-2-FMPNN for solving SDPs against theoretically weaker baselines, i.e., VC-MPNN and VC-2-MPNN, to verify whether empirical results align with our expressivity hierarchy. Besides, there are a few interesting neural architectures. The $\delta$-VC-2-MPNN and VC-2-IGN, adapted from $\delta$-2-WL (Morris et al., 2020) and 2-IGN (Maron et al., 2019b) respectively, are similar in form to VC-2-MPNN, see Appendix E.2 and Appendix E.3 for their definition and expressivity analysis. Furthermore, we benchmark against the VC-ET, a variant of Edge Transformer (ET) with expressivity equivalent to 2-FWL (Müller et al., 2024), see Appendix E.4 for details.

**Datasets** We employ synthetic SDP problems as relaxations of classical CO problems, i.e., max-cut, max-clique, max independent set (MIS), vertex cover, and max 2-SAT, as well as linear matrix inequality (LMI) problems from control theory. Detailed problem formulations are provided in Appendix D. For each problem class, we generate 10 000 instances and partition them into training, validation, and test sets with 8:1:1 ratio. For the graph-based tasks, i.e., max-cut, max-clique, MIS, and vertex cover, we construct instances from Erdős–Rényi graphs (Erdős & Rényi, 1959) with 100 nodes and edge probability of 0.1, resulting in SDP problems with 10 000 variables. To specifically challenge the expressivity of weaker baselines and expose approximation gaps, we also curate a more difficult max-cut dataset comprising 10-regular graphs with 100 nodes. Beyond the graph problems, the suite includes max 2-SAT instances with 100 variables and 1000 clauses, as well as LMI control problems with 2500 variables and 500 constraints. We emphasize that our experimental design prioritizes learnability and structural expressivity over scalability. While the instance sizes are moderate, they are sufficient to: (i) reveal the limitations of weaker architectures and (ii) demonstrate significant acceleration for classical SDP solvers.

To demonstrate real-world applicability beyond synthetic benchmarks, we also evaluate our method on real-world instances from the SDPLIB (Borchers, 1999). We select the class of max-cut problems, which consists of 13 instances

*Table 1.* Loss and relative objective gap (%) on test set. The best results across all are highlighted.

| Target | Model | Problems | | | | | |
| --- | --- | --- | --- | --- | --- | --- | --- |
| | | Max-Cut | Max-Cut (reg) | Max-Clique | MIS | Vertex Cover | Max 2-SAT |
| | VC-MPNN | $0.216\pm_{0.000}$ | $0.221\pm_{0.000}$ | $1.016e\text{-}5\pm_{0.000}$ | $9.945e\text{-}6\pm_{0.000}$ | $0.099\pm_{0.000}$ | $0.217\pm_{0.000}$ |
| | VC-2-MPNN | $0.119\pm_{0.018}$ | $0.221\pm_{0.000}$ | $5.108e\text{-}6\pm_{0.000}$ | $5.032e\text{-}6\pm_{0.000}$ | $0.029\pm_{0.001}$ | $0.053\pm_{0.009}$ |
| Test loss | $\delta$-VC-2-MPNN | $0.014\pm_{0.015}$ | $0.026\pm_{0.003}$ | $5.395e\text{-}6\pm_{0.000}$ | $5.058e\text{-}6\pm_{0.000}$ | $0.028\pm_{0.001}$ | $0.038\pm_{0.009}$ |
| | VC-2-IGN | $0.215\pm_{0.000}$ | $0.221\pm_{0.000}$ | $6.310e\text{-}6\pm_{0.000}$ | $6.120e\text{-}6\pm_{0.000}$ | $0.030\pm_{0.002}$ | $0.082\pm_{0.007}$ |
| | VC-2-FMPNN | $\mathbf{5.515e\text{-}5}\pm_{\mathbf{0.000}}$ | $\mathbf{5.140e\text{-}5}\pm_{\mathbf{0.000}}$ | $\mathbf{5.197e\text{-}7}\pm_{\mathbf{0.000}}$ | $\mathbf{4.772e\text{-}7}\pm_{\mathbf{0.000}}$ | $\mathbf{0.001}\pm_{\mathbf{0.000}}$ | $\mathbf{0.001}\pm_{\mathbf{0.000}}$ |
| | VC-ET | $0.0001\pm_{0.000}$ | $0.0002\pm_{0.000}$ | $2.906e\text{-}6\pm_{0.000}$ | $1.895e\text{-}6\pm_{0.000}$ | $0.013\pm_{0.008}$ | $0.017\pm_{0.011}$ |
| | VC-MPNN | $1.297\pm_{0.003}$ | $1.226\pm_{0.001}$ | $1.228\pm_{0.005}$ | $1.148\pm_{0.006}$ | $1.048\pm_{0.039}$ | $2.044\pm_{0.166}$ |
| | VC-2-MPNN | $1.046\pm_{0.073}$ | $1.225\pm_{0.001}$ | $0.889\pm_{0.023}$ | $0.858\pm_{0.063}$ | $1.235\pm_{0.050}$ | $0.882\pm_{0.053}$ |
| Raw obj gap (%) | $\delta$-VC-2-MPNN | $0.835\pm_{0.041}$ | $0.841\pm_{0.002}$ | $0.915\pm_{0.014}$ | $0.882\pm_{0.035}$ | $1.203\pm_{0.031}$ | $\mathbf{0.823}\pm_{\mathbf{0.093}}$ |
| | VC-2-IGN | $1.282\pm_{0.005}$ | $1.225\pm_{0.001}$ | $1.305\pm_{0.071}$ | $1.224\pm_{0.121}$ | $1.274\pm_{0.067}$ | $0.918\pm_{0.084}$ |
| | VC-2-FMPNN | $\mathbf{0.126}\pm_{\mathbf{0.003}}$ | $\mathbf{0.111}\pm_{\mathbf{0.009}}$ | $\mathbf{0.438}\pm_{\mathbf{0.036}}$ | $\mathbf{0.412}\pm_{\mathbf{0.026}}$ | $\mathbf{0.429}\pm_{\mathbf{0.044}}$ | $0.901\pm_{0.201}$ |
| | VC-ET | $0.142\pm_{0.043}$ | $0.125\pm_{0.038}$ | $0.659\pm_{0.104}$ | $0.525\pm_{0.124}$ | $0.503\pm_{0.019}$ | $1.149\pm_{0.262}$ |
| | VC-MPNN | $26.027\pm_{0.144}$ | $24.722\pm_{0.018}$ | $1.217\pm_{0.006}$ | $1.139\pm_{0.005}$ | $4.948\pm_{0.108}$ | $29.341\pm_{0.254}$ |
| | VC-2-MPNN | $13.662\pm_{2.078}$ | $24.729\pm_{0.019}$ | $0.884\pm_{0.020}$ | $0.852\pm_{0.059}$ | $1.478\pm_{0.265}$ | $7.173\pm_{0.128}$ |
| Proj. obj gap (%) | $\delta$-VC-2-MPNN | $1.929\pm_{0.021}$ | $3.781\pm_{0.045}$ | $0.906\pm_{0.016}$ | $0.874\pm_{0.033}$ | $1.321\pm_{0.062}$ | $4.208\pm_{1.222}$ |
| | VC-2-IGN | $24.796\pm_{0.007}$ | $24.737\pm_{0.008}$ | $1.344\pm_{0.035}$ | $1.242\pm_{0.145}$ | $1.677\pm_{0.447}$ | $10.607\pm_{0.927}$ |
| | VC-2-FMPNN | $\mathbf{0.111}\pm_{\mathbf{0.017}}$ | $\mathbf{0.092}\pm_{\mathbf{0.004}}$ | $\mathbf{0.436}\pm_{\mathbf{0.035}}$ | $\mathbf{0.411}\pm_{\mathbf{0.025}}$ | $\mathbf{0.469}\pm_{\mathbf{0.051}}$ | $\mathbf{0.862}\pm_{\mathbf{0.211}}$ |
| | VC-ET | $0.159\pm_{0.043}$ | $0.128\pm_{0.032}$ | $0.647\pm_{0.102}$ | $0.523\pm_{0.135}$ | $0.510\pm_{0.015}$ | $1.104\pm_{0.146}$ |

that are homogeneous in problem structure but vary significantly in scale, as detailed in Table 9. Due to the limited number of available instances, the standard train-validation split is infeasible. Instead, we treat this as a direct test of *trainability*, i.e., assessing whether the architectures can overfit and effectively predict the near-optimal solution.

**Training and evaluation**   All models were trained using the Adam optimizer (Kingma & Ba, 2015) with default hyperparameters and batch size of 256. The training objective is to minimize the supervised mean squared error (MSE) between the predicted and ground-truth solution matrices. Training for synthetic datasets were conducted on a compute cluster with four NVIDIA L40S GPUs, whereas SDPLIB utilized a single GPU. We trained for at most 1000 epochs, using a learning rate scheduler that decays the learning rate after 100 consecutive epochs without improvement in the relative objective gap on the validation set. Additionally, early stopping was applied if this metric failed to improve for 200 consecutive epochs.

To quantify the computational efficiency of neural SDP solvers, we benchmarked inference time against classical solvers using the max-cut problem as a representative case study. We compared against SDPLR, a solver based on Burer-Monteiro method (Burer & Monteiro, 2003), MOSEK (ApS, 2022), an IPM solver, and SCS (O'Donoghue, 2021), a first-order solver based on the ADMM algorithm (Wen et al., 2010). SDPLR and MOSEK were executed on an Intel® Xeon® Silver 4510 CPU, as they lack native GPU support. Both the SCS solver and all the neural architectures were executed on a single NVIDIA L40S GPU. Additionally, we investigate the practical utility of our approach by using the VC-2-FMPNN prediction to warm-start the SCS solver.

For all experiments, including model training and solver

timing, we compute evaluation metrics, specifically loss and objective gap, by averaging over the full dataset. We repeat this process using five different random seeds and report the resulting mean and standard deviation.

### 3.2. Results and discussion

In the following, we present results and discuss them.

**Approximation performance**   The results of test loss and relative objective gap are reported in Table 1. The optimal objective values for LMI are 0, so we report the predicted objective in Table 6. We first examine the test loss. Consistent with the theoretical hierarchy, our VC-2-FMPNN and equally powerful VC-ET achieve the lowest losses across all problems, suggesting that 2-FWL-like updates are essential for SDP approximation. This advantage is most significant on max-cut, where VC-2-FMPNN outperforms VC-MPNN, VC-2-MPNN, $\delta$-VC-2-MPNN, and VC-2-IGN by orders of magnitude, revealing a clear expressivity hierarchy and the incapacity in those weaker baselines. To further evaluate the quality of the predicted solutions, we compute the relative objective gap over the test set $\mathcal{D}$, defined as

$$\frac{1}{|\mathcal{D}|} \sum_{I \in \mathcal{D}} \left| \frac{(\text{obj}(I) - \text{obj}^*(I))}{\text{obj}^*(I)} \right| \cdot 100\%,$$

where $\text{obj}(I)$ denotes the predicted objective value and $\text{obj}^*(I)$ the optimal value for instance $I$. Furthermore, the predicted solution matrix $\boldsymbol{X}$ can be projected onto the PSD cone, as shown in Algorithm 1. The results in Table 1 mirror the test loss trends. For the raw objective gap, VC-2-FMPNN outperforms almost all baselines across all tasks, except slightly worse than $\delta$-VC-2-MPNN on max 2-SAT. However, the objective gaps for VC-2-FMPNN and VC-ET after PSD projection remain low, indicating that

their predictions are not only accurate in objective value but also geometrically close to the PSD cone. In sharp contrast, models with weaker expressivity often produce solutions far from that. This is evidenced by the drastic degradation in their performance after projection. For instance, the gap for VC-MPNN on max-cut rockets from $\sim 1.3\%$ to over $26\%$, with similar collapses observed on max 2-SAT. Interestingly, VC-MPNN, VC-2-MPNN and VC-2-IGN fail on the regular graph dataset, producing identical losses and gaps, but $\delta$-VC-2-MPNN shows significant advantage over them, reflecting our theory of their hierarchy in Appendix E.2.

*Table 2.* Mean absolute residuals on constraints of max-cut problems on test set, repeated with 5 seeds.

| Model | Cons. vio. | Proj. cons. vio. |
|---|---|---|
| VC-MPNN | **0.001**±**0.000** | 0.298±0.001 |
| VC-2-MPNN | 0.016±0.006 | 0.180±0.001 |
| $\delta$-VC-2-MPNN | 0.006±0.001 | 0.069±0.019 |
| VC-2-IGN | **0.001**±**0.000** | 0.304±0.001 |
| VC-2-FMPNN | 0.003±0.001 | **0.003**±**0.001** |
| VC-ET | **0.001**±**0.000** | **0.003**±**0.001** |

Besides loss and objective, we also probe the structural feasibility of the predictions by measuring the mean absolute residuals on constraints for each problem instance:

$$\frac{1}{m} \sum_{k \in [m]} |\langle \boldsymbol{A}_k, \boldsymbol{X} \rangle - b_k|$$

and averaged over the dataset. We show the results on max-cut problem as representative in Table 2. While some weaker methods show low level of constraint violation, such as VC-MPNN and VC-2-IGN, the projected solutions exhibit a huge gap, indicating their predicted solutions are far from feasible region. In comparison, VC-2-FMPNN maintains negligible violation ($\sim 0.003$) both before and after projection, demonstrating that it successfully approximates the true geometry of the feasible solution. For more results on constraint violation, see Table 11.

We compare VC-MPNN, VC-2-MPNN and VC-2-FMPNN on SDPLIB. Table 3 reports the training MSE loss. For more complete results, see Table 10. The results reveal the dramatic gap in trainability: VC-MPNN and VC-2-MPNN fail to even overfit the training data, observing from the high losses. In contrast, VC-2-FMPNN achieves negligible training loss around $10^{-4}$ magnitude.

Overall, these results align closely with our theory: VC-2-FMPNN and VC-ET, with VC-2-FWL expressive power, not only approximate objective values more accurately but also produce solutions closer to the feasible region, showing their ability to learn high-quality SDP solutions.

*Table 3.* Training loss on SDPLIB instances. The best results across all are highlighted.

| Name | Loss | | |
|---|---|---|---|
| | VC-MPNN | VC-2-MPNN | VC-2-FMPNN |
| MCP100 | 0.203±0.000 | 0.199±0.004 | **1.974e-4**±**0.000** |
| MCP124-1 | 0.225±0.000 | 0.224±0.000 | **1.546e-4**±**0.000** |
| MCP124-2 | 0.220±0.000 | 0.218±0.002 | **2.303e-4**±**0.000** |
| MCP124-3 | 0.215±0.000 | 0.214±0.002 | **3.076e-4**±**0.000** |
| MCP124-4 | 0.270±0.000 | 0.272±0.007 | **3.374e-4**±**0.000** |
| MCP250-1 | 0.215±0.000 | 0.215±0.000 | **3.034e-4**±**0.000** |
| MCP250-2 | 0.158±0.015 | 0.160±0.007 | **4.176e-4**±**0.000** |
| MCP250-3 | 0.155±0.000 | 0.156±0.000 | **6.266e-4**±**0.000** |
| MCP250-4 | 0.182±0.013 | 0.186±0.004 | **5.892e-4**±**0.000** |
| MCP500-1 | 0.118±0.000 | 0.118±0.001 | **4.536e-4**±**0.000** |
| MCP500-2 | 0.123±0.000 | 0.124±0.001 | **4.948e-4**±**0.000** |
| MCP500-3 | 0.121±0.000 | 0.122±0.000 | **6.745e-4**±**0.000** |
| MCP500-4 | 0.126±0.000 | 0.130±0.003 | **9.761e-4**±**0.000** |

*Table 4.* Timing (in seconds) on various sizes of max-cut problem.

| Model | Sizes | | |
|---|---|---|---|
| | $50 \times 50$ | $100 \times 100$ | $150 \times 150$ |
| SDPLR | 0.111±0.050 | 0.195±0.041 | 0.285±0.062 |
| MOSEK | 0.206±0.008 | 4.206±0.151 | 28.839±1.814 |
| SCS | 0.061±0.004 | 0.386±0.028 | 3.357±0.240 |
| SCS (Warm s.) | 0.042±0.005 | 0.280±0.041 | 2.092±0.298 |
| VC-MPNN | **0.006**±**0.000** | **0.006**±**0.000** | **0.008**±**0.000** |
| VC-2-MPNN | 0.009±0.000 | 0.009±0.000 | 0.012±0.000 |
| $\delta$-VC-2-MPNN | 0.011±0.000 | 0.012±0.002 | 0.017±0.003 |
| VC-2-IGN | 0.010±0.000 | 0.010±0.000 | 0.015±0.000 |
| VC-2-FMPNN | 0.009±0.000 | 0.010±0.000 | 0.013±0.000 |
| VC-ET | 0.010±0.000 | 0.020±0.000 | 0.050±0.001 |

**Timing** As detailed in Table 4, classical solvers' runtime scales aggressively with problem size, especially MOSEK. In contrast, neural inference time is negligible and exhibit minimal growth, due to GPU parallelism. Among the neural architectures, VC-2-FMPNN exhibits an optimal balance between efficiency and expressivity. It is only marginally slower than VC-MPNN, yet significantly faster than VC-ET, particularly as problem size increases. Finally, the warm-start experiments demonstrate the practical value of VC-2-FMPNN: initializing SCS with its prediction consistently reduces convergence time, e.g., from 3.35s to 2.09s on the largest instances, highlighting the potential application of neural approximation into classical optimization. For more timing results on SCS and our neural approach, see Table 11.

We extend our timing analysis to the SDPLIB, where problem difficulty varies significantly. Table 5 shows the timing of the SCS solver versus our VC-2-FMPNN inference, as well as the performance gains achieved by using the neural prediction to warm-start the solver. The results underscore the scalability of our approach. While the classical solver's runtime grows rapidly over 5 minutes for the

*Table 5.* Timing (in seconds) and warm start the solver on SDPLIB.

| Name | SCS | VC-2-FMPNN | Warm start | Improvement |
|---|---|---|---|---|
| MCP100 | $0.560_{\pm 0.005}$ | $0.015_{\pm 0.001}$ | $0.287_{\pm 0.005}$ | 48.7% |
| MCP124-1 | $3.243_{\pm 0.006}$ | $0.016_{\pm 0.000}$ | $0.623_{\pm 0.067}$ | **80.7%** |
| MCP124-2 | $1.152_{\pm 0.003}$ | $0.016_{\pm 0.000}$ | $0.532_{\pm 0.002}$ | 53.8% |
| MCP124-3 | $0.759_{\pm 0.002}$ | $0.016_{\pm 0.000}$ | $0.467_{\pm 0.050}$ | 38.4% |
| MCP124-4 | $0.546_{\pm 0.005}$ | $0.016_{\pm 0.000}$ | $0.484_{\pm 0.056}$ | 11.3% |
| MCP250-1 | $17.274_{\pm 0.044}$ | $0.024_{\pm 0.000}$ | $4.717_{\pm 0.035}$ | 72.6% |
| MCP250-2 | $12.308_{\pm 0.035}$ | $0.024_{\pm 0.000}$ | $5.971_{\pm 1.029}$ | 51.5% |
| MCP250-3 | $6.460_{\pm 0.017}$ | $0.024_{\pm 0.000}$ | $3.302_{\pm 0.013}$ | 48.8% |
| MCP250-4 | $4.556_{\pm 0.011}$ | $0.024_{\pm 0.000}$ | $3.316_{\pm 0.035}$ | 27.2% |
| MCP500-1 | $328.471_{\pm 1.236}$ | $0.147_{\pm 0.000}$ | $77.471_{\pm 0.796}$ | 76.4% |
| MCP500-2 | $217.164_{\pm 1.629}$ | $0.151_{\pm 0.005}$ | $68.822_{\pm 0.849}$ | 68.3% |
| MCP500-3 | $134.705_{\pm 0.198}$ | $0.147_{\pm 0.000}$ | $40.586_{\pm 0.542}$ | 69.8% |
| MCP500-4 | $81.779_{\pm 0.944}$ | $0.148_{\pm 0.000}$ | $39.425_{\pm 0.242}$ | 51.8% |

hardest instance (MCP500-1), the VC-2-FMPNN inference time remains negligible, completing under 0.15 seconds across all cases. More importantly, initializing SCS with the VC-2-FMPNN prediction consistently accelerates convergence, yielding runtime reductions up to 80.7%. This confirms that VC-2-FMPNN is able to capture high-quality approximation of the global optimum, significantly reducing the workload for the iterative solver.

## 4. Conclusion

We studied the expressive power required of GNNs to predict optimal solutions of linear SDPs, using the unique minimum-Frobenius-norm solution as the learning target. We proved that VC-WL and higher-order VC-2-WL can fail on SDPs, whereas VC-2-FWL-equivalent architectures are sufficient because they can emulate PDHG solver updates until convergence. Empirically, these more expressive architectures achieve the lowest prediction error and the smallest objective gaps across synthetic and real-world benchmarks. Moreover, their predictions can warm-start a first-order solver (SCS), yielding substantial speedups on SDPLIB instances. *Overall, our results sharpen the understanding of the expressivity required to learn solutions for a large, practically important class of convex optimization problems.*

## Impact Statement

This paper presents work aimed at advancing the field of machine learning. There are many potential societal consequences of our work, none of which we feel must be specifically highlighted here.

## Acknowledgements

Christopher Morris and Chendi Qian are partially funded by a DFG Emmy Noether grant (468502433) and RWTH Junior Principal Investigator Fellowship under Germany's Excellence Strategy. We thank Erik Müller for crafting the figures.

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

# A. Limitations and future directions

A key limitation of our study is that the required 2-FWL-level expressivity comes at a nontrivial computational cost, as it requires considering every variable pair. In addition, our theoretical results demonstrate only that there exist parameter assignments that allow recovery of the optimal low-norm solution, and do not guarantee that gradient-descent-based methods will converge. In addition, our expressivity guarantees are non-uniform, in the sense that we only show the existence of a parameter for a given SDP problem instance size and do not shed light on the architectures' size-generalization capabilities (Levin et al., 2025).

*Looking forward*, we identify three key directions. First, we propose designing efficient architectures for specific SDP subclasses where lower-order expressivity suffices. Second, the rigorous alignment between our architecture and the PDHG algorithm opens a promising avenue for self-supervised primal-dual learning (Park & Van Hentenryck, 2023; Tanneau & Van Hentenryck, 2024). Rather than relying on supervised learning, future models could be trained by directly minimizing algorithmic residuals or duality gaps. Finally, we aim to extend this framework to broader conic programs and integrate these learned components into combinatorial optimization pipelines, such as branch-and-cut frameworks.

# B. More background

## B.1. Higher order Weisfeiler–Leman

We introduce some standard notations first. For $n \geq 1$, let $[n] := \{1, \ldots, n\} \subset \mathbb{N}$. We use $\{\!\{\ldots\}\!\}$ to denote multisets, i.e., the generalization of sets allowing for multiple instances of each of its elements. A *graph $G$* is a pair $(V(G), E(G))$ with *finite* sets of *vertices* $V(G)$ and *edges* $E(G) \subseteq \{\{u, v\} \subseteq V(G) \mid u \neq v\}$. For ease of notation, we denote the edge $\{u, v\}$ in $E(G)$ by $(u, v)$ or $(v, u)$. A *labeled graph $G$* is a triple $(V, E, l)$ with *(node) coloring* or *label function* $l \colon V(G) \to \mathbb{N}$. The *neighborhood* of $v$ in $G$ is denoted by $N(v) := \{u \in V(G) \mid \{v, u\} \in E(G)\}$.

Two graphs $G$ and $H$ are *isomorphic* and we write $G \simeq H$ if there exists a bijection $\varphi \colon V(G) \to V(H)$ that preserves the adjacency relation, i.e., $(u, v) \in E(G) \iff (\varphi(u), \varphi(v)) \in E(H)$. Then $\varphi$ is an *isomorphism* between $G$ and $H$. In the case of labeled graphs, we additionally require that $l(v) = l(\varphi(v))$ for $v$ in $V(G)$. We further define the atomic type $\mathsf{atp} \colon V(G)^k \to \mathbb{N}$ such that $\mathsf{atp}(\boldsymbol{v}) = \mathsf{atp}(\boldsymbol{w})$ for $\boldsymbol{v}, \boldsymbol{w} \in V(G)^k$ if and only if the mapping $\varphi \colon V(G)^k \to V(G)^k$ where $v_i \to w_i$ induces a partial isomorphism, i.e., we have $v_i = v_j \iff w_i = w_j$ and $(v_i, v_j) \in E(G) \iff (\varphi(v_i), \varphi(v_j)) \in E(G)$.

The 1-WL or color refinement is a simple heuristic for the graph isomorphism problem, originally proposed by Weisfeiler & Leman (1968). Intuitively, the algorithm determines if two graphs are non-isomorphic by iteratively coloring or labeling vertices. Given an initial coloring or labeling of the vertices of both graphs, e.g., their degree or application-specific information, in each iteration, two vertices with the same label get different labels if the number of identically labeled neighbors is not equal. If, after some iteration, the number of vertices annotated with a specific label is different in both graphs, the algorithm terminates and a stable coloring (partition) is obtained. We can then conclude that the two graphs are not isomorphic. It is easy to see that the algorithm cannot distinguish all non-isomorphic graphs (Cai et al., 1992).

Formally, let $G = (V, E, l)$ be a labeled graph. In each iteration $t + 1$, the 1-WL computes a node coloring for node $v$ as $\boldsymbol{c}_v^t$, which depends on the coloring of the neighbors. That is,

$$\boldsymbol{c}_v^{t+1} := \mathsf{hash}\left(\boldsymbol{c}_v^t, \{\!\{\boldsymbol{c}_u^t \mid u \in N(v)\}\!\}\right),$$

where hash injectively maps the above pair to a unique natural number, which has not been used in previous iterations. In iteration 0, the coloring $\boldsymbol{c}_v^0 := l(v)$.

To test if two graphs $G$ and $H$ are non-isomorphic, we run the above algorithm in "parallel" on both graphs. If the two graphs have a different number of vertices colored $c$ in $\mathbb{N}$ at some iteration, the 1-WL *distinguishes* the graphs as non-isomorphic. Moreover, if

$$\boldsymbol{c}_v^t = \boldsymbol{c}_w^t \iff \boldsymbol{c}_v^{t+1} = \boldsymbol{c}_w^{t+1},$$

for all vertices $v$ and $w$ in $V(G)$, the algorithm terminates. For such $t$, we define the *stable coloring* $\boldsymbol{c}_v^\infty = \boldsymbol{c}_v^t$ for $v$ in $V(G)$.

Due to the shortcomings of the 1-WL or color refinement in distinguishing non-isomorphic graphs, several researchers (Babai, 1979; Immerman & Lander, 1990; Cai et al., 1992; Grohe, 2017; Morris et al., 2019; Grohe, 2021) devised a more powerful generalization of the former, today known as the *$k$-dimensional Weisfeiler-Leman algorithm* ($k$-WL). There are two versions

of $k$-WL algorithm, known as *Folklore Weisfeiler-Leman* ($k$-FWL) and *Oblivious Weisfeiler-Leman* that is often used in the graph learning community, both will be discussed in this section.

Intuitively, to surpass the limitations of the 1-WL, the $k$-FWL algorithm colors subgraphs instead of a single node. More precisely, given a graph $G$, it colors the tuples from $V(G)^k$ for $k \geq 2$ instead of the vertices. By defining a neighborhood between these tuples, we can define a coloring similar to the 1-WL. Formally, let $G$ be a graph, and let $k \geq 2$. In each iteration $t \geq 0$, the algorithm, similarly to the 1-WL, computes a *coloring* $c_v^t$ for a tuple $v$. In the first iteration, $t = 0$, the tuples $v, w \in V(G)^k$ get the same color if they have the same atomic type, i.e., $c_v^0 := \mathsf{atp}(v)$. The update of the coloring is defined by

$$c_v^{t+1} := \mathsf{hash}\left(c_v^t, \left\{\!\left\{\left(c_{\phi_1(v,w)}^t, \ldots, c_{\phi_k(v,w)}^t\right) \mid w \in V(G)\right\}\!\right\}\right),$$

where

$$\phi_j(v, w) := (v_1, \ldots, v_{j-1}, w, v_{j+1}, \ldots, v_k).$$

That is, $\phi_j(v, w)$ replaces the $j$-th component of the tuple $v$ with the node $w$. Hence, two tuples are *adjacent* or *$j$-neighbors*, with respect to a node $w$, if they are different in the $j$th component (or equal, in the case of self-loops).

$k$-WL differs from $k$-FWL in the way the $j$-neighbors are collected and aggregated. Specifically, the update of $k$-WL is

$$c_v^{t+1} := \mathsf{hash}\left(c_v^t, \left(\{\!\{c_{\phi_1(v,w)}^t \mid w \in V(G)\}\!\}, \cdots, \{\!\{c_{\phi_k(v,w)}^t \mid w \in V(G)\}\!\}\right)\right).$$

Again, we run the $k$-WL or $k$-FWL algorithm until convergence, i.e.,

$$c_v^t = c_w^t \iff c_v^{t+1} = c_w^{t+1},$$

for all $v$ and $w$ in $V(G)^k$ holds, and call the partition of $V(G)^k$ induced by $c^t$ the stable partition. For such $t$, we define the stable coloring $c_v^\infty = c_v^t$ for $v$ in $V(G)^k$. Hence, two tuples $v$ and $w$ with the same color in iteration $(t-1)$ get different colors in iteration $t$ if there exists a $j$ in $[k]$ such that the number of $j$-neighbors of $v$ and $w$, respectively, colored with a certain color is different.

To test whether two graphs $G$ and $H$ are non-isomorphic, we run the $k$-FWL or $k$-WL in "parallel" on both graphs. Then, if the two graphs have a different number of vertices colored $c$ in $\mathbb{N}$, the $k$-WL *distinguishes* the graphs as non-isomorphic.

Finally, we characterize the relative expressive power of these algorithms. We say that an algorithm $\mathcal{A}$ *refines* $\mathcal{B}$, denoted $\mathcal{A} \sqsubseteq \mathcal{B}$, if every pair of graphs distinguished by $\mathcal{B}$ is also distinguished by $\mathcal{A}$. And we say an algorithm $\mathcal{A}$ *strictly refines* $\mathcal{B}$, denoted $\mathcal{A} \sqsubset \mathcal{B}$ if there exists a pair of graphs that cannot be distinguished by $\mathcal{B}$ but $\mathcal{A}$. Observing the update rules, $k$-FWL aggregates the joint structure of neighbors across all tuple positions, whereas $k$-WL aggregates each position independently. Consequently, $k$-FWL is strictly more expressive than $k$-WL for $k \geq 2$. In fact, it has been established that $k$-FWL possesses the same expressive power as $(k+1)$-WL (Cai et al., 1992; Grohe, 2017). Since increasing the dimension $k$ strictly increases discriminative power (Cai et al., 1992), we obtain the following hierarchy of expressivity:

$$\cdots \equiv (k+1)\text{-FWL} \sqsubset (k+1)\text{-WL} \equiv k\text{-FWL} \sqsubset k\text{-WL} \equiv (k-1)\text{-FWL} \sqsubset \ldots$$

## C. Omitted proofs

**Proposition C.1** (Restatement of Proposition 1.1). *The SDP problem defined in Equation* (1) *has a unique primal solution $X^*$ with the minimum Frobenius norm $\|X^*\|_F^2$.*

*Proof for Proposition 1.1.* We prove by contradiction. Suppose there exists another feasible solution $X' \neq X^* \in \mathbb{S}_+^n$, such that $\mathcal{A}(X') = b$, and $\langle C, X' \rangle = \langle C, X^* \rangle$, and $\|X'\|_F^2 = \|X^*\|_F^2$. Note that we are not assuming $\|X'\|_F^2 < \|X^*\|_F^2$ or $\langle C, X' \rangle < \langle C, X^* \rangle$, otherwise the $X'$ will replace $X^*$ to be the optimal solution with least Frobenius norm. Assuming the existence of $X'$, we can find a third solution $X'' = \frac{1}{2}(X^* + X')$, which is feasible and optimal, in that:

- $X'' \in \mathbb{S}_+^n$, as $X^* \in \mathbb{S}_+^n$ and $X' \in \mathbb{S}_+^n$, and the PSD cone is a convex cone.

- $\mathcal{A}(X'') = \frac{1}{2}\mathcal{A}(X^*) + \frac{1}{2}\mathcal{A}(X') = b$, i.e., $X''$ is feasible w.r.t. the constraints.

- $\langle \boldsymbol{C}, \boldsymbol{X}'' \rangle = \frac{1}{2} \langle \boldsymbol{C}, \boldsymbol{X}^* \rangle + \frac{1}{2} \langle \boldsymbol{C}, \boldsymbol{X}' \rangle = \langle \boldsymbol{C}, \boldsymbol{X}^* \rangle = \langle \boldsymbol{C}, \boldsymbol{X}' \rangle$, i.e., $\boldsymbol{X}''$ has the same objective value.

Besides, $\|\boldsymbol{X}''\|_F^2 = \frac{1}{4}\|\boldsymbol{X}^* + \boldsymbol{X}'\|_F^2 \le \frac{\|\boldsymbol{X}^*\|_F^2 + \|\boldsymbol{X}'\|_F^2}{2} = \|\boldsymbol{X}^*\|_F^2 = \|\boldsymbol{X}'\|_F^2$, and equality holds only when $\boldsymbol{X}^* = \boldsymbol{X}'$. Since we assume $\boldsymbol{X}^* \ne \boldsymbol{X}'$, this implies $\boldsymbol{X}''$ has a strictly lower Frobenius norm than $\boldsymbol{X}^*$ and $\boldsymbol{X}''$, contradicting the assumption that $\boldsymbol{X}^*$ and $\boldsymbol{X}'$ are minimum norm solutions. Thus, the minimum norm solution must be unique. □

### C.1. Proof for failure of VC-WL and VC-2-WL

**Proposition C.2** (Restatement of Proposition 2.1)**.** *The* VC-WL *fails to represent linear SDP solutions. That is, there exist instances where the stable coloring of the* VC-WL *satisfies* $\boldsymbol{v}_{ij}^\infty = \boldsymbol{v}_{pq}^\infty$ *for distinct variable indices* $(i, j)$ *and* $(p, q)$, *yet the entries in the unique optimal solution differ, i.e.,* $X_{ij}^* \ne X_{pq}^*$.

*Proof for Proposition 2.1.* We prove by providing the following counter-example

$$\boldsymbol{C} = \begin{bmatrix} 1 & 1 & 0 \\ 1 & 1 & 0 \\ 0 & 0 & 1 \end{bmatrix}, \quad \boldsymbol{A}_1 = \begin{bmatrix} 0 & 1 & 0 \\ 1 & 0 & 0 \\ 0 & 0 & 0 \end{bmatrix}, \quad \boldsymbol{A}_2 = \begin{bmatrix} 0 & 0 & 1 \\ 0 & 0 & 0 \\ 1 & 0 & 0 \end{bmatrix}, \quad \boldsymbol{b} = \begin{bmatrix} 1 & 1 \end{bmatrix}$$

on which VC-WL converges to colors $\begin{bmatrix} a & b & c \\ b & a & d \\ c & d & a \end{bmatrix}$, where $\{a, b, c, \cdots\}$ are from an alphabet to represent colors. Note that the diagonal entries $(1, 1)$ and $(3, 3)$ [5] are assigned the same color $a$. However, the unique optimal solution yields distinct values $X_{11}^* \approx 0.707$ and $X_{33}^* \approx 0.354$. VC-WL fails because it treats variables independently and ignores the matrix structure. □

*Remark* C.3. From a measure-theoretic perspective, one might argue that a specific counterexample could merely reside on a set of measure zero, thereby still allowing a model to maintain universal approximation *almost everywhere*. However, this defense is inadequate for SDP relaxations of CO problems. First, the input distributions of CO problems are not uniformly drawn from $\mathbb{R}^{n \times n}$. Rather, they are heavily concentrated on specific discrete topological structures. Second, our identified failure cases are not isolated points. As established in Lemma C.24, scaling any failing instance by a scalar $\alpha > 0$ generates an uncountable infinite set (a ray) of failures. This specific construction represents just one of many systematic methods to expose these representational blind spots. Ultimately, the empirical collapse of the less expressive models observed in Table 1 confirms that this theoretical failure translates into a severe, practical expressivity limitation, rather than a negligible mathematical anomaly.

**Proposition C.4** (Restatement of Proposition 2.2)**.** *The* VC-2-WL *fails to represent linear SDP solutions. That is, there exist instances where the stable colors under* VC-2-WL *satisfy* $\boldsymbol{v}_{ij}^\infty = \boldsymbol{v}_{pq}^\infty$ *for distinct indices* $(i, j)$ *and* $(p, q)$, *yet the unique optimal solution entries differ* $X_{ij}^* \ne X_{pq}^*$.

*Proof for Proposition 2.2.* We prove this by construction, using a counterexample in which the objective matrix $\boldsymbol{C}$ forms a Latin square. Consider the following SDP instance with $n = 6$,

$$\boldsymbol{C} = \begin{bmatrix} 1 & 2 & 3 & 4 & 5 & 6 \\ 2 & 1 & 4 & 5 & 6 & 3 \\ 3 & 4 & 1 & 6 & 2 & 5 \\ 4 & 5 & 6 & 1 & 3 & 2 \\ 5 & 6 & 2 & 3 & 1 & 4 \\ 6 & 3 & 5 & 2 & 4 & 1 \end{bmatrix}, \quad \boldsymbol{A}_1 = \boldsymbol{I}_6, \quad \boldsymbol{b} = [1].$$

We focus on the entries $(1, 5)$ and $(2, 4)$. First, observe that their initial features are identical, i.e., $\boldsymbol{C}_{15} = \boldsymbol{C}_{24} = 5$, and both are non-diagonal entries. Secondly, observe that every row and column in $\boldsymbol{C}$ contains the set of values $\{1, \ldots, 6\}$. Consequently, for *any* index $(u, v)$, the row and column multisets are identical, i.e.,

$$\{\!\{ \boldsymbol{v}_{uk}^0 \mid k \in [n] \}\!\} = \{\!\{ \boldsymbol{v}_{kv}^0 \mid k \in [n] \}\!\} = \{\!\{ 1, 2, 3, 4, 5, 6 \}\!\}.$$

---

[5] All matrix indices in this paper are 1-indexed.

Since VC-2-WL aggregates row and column multisets independently as in Equation (3), and both the node features and their neighborhoods are identical, the algorithm cannot distinguish $(1, 5)$ from $(2, 4)$ at initialization or any subsequent iteration. Thus, $v_{15}^\infty = v_{24}^\infty$ under VC-2-WL algorithm. However, the unique minimum-norm solution yields $X_{15}^* \approx -0.115$ and $X_{24}^* \approx -0.172$. Hence, the inability of VC-2-WL to distinguish these variables indicates it is insufficient for SDPs. $\quad\square$

### C.2. Proof for expressivity hierarchy

We first compare VC-WL and VC-2-WL by showing the following result.

**Proposition C.5.** *The* VC-*2*-WL *algorithm strictly refines* VC-WL*, denoted as*

$$\text{VC-2-WL} \sqsubset \text{VC-WL}.$$

*Proof.* Observing from the update functions of Equation (2) and Equation (3), we notice that the update of constraint nodes is exactly the same. Now, consider the variable nodes. Observe that the information aggregated by the VC-WL hash function for a variable $v_{ij}$ consists solely of its constraint neighbors. By definition, the corresponding input for VC-2-WL includes this constraint information as a subset, strictly augmented by the structural row and column multisets.

We now formally prove VC-2-WL $\sqsubseteq$ VC-WL, that is, for any iteration $t$, if VC-2-WL fails to distinguish $v_{ij}^t$ and $v_{pq}^t$, so cannot VC-WL. First, for $t = 0$, the color $v_{ij}^0$ is basically the encoding of the problem $C_{ij}$, therefore if for VC-2-WL: $v_{ij}^0 = v_{pq}^0$, it also holds for VC-WL. Then, for all $t > 0$, we have, by definition of VC-2-WL:

$$
\begin{aligned}
& v_{ij}^t = v_{pq}^t \\
&\implies \text{hash}\Big(v_{ij}^{t-1}, \{\!\!\{ v_{uj}^{t-1} \mid u \in [n] \}\!\!\}, \{\!\!\{ v_{iu}^{t-1} \mid u \in [n] \}\!\!\}, \{\!\!\{ (A_{k,ij}, c_k^{t-1}) \mid k \in N(ij) \}\!\!\} \\
&= \text{hash}\Big(v_{pq}^{t-1}, \{\!\!\{ v_{uq}^{t-1} \mid u \in [n] \}\!\!\}, \{\!\!\{ v_{pu}^{t-1} \mid u \in [n] \}\!\!\}, \{\!\!\{ (A_{k,pq}, c_k^{t-1}) \mid k \in N(pq) \}\!\!\} \\
&\implies \text{hash}\Big(v_{ij}^{t-1}, \{\!\!\{ (A_{k,ij}, c_k^{t-1}) \mid k \in N(ij) \}\!\!\}\Big) \\
&= \text{hash}\Big(v_{pq}^{t-1}, \{\!\!\{ (A_{k,pq}, c_k^{t-1}) \mid k \in N(pq) \}\!\!\}\Big).
\end{aligned}
$$

Since this corresponds exactly to the update rule of VC-WL, it follows that VC-WL also cannot distinguish them. Thus, VC-2-WL $\sqsubseteq$ VC-WL.

To show the strict refinement, we pick the example in the proof of Proposition 2.1 where VC-WL fails to distinguish the variables $X_{11}^*$ and $X_{33}^*$. In contrast, VC-2-WL converges to $\begin{bmatrix} a & b & c \\ b & e & d \\ c & d & f \end{bmatrix}$, where $\{a, b, c, \cdots\}$ are from an alphabet to represent colors. Here, the diagonal entries are distinguished ($a \neq f$), allowing the VC-2-WL algorithm to distinguish the solutions.

Therefore, VC-2-WL $\sqsubset$ VC-WL. $\quad\square$

Similarly, VC-2-FWL $\sqsubset$ VC-WL can be proven in exactly the same way with the same example; we omit the proof.

Next, we show the hierarchy between VC-2-FWL+ and VC-2-WL.

**Proposition C.6.** *The* VC-*2*-FWL+ *algorithm strictly refines* VC-*2*-WL*, denoted as*

$$\text{VC-2-FWL+} \sqsubset \text{VC-2-WL}.$$

*Proof.* The update of constraint nodes is exactly the same for VC-2-WL and VC-2-FWL+. Now, we consider the variable nodes.

We first prove VC-2-FWL+ $\sqsubseteq$ VC-2-WL. First, for $t = 0$, the colors $v_{ij}^0$ is basically the encoding of the problem $C_{ij}$, therefore if for VC-2-FWL+: $v_{ij}^0 = v_{pq}^0$, it also holds for VC-2-WL. For any $t > 0$, we want to show that if for VC-2-FWL+:

$\boldsymbol{v}_{ij}^t = \boldsymbol{v}_{pq}^t$ holds, it also holds for VC-2-WL update. Specifically, for VC-2-FWL+:

$$\boldsymbol{v}_{ij}^t = \boldsymbol{v}_{pq}^t$$
$$\Longrightarrow \mathsf{hash}\Big(\boldsymbol{v}_{ij}^{t-1}, \{\!\{(\boldsymbol{v}_{uj}^{t-1}, \boldsymbol{v}_{iu}^{t-1}) \mid u \in [n]\}\!\}, \{\!\{(A_{k,ij}, \boldsymbol{c}_k^{t-1}) \mid k \in N(ij)\}\!\}\Big)$$
$$= \mathsf{hash}\Big(\boldsymbol{v}_{pq}^{t-1}, \{\!\{(\boldsymbol{v}_{uq}^{t-1}, \boldsymbol{v}_{pu}^{t-1}) \mid u \in [n]\}\!\}, \{\!\{(A_{k,pq}, \boldsymbol{c}_k^{t-1}) \mid k \in N(pq)\}\!\}\Big)$$
$$\Longrightarrow \boldsymbol{v}_{ij}^{t-1} = \boldsymbol{v}_{pq}^{t-1} \wedge$$
$$\{\!\{(\boldsymbol{v}_{uj}^{t-1}, \boldsymbol{v}_{iu}^{t-1}) \mid u \in [n]\}\!\} = \{\!\{(\boldsymbol{v}_{uq}^{t-1}, \boldsymbol{v}_{pu}^{t-1}) \mid u \in [n]\}\!\} \wedge$$
$$\{\!\{(A_{k,ij}, \boldsymbol{c}_k^{t-1}) \mid k \in N(ij)\}\!\} = \{\!\{(A_{k,pq}, \boldsymbol{c}_k^{t-1}) \mid k \in N(ij)\}\!\}$$
$$\Longrightarrow \boldsymbol{v}_{ij}^{t-1} = \boldsymbol{v}_{pq}^{t-1} \wedge$$
$$\{\!\{\boldsymbol{v}_{uj}^{t-1} \mid u \in [n]\}\!\} = \{\!\{\boldsymbol{v}_{uq}^{t-1} \mid u \in [n]\}\!\} \wedge$$
$$\{\!\{\boldsymbol{v}_{iu}^{t-1} \mid u \in [n]\}\!\} = \{\!\{\boldsymbol{v}_{pu}^{t-1} \mid u \in [n]\}\!\} \wedge$$
$$\{\!\{(A_{k,ij}, \boldsymbol{c}_k^{t-1}) \mid k \in N(ij)\}\!\} = \{\!\{(A_{k,pq}, \boldsymbol{c}_k^{t-1}) \mid k \in N(ij)\}\!\}$$
$$\Longrightarrow \mathsf{hash}\Big(\boldsymbol{v}_{ij}^{t-1},$$
$$\{\!\{\boldsymbol{v}_{uj}^{t-1} \mid u \in [n]\}\!\}, \{\!\{\boldsymbol{v}_{iu}^{t-1} \mid u \in [n]\}\!\}$$
$$\{\!\{(A_{k,ij}, \boldsymbol{c}_k^{t-1}) \mid k \in N(ij)\}\!\}\Big)$$
$$= \mathsf{hash}\Big(\boldsymbol{v}_{pq}^{t-1}$$
$$\{\!\{\boldsymbol{v}_{uq}^{t-1} \mid u \in [n]\}\!\}, \{\!\{\boldsymbol{v}_{pu}^{t-1} \mid u \in [n]\}\!\}$$
$$\{\!\{(A_{k,pq}, \boldsymbol{c}_k^{t-1}) \mid k \in N(ij)\}\!\}\Big),$$

thus with VC-2-WL update they also have the same color. The implication holds because the tuples are ordered, which ensures that their marginal entries form multisets that are respectively identical.

We then prove strict refinement by giving an example in the proof of Proposition 2.2 where VC-2-WL fails. In this example, the converged VC-2-FWL+ assigns different colors to every primal variable, except symmetric ones, effectively distinguishing the variables in the solution.

Therefore, VC-2-FWL+ $\sqsubset$ VC-2-WL. $\qquad\qquad\square$

**Proposition C.7.** *The* VC-2-FWL+ *algorithm strictly refines* VC-2-FWL*, denoted as*

$$\text{VC-}2\text{-FWL+} \sqsubset \text{VC-}2\text{-FWL}.$$

*Proof.* We first prove VC-2-FWL+ $\sqsubseteq$ VC-2-FWL.

First, for $t = 0$, the colors initializations are the same, therefore if for VC-2-FWL+: $\boldsymbol{v}_{ij}^0 = \boldsymbol{v}_{pq}^0$, it also holds for VC-2-FWL. For some $t > 0$, we want to show that if for VC-2-FWL+: $\boldsymbol{v}_{ij}^t = \boldsymbol{v}_{pq}^t$ holds, it also holds for VC-2-FWL update. Specifically, for VC-2-FWL+:

$$\boldsymbol{v}_{ij}^t = \boldsymbol{v}_{pq}^t \Longrightarrow$$
$$\mathsf{hash}\Big(\boldsymbol{v}_{ij}^{t-1}, \{\!\{(\boldsymbol{v}_{uj}^{t-1}, \boldsymbol{v}_{iu}^{t-1}) \mid u \in [n]\}\!\}, \{\!\{(A_{k,ij}, \boldsymbol{c}_k^{t-1}) \mid k \in N(ij)\}\!\}\Big)$$
$$= \mathsf{hash}\Big(\boldsymbol{v}_{pq}^{t-1}, \{\!\{(\boldsymbol{v}_{uq}^{t-1}, \boldsymbol{v}_{pu}^{t-1}) \mid u \in [n]\}\!\}, \{\!\{(A_{k,pq}, \boldsymbol{c}_k^{t-1}) \mid k \in N(pq)\}\!\}\Big)$$
$$\Longrightarrow \mathsf{hash}\Big(\boldsymbol{v}_{ij}^{t-1}, \{\!\{\{\!\{\boldsymbol{v}_{uj}^{t-1}, \boldsymbol{v}_{iu}^{t-1}\}\!\} \mid u \in [n]\}\!\}, \{\!\{(A_{k,ij}, \boldsymbol{c}_k^{t-1}) \mid k \in N(ij)\}\!\}\Big)$$
$$= \mathsf{hash}\Big(\boldsymbol{v}_{pq}^{t-1}, \{\!\{\{\!\{\boldsymbol{v}_{uq}^{t-1}, \boldsymbol{v}_{pu}^{t-1}\}\!\} \mid u \in [n]\}\!\}, \{\!\{(A_{k,pq}, \boldsymbol{c}_k^{t-1}) \mid k \in N(pq)\}\!\}\Big).$$

The last implication is straightforward because a 2-multiset is an unordered 2-tuple. The last equation represents the VC-2-FWL update, which cannot distinguish them either. This holds for all $t$, therefore by induction, VC-2-FWL+ $\sqsubseteq$ VC-2-FWL.

Next, we provide the strict refinement by providing an example where VC-2-FWL+ distinguishes a pair, where VC-2-FWL does not. Consider the example

$$C = \begin{bmatrix} 0 & 1 & 2 & 3 \\ 1 & 0 & 4 & 2 \\ 2 & 4 & 0 & 1 \\ 3 & 2 & 1 & 0 \end{bmatrix}, \quad A_1 = \mathbf{1}_{4\times 4}, \quad b = [1].$$

We simply let $v_{ij}^0 = C_{ij}$. Consider the update for the indices $(1, 2)$ and $(3, 4)$. Under VC-2-FWL:

$$v_{12}^1 := \mathsf{hash}\left(v_{12}^0, \{\!\{ \{\!\{v_{12}^0, v_{11}^0\}\!\}, \{\!\{v_{22}^0, v_{12}^0\}\!\}, \{\!\{v_{32}^0, v_{13}^0\}\!\}, \{\!\{v_{42}^0, v_{14}^0\}\!\} \}\!\}, (1, c_1^0)\right)$$
$$= \mathsf{hash}\left(1, \{\!\{ \{\!\{1, 0\}\!\}, \{\!\{0, 1\}\!\}, \{\!\{4, 2\}\!\}, \{\!\{2, 3\}\!\} \}\!\}, (1, c_1^0)\right),$$

and

$$v_{34}^1 := \mathsf{hash}\left(v_{34}^0, \{\!\{ \{\!\{v_{14}^0, v_{31}^0\}\!\}, \{\!\{v_{24}^0, v_{32}^0\}\!\}, \{\!\{v_{34}^0, v_{33}^0\}\!\}, \{\!\{v_{44}^0, v_{34}^0\}\!\} \}\!\}, (1, c_1^0)\right)$$
$$= \mathsf{hash}\left(1, \{\!\{ \{\!\{3, 2\}\!\}, \{\!\{2, 4\}\!\}, \{\!\{1, 0\}\!\}, \{\!\{0, 1\}\!\} \}\!\}, (1, c_1^0)\right).$$

If we calculate the bins of 2-multisets, we notice they both have: two $\{\!\{0, 1\}\!\}$, one $\{\!\{2, 3\}\!\}$, one $\{\!\{2, 4\}\!\}$. Therefore, $v_{12}^1$ and $v_{34}^1$ take on the same color under VC-2-FWL. VC-2-FWL+ differs in that those 2-multisets become 2-tuples. Therefore, for VC-2-FWL+, $v_{12}^1$ update contains a tuple $(2, 3)$ and a $(4, 2)$, and $v_{34}^1$ contains a $(3, 2)$ and a $(2, 4)$, thus $v_{12}^1 \neq v_{34}^1$ under VC-2-FWL+. That means VC-2-FWL+ distinguishes $v_{12}$ and $v_{34}$ at the 1st iteration while VC-2-FWL not.

Therefore, VC-2-FWL+ $\sqsubset$ VC-2-FWL. $\qquad\square$

Finally, we show that VC-2-WL and VC-2-FWL are incomparable.

**Proposition C.8.** VC-*2*-WL *and* VC-*2*-FWL *are incomparable in expressive power, denoted as*

$$\text{VC-2-WL} \not\equiv \text{VC-2-FWL}.$$

*Proof.* We first show that there exists an example where VC-2-WL can distinguish a pair of variables and VC-2-FWL cannot. Consider the following example:

$$C = \begin{bmatrix} 1 & 0 & 4 & 2 \\ 0 & 1 & 2 & 3 \\ 4 & 2 & 1 & 0 \\ 2 & 3 & 0 & 1 \end{bmatrix}, \quad A_1 = \mathbf{1}_{4\times 4}, \quad b = [1].$$

We simply let $v_{ij}^0 = C_{ij}$. Consider the update for the indices $(1, 4)$ and $(2, 3)$. Under VC-2-FWL:

$$v_{14}^1 := \mathsf{hash}\left(v_{14}^0, \{\!\{ \{\!\{v_{14}^0, v_{11}^0\}\!\}, \{\!\{v_{24}^0, v_{12}^0\}\!\}, \{\!\{v_{34}^0, v_{13}^0\}\!\}, \{\!\{v_{44}^0, v_{14}^0\}\!\} \}\!\}, (1, c_1^0)\right)$$
$$= \mathsf{hash}\left(2, \{\!\{ \{\!\{2, 1\}\!\}, \{\!\{3, 0\}\!\}, \{\!\{0, 4\}\!\}, \{\!\{1, 2\}\!\} \}\!\}, (1, c_1^0)\right),$$

and

$$v_{23}^1 := \mathsf{hash}\left(v_{23}^0, \{\!\{ \{\!\{v_{13}^0, v_{21}^0\}\!\}, \{\!\{v_{23}^0, v_{22}^0\}\!\}, \{\!\{v_{33}^0, v_{23}^0\}\!\}, \{\!\{v_{43}^0, v_{24}^0\}\!\} \}\!\}, (1, c_1^0)\right)$$
$$= \mathsf{hash}\left(2, \{\!\{ \{\!\{4, 0\}\!\}, \{\!\{2, 1\}\!\}, \{\!\{1, 2\}\!\}, \{\!\{0, 3\}\!\} \}\!\}, (1, c_1^0)\right).$$

Therefore $v_{14}^1 = v_{23}^1$ under VC-2-FWL. While for VC-2-WL update:

$$v_{14}^1 := \mathsf{hash}\left(v_{14}^0, \{\!\{v_{14}^0, v_{24}^0, v_{34}^0, v_{44}^0\}\!\}, \{\!\{v_{11}^0, v_{12}^0, v_{13}^0, v_{14}^0\}\!\}, (1, c_1^0)\right)$$
$$= \mathsf{hash}\left(2, \{\!\{2, 3, 0, 1\}\!\}, \{\!\{1, 0, 4, 2\}\!\}, (1, c_1^0)\right),$$

and

$$\boldsymbol{v}_{23}^1 \coloneqq \mathsf{hash}\Big(\boldsymbol{v}_{23}^0, \{\!\{\boldsymbol{v}_{13}^0, \boldsymbol{v}_{23}^0, \boldsymbol{v}_{33}^0, \boldsymbol{v}_{43}^0\}\!\}, \{\!\{\boldsymbol{v}_{21}^0, \boldsymbol{v}_{22}^0, \boldsymbol{v}_{23}^0, \boldsymbol{v}_{24}^0\}\!\}, (1, \boldsymbol{c}_1^0)\Big)$$
$$= \mathsf{hash}\Big(2, \{\!\{4, 2, 1, 0\}\!\}, \{\!\{0, 1, 2, 3\}\!\}, (1, \boldsymbol{c}_1^0)\Big)$$

thus $\boldsymbol{v}_{14}^1 \neq \boldsymbol{v}_{23}^1$ under VC-2-WL.

Next, we show there exists an example where VC-2-FWL can distinguish a pair of variables and VC-2-WL cannot. We reuse the example in the proof of Proposition 2.2. VC-2-FWL effectively distinguishes all the variables except symmetric ones, and is able to distinguish variables $(1, 5)$ and $(2, 4)$ where VC-2-WL fails.

Therefore, VC-2-WL and VC-2-FWL are incomparable in expressive power. □

*Remark* C.9. The incomparability mainly stems from the directness of VC-2-WL. If we define a variant of VC-2-WL, supposedly VC-2-WL−, where $\{\!\{\boldsymbol{v}_{uj}^{t-1} \mid u \in [n]\}\!\}$ and $\{\!\{\boldsymbol{v}_{iu}^{t-1} \mid u \in [n]\}\!\}$ are aggregated in an unordered way then used to update $\boldsymbol{v}_{ij}$, it will not distinguish the variables $\boldsymbol{v}_{14}$ and $\boldsymbol{v}_{23}$ in the case above. However, it will be less expressive than VC-2-WL, and also fails on the example of Proposition 2.2. Therefore, it is not valuable to investigate such a VC-2-WL variant.

*Remark* C.10. Although VC-2-WL and VC-2-FWL are theoretically incomparable, with VC-2-WL possessing superior distinguishing power in specific cases, the optimization dynamics of SDPs are still inherently subsumed by VC-2-FWL rather than VC-2-WL. We will later demonstrate that the algorithmic trajectory of an iterative solver, including both intermediate steps and the final converged solution, is strictly refined by the VC-2-FWL process.

### C.3. Convergence of PDHG algorithm on SDP with least-norm solution

As a prerequisite to establishing the expressivity of VC-2-FWL, we first rigorously demonstrate the convergence of the PDHG algorithm applied to the SDP problem with Frobenius norm regularization. First, we briefly introduce some conventional notations based on Eckstein & Bertsekas (1992); O'Connor & Vandenberghe (2020); Wang et al. (2024).

Let $F$ be a set-valued operator which maps every $\boldsymbol{x} \in \mathbb{R}^n$ to a subset of $\mathbb{R}^n$. The notation is useful for subgradient methods, as the subgradient of a non-smooth function is usually a set. The *graph* gr of an operator $F$ is the set

$$\mathsf{gr}(F) \coloneqq \{(\boldsymbol{x}, \boldsymbol{y}) \in \mathbb{R}^n \times \mathbb{R}^n \mid \boldsymbol{y} \in F(\boldsymbol{x})\}.$$

An operator $F$ is *monotone* if

$$(\boldsymbol{y} - \hat{\boldsymbol{y}})^\mathsf{T}(\boldsymbol{x} - \hat{\boldsymbol{x}}) \geq 0, \quad \text{for all } (\boldsymbol{x}, \boldsymbol{y}), (\hat{\boldsymbol{x}}, \hat{\boldsymbol{y}}) \in \mathsf{gr}(F).$$

A monotone operator is *maximal monotone* if its graph is not contained in the graph of another monotone operator. An operator $F$ is $\mu$-strongly monotone if and only if

$$(\boldsymbol{y} - \hat{\boldsymbol{y}})^\mathsf{T}(\boldsymbol{x} - \hat{\boldsymbol{x}}) \geq \mu(\boldsymbol{x} - \hat{\boldsymbol{x}})^\mathsf{T}(\boldsymbol{x} - \hat{\boldsymbol{x}}), \quad \text{for all } (\boldsymbol{x}, \boldsymbol{y}), (\hat{\boldsymbol{x}}, \hat{\boldsymbol{y}}) \in \mathsf{gr}(F).$$

Given a positive scalar $c$ and an operator $F$, the *resolvent* of an operator $F$ is defined as $J_{cF} \coloneqq (I + cF)^{-1}$ where $I$ is identity mapping. $J_{cF}(\boldsymbol{x})$ is the set of all the values $\boldsymbol{y}$ satisfying

$$\boldsymbol{x} - \boldsymbol{y} \in cF(\boldsymbol{y}). \tag{7}$$

The resolvent of the subdifferential of a *closed, convex and proper* function $f$ is called the *proximal operator* of $f$, denoted $\mathrm{prox}_{cf} = J_{c\partial f}$. Equation (7) is the optimality condition of the optimization problem

$$\mathrm{prox}_{cf}(\boldsymbol{x}) \coloneqq \arg\min_{\boldsymbol{y}} \left( cf(\boldsymbol{y}) + \frac{1}{2}\|\boldsymbol{y} - \boldsymbol{x}\|_2^2 \right).$$

In plain language, the proximal operator finds an optimal solution that minimizes the function $f$ without searching too far

from the input $\boldsymbol{x}$. The equivalence can be verified,

$$
\begin{aligned}
\mathbf{0} &\in \partial \left( cf(\boldsymbol{y}) + \frac{1}{2}\|\boldsymbol{y} - \boldsymbol{x}\|_2^2 \right) \\
\implies \mathbf{0} &\in c\partial f(\boldsymbol{y}) + (\boldsymbol{y} - \boldsymbol{x}) \\
\implies \boldsymbol{x} &\in c\partial f(\boldsymbol{y}) + \boldsymbol{y} \\
\implies \boldsymbol{x} &\in (I + c\partial f)(\boldsymbol{y}) \\
\implies \boldsymbol{y} &= (I + c\partial f)^{-1}(\boldsymbol{x}) = J_{c\partial f}(\boldsymbol{x}).
\end{aligned}
\tag{8}
$$

Now let's state our problem setting. First, it is obvious that finding the primal solution $\boldsymbol{X}^*$ to

$$
\begin{aligned}
\min_{\boldsymbol{X} \in \mathbb{S}_+^n} \quad & \langle \boldsymbol{C}, \boldsymbol{X} \rangle \\
\text{s.t.} \quad & \mathcal{A}(\boldsymbol{X}) = \boldsymbol{b}
\end{aligned}
$$

with the minimum Frobenius norm is equivalent to finding the primal solution to

$$
\begin{aligned}
\min_{\boldsymbol{X} \in \mathbb{S}_+^n} \quad & \langle \boldsymbol{C}, \boldsymbol{X} \rangle + \frac{\varepsilon}{2} \langle \boldsymbol{X}, \boldsymbol{X} \rangle \\
\text{s.t.} \quad & \mathcal{A}(\boldsymbol{X}) = \boldsymbol{b}
\end{aligned}
\tag{9}
$$

with arbitrarily small $\varepsilon \to 0$. The SDP with a Frobenius regularization term has a unique primal solution, as the objective is strictly convex and the feasible set is a convex set (Wright, 2022, p. 21), which aligns with our unique solution assumption Proposition 1.1.

Then, we introduce the PDHG method Chambolle & Pock (2016) on this problem. We extend the PDHG for linear SDP in Wang et al. (2024) by inserting the quadratic term $\frac{\varepsilon}{2} \langle \boldsymbol{X}, \boldsymbol{X} \rangle$. Consequently, instead of the PDHG update form in Wang et al. (2024, Algorithm 1), we have the following form

$$
\begin{aligned}
\boldsymbol{X}^{t+1} &:= \mathrm{Proj}_{\mathbb{S}_+^n} \left[ \frac{\boldsymbol{X}^t - \alpha_t \mathcal{A}^*(\boldsymbol{y}^t) - \alpha_t \boldsymbol{C}}{1 + \alpha_t \varepsilon} \right] \\
\boldsymbol{y}^{t+1} &:= \boldsymbol{y}^t + \beta_t \mathcal{A}\left( \boldsymbol{X}^{t+1} + \theta_t \left( \boldsymbol{X}^{t+1} - \boldsymbol{X}^t \right) \right) - \beta_t \boldsymbol{b},
\end{aligned}
\tag{10}
$$

where $\boldsymbol{X}^t, \boldsymbol{y}^t$ are primal-dual solutions at different iterations, and $\alpha_t, \beta_t, \theta_t$ are suitable step lengths.

The $\mathrm{Proj}_{\mathbb{S}_+^n}$ projects the matrix into the PSD cone $\mathbb{S}_+^n$, specifically, by solving

$$
\boldsymbol{X}^{t+1} := \arg\min_{\boldsymbol{X} \in \mathbb{S}_+^n} \left\| \boldsymbol{X} - \frac{\boldsymbol{X}^t - \alpha_t \mathcal{A}^*(\boldsymbol{y}^t) - \alpha_t \boldsymbol{C}}{1 + \alpha_t \varepsilon} \right\|_F^2.
$$

It can be done by spectral decomposition (Boyd & Vandenberghe, 2004, p. 399) (Rontsis et al., 2022), see Algorithm 1 for detailed process.

---

**Algorithm 1** Projection onto the PSD Cone

---

**Require:** A symmetric matrix $\boldsymbol{X} \in \mathbb{S}^n$
**Ensure:** The projected matrix $\boldsymbol{X}_{\mathbb{S}} := \arg\min_{\boldsymbol{X}' \in \mathbb{S}_+^n} \|\boldsymbol{X}' - \boldsymbol{X}\|_F$
 1: Compute the spectral decomposition $\boldsymbol{X} = \boldsymbol{V}\boldsymbol{\Lambda}\boldsymbol{V}^{\mathsf{T}}$
 2: $\{\boldsymbol{\Lambda} := \mathrm{diag}(\lambda_1, \ldots, \lambda_n)$ is the diagonal matrix of eigenvalues.$\}$
 3: $\{\boldsymbol{V}$ is the orthonormal matrix of corresponding eigenvectors $\boldsymbol{v}_i$ as its columns.$\}$
 4: Construct the projected matrix by dropping the negative eigen-pairs:
 5: $\boldsymbol{X}_{\mathbb{S}} \leftarrow \sum_{i=1}^n \max(\lambda_i, 0) \cdot \boldsymbol{v}_i \boldsymbol{v}_i^{\mathsf{T}}$
 6: **return** $\boldsymbol{X}_{\mathbb{S}}$

---

As the constraints of the SDP problem are equality constraints, there is no projection on $\boldsymbol{y}^t \in \mathbb{R}^m$ needed.

Wang et al. (2024, Theorem 2.1) showed that the PDHG algorithm on linear SDP (without our Frobenius term) weakly converges (Bauschke & Combettes, 2020, p. 35) to an optimal point. We hereby prove a similar result for Equation (10).

**Proposition C.11.** *If the adjustment of $\left\{ \left( \alpha_t, \theta_t, \beta_t \right) \right\}_t$ in Equation* (10) *follow*

$$\alpha_t \in (\alpha_{\min}, \alpha_{\max}), \quad \sum_{t=1}^{\infty} |\alpha_{t+1} - \alpha_t| < \infty, \quad \theta_t = \frac{\alpha_t}{\alpha_{t-1}}, \quad \alpha_t \beta_t = R,$$

*where $0 < \alpha_{\min} \le \alpha_{\max} < \infty$ and $R < \frac{1}{\lambda_{\max}(\mathcal{A}^*\mathcal{A})}$. Then PDHG algorithm in Equation* (10) *weakly converges to $(\boldsymbol{X}^*, \boldsymbol{y}^*)$ such that $0 \in \boldsymbol{C} + \varepsilon \boldsymbol{X}^* + \partial_{\boldsymbol{X}} \mathbb{I}_{\mathbb{S}_+^n}(\boldsymbol{X}^*) + \partial_{\boldsymbol{X}} \mathbb{I}_{=\boldsymbol{b}}(\mathcal{A}(\boldsymbol{X}^*))$, for any $\varepsilon > 0$.*

*Proof.* Our proof establishes convergence by framing the algorithm as an instance of non-stationary Douglas-Rachford Splitting (DRS), following the methodology of Wang et al. (2024). We adapt their derivation to account for the Frobenius regularization term, which alters the primal variable update.

We rewrite Equation (9) into the following non-smooth optimization problem:

$$\min_{\boldsymbol{X}} \quad \langle \boldsymbol{C}, \, \boldsymbol{X} \rangle + \frac{\varepsilon}{2} \langle \boldsymbol{X}, \, \boldsymbol{X} \rangle + \mathbb{I}_{\mathbb{S}_+^n}(\boldsymbol{X}) + \mathbb{I}_{=\boldsymbol{b}}(\mathcal{A}(\boldsymbol{X})), \tag{11}$$

with indicator function $\mathbb{I}_{\mathbb{S}_+^n}(\boldsymbol{X}) = 0$ if $\boldsymbol{X} \in \mathbb{S}_+^n$ otherwise $+\infty$, and $\mathbb{I}_{=\boldsymbol{b}}(\boldsymbol{y}) = 0$ if $\boldsymbol{y} = \boldsymbol{b}$ and $+\infty$ otherwise.

To apply DRS, we introduce an auxiliary variable $\hat{\boldsymbol{X}}$ (constrained to be zero) and a linear operator $\mathcal{T} : \mathbb{S}^n \to \mathbb{R}^m$. The regularized problem in Equation (11) is reformulated as

$$\min_{\boldsymbol{X}, \hat{\boldsymbol{X}}} \quad \langle \boldsymbol{C}, \, \boldsymbol{X} \rangle + \frac{\varepsilon}{2} \langle \boldsymbol{X}, \, \boldsymbol{X} \rangle + \mathbb{I}_{\mathbb{S}_+^n}(\boldsymbol{X}) + \mathbb{I}_{=0}(\hat{\boldsymbol{X}}) + \mathbb{I}_{=\boldsymbol{b}}\left( \mathcal{A}(\boldsymbol{X}) + \mathcal{T}(\hat{\boldsymbol{X}}) \right).$$

As the problem is convex, it is equivalent to finding the first-order optimality condition:

$$\text{Find } \begin{pmatrix} \boldsymbol{X} \\ \hat{\boldsymbol{X}} \end{pmatrix} \quad \text{s.t.} \quad 0 \in \underbrace{\begin{pmatrix} \boldsymbol{C} + \varepsilon \boldsymbol{X} \\ 0 \end{pmatrix} + \begin{pmatrix} \partial_{\boldsymbol{X}} \mathbb{I}_{\mathbb{S}_+^n}(\boldsymbol{X}) \\ 0 \end{pmatrix} + \begin{pmatrix} 0 \\ \partial_{\hat{\boldsymbol{X}}} \mathbb{I}_{=0}(\hat{\boldsymbol{X}}) \end{pmatrix}}_{B = \partial f(\boldsymbol{X}, \hat{\boldsymbol{X}})} + \underbrace{\begin{pmatrix} \partial_{\boldsymbol{X}} \mathbb{I}_{=\boldsymbol{b}}(\mathcal{A}(\boldsymbol{X}) + \mathcal{T}(\hat{\boldsymbol{X}})) \\ \partial_{\hat{\boldsymbol{X}}} \mathbb{I}_{=\boldsymbol{b}}(\mathcal{A}(\boldsymbol{X}) + \mathcal{T}(\hat{\boldsymbol{X}})) \end{pmatrix}}_{A = \partial g(\boldsymbol{X}, \hat{\boldsymbol{X}})}, \tag{12}$$

Operators $A$ and $B$ are maximally monotone. Notably, due to the regularization term $\frac{\varepsilon}{2} \langle \boldsymbol{X}, \, \boldsymbol{X} \rangle$, the operator $B$ in our case is $\varepsilon$-strongly monotone with respect to $\boldsymbol{X}$. Following Wang et al. (2024) and applying non-stationary DRS to Equation (12) yields the following fixed-point iterations:

$$\begin{pmatrix} \boldsymbol{X}^t \\ \hat{\boldsymbol{X}}^t \end{pmatrix} = J_{\alpha_{t-1}\partial f}\left( \begin{pmatrix} \boldsymbol{Z}^t \\ \hat{\boldsymbol{Z}}^t \end{pmatrix} \right) \tag{13}$$

$$\begin{pmatrix} \boldsymbol{X}^{t+\frac{1}{2}} \\ \hat{\boldsymbol{X}}^{t+\frac{1}{2}} \end{pmatrix} = J_{\alpha_t \partial g}\left( \begin{pmatrix} \boldsymbol{X}^t \\ \hat{\boldsymbol{X}}^t \end{pmatrix} + \theta_t\left( \begin{pmatrix} \boldsymbol{X}^t \\ \hat{\boldsymbol{X}}^t \end{pmatrix} - \begin{pmatrix} \boldsymbol{Z}^t \\ \hat{\boldsymbol{Z}}^t \end{pmatrix} \right) \right) \tag{14}$$

$$\begin{pmatrix} \boldsymbol{Z}^{t+1} \\ \hat{\boldsymbol{Z}}^{t+1} \end{pmatrix} = \begin{pmatrix} \boldsymbol{X}^{t+\frac{1}{2}} \\ \hat{\boldsymbol{X}}^{t+\frac{1}{2}} \end{pmatrix} + \theta_t\left( \begin{pmatrix} \boldsymbol{Z}^t \\ \hat{\boldsymbol{Z}}^t \end{pmatrix} - \begin{pmatrix} \boldsymbol{X}^t \\ \hat{\boldsymbol{X}}^t \end{pmatrix} \right). \tag{15}$$

While the overall structure matches the standard DRS form in Wang et al. (2024), the specific update in Equation (13) differs. In our case, Equation (13), by definition of the resolvent, can be simplified to

$$\boldsymbol{X}^t = \arg\min_{\boldsymbol{X}} \left\{ \langle \boldsymbol{C}, \, \boldsymbol{X} \rangle + \frac{\varepsilon}{2} \langle \boldsymbol{X}, \, \boldsymbol{X} \rangle + \mathbb{I}_{\mathbb{S}_+^n}(\boldsymbol{X}) + \frac{1}{2\alpha_{t-1}} \left\| \boldsymbol{X} - \boldsymbol{Z}^t \right\|_F^2 \right\} \tag{16}$$

$$\hat{\boldsymbol{X}}^t = 0$$

Equation (14) follows Wang et al. (2024), and we obtain:

$$\boldsymbol{y}^t = \arg\min_{\boldsymbol{y}} \left\{ \mathbb{I}_{=\boldsymbol{b}}(\boldsymbol{y}) - \left\langle \mathcal{A}\left( \boldsymbol{X}^t + \theta_t(\boldsymbol{X}^t - \boldsymbol{Z}^t) \right) - \theta_t \mathcal{T}(\hat{\boldsymbol{Z}}^t), \, \boldsymbol{y} \right\rangle \right.$$

$$\left. + \frac{\alpha_t}{2}\left( \|\mathcal{A}^*(\boldsymbol{y})\|_F^2 + \|\mathcal{T}^*(\boldsymbol{y})\|_F^2 \right) \right\} \tag{17}$$

$$\boldsymbol{X}^{t+\frac{1}{2}} = \boldsymbol{X}^t + \theta_t(\boldsymbol{X}^t - \boldsymbol{Z}^t) - \alpha_t \mathcal{A}^*(\boldsymbol{y}^t) \tag{18}$$

$$\hat{\boldsymbol{X}}^{t+\frac{1}{2}} = -\theta_t \hat{\boldsymbol{Z}}^t - \alpha_t \mathcal{T}^*(\boldsymbol{y}^t), \tag{19}$$

Subsequently, Equation (15) can be simplified to

$$\boldsymbol{Z}^{t+1} = \boldsymbol{X}^t - \alpha_t \mathcal{A}^*(\boldsymbol{y}^t) \tag{20}$$

$$\hat{\boldsymbol{Z}}^{t+1} = -\alpha_t \mathcal{T}^*(\boldsymbol{y}^t). \tag{21}$$

Substituting Equation (20) into Equation (16), we get

$$
\begin{aligned}
\boldsymbol{X}^{t+1} &:= \arg\min_{\boldsymbol{X}} \left\{ \langle \boldsymbol{C}, \boldsymbol{X} \rangle + \frac{\varepsilon}{2} \langle \boldsymbol{X}, \boldsymbol{X} \rangle + \mathbb{I}_{\mathbb{S}^n_+}(\boldsymbol{X}) + \frac{1}{2\alpha_t} \left\| \boldsymbol{X} - \left( \boldsymbol{X}^t - \alpha_t \mathcal{A}^*(\boldsymbol{y}^t) \right) \right\|_F^2 \right\} \\
&= \arg\min_{\boldsymbol{X}} \left\{ \mathbb{I}_{\mathbb{S}^n_+}(\boldsymbol{X}) + \left( \frac{\varepsilon}{2} + \frac{1}{2\alpha_t} \right) \langle \boldsymbol{X}, \boldsymbol{X} \rangle + \left\langle \boldsymbol{C} - \frac{1}{\alpha_t} \left( \boldsymbol{X}^t - \alpha_t \mathcal{A}^*(\boldsymbol{y}^t) \right), \boldsymbol{X} \right\rangle \right\} \\
&= \arg\min_{\boldsymbol{X}} \left\{ \mathbb{I}_{\mathbb{S}^n_+}(\boldsymbol{X}) + \left\| \boldsymbol{X} - \frac{\boldsymbol{X}^t - \alpha_t \mathcal{A}^*(\boldsymbol{y}^t) - \alpha_t \boldsymbol{C}}{1 + \alpha_t \varepsilon} \right\|_F^2 \right\} \\
&= \mathrm{Prox}_{\frac{1}{2}\mathbb{I}_{\mathbb{S}^n_+}(\cdot)} \left[ \frac{\boldsymbol{X}^t - \alpha_t \mathcal{A}^*(\boldsymbol{y}^t) - \alpha_t \boldsymbol{C}}{1 + \alpha_t \varepsilon} \right] \\
&= \mathrm{Proj}_{\mathbb{S}^n_+} \left[ \frac{\boldsymbol{X}^t - \alpha_t \mathcal{A}^*(\boldsymbol{y}^t) - \alpha_t \boldsymbol{C}}{1 + \alpha_t \varepsilon} \right]
\end{aligned}
\tag{22}
$$

The derivation from $\arg\min$ to Proj holds, as the the term

$$\left\| \boldsymbol{X} - \frac{\boldsymbol{X}^t - \alpha_t \mathcal{A}^*(\boldsymbol{y}^t) - \alpha_t \boldsymbol{C}}{1 + \alpha_t \varepsilon} \right\|_F^2 \tag{23}$$

is isotropic w.r.t. each entry of $\boldsymbol{X}$ and centered at

$$\frac{\boldsymbol{X}^t - \alpha_t \mathcal{A}^*(\boldsymbol{y}^t) - \alpha_t \boldsymbol{C}}{1 + \alpha_t \varepsilon}. \tag{24}$$

Therefore, finding the minimizer to Equation (22) is equivalent to finding the minimizer to Equation (23), that is Equation (24), then do Euclidean projection to the space of PSD cone, which is exactly the definition of PSD projection.

We notice that when $\epsilon \to 0$, the problem reduces to linear SDP and the formulation is exactly the same as in Wang et al. (2024).

The dual update follows the derivation in Wang et al. (2024), the resolvent step Equation (17) simplifies to:

$$\boldsymbol{y}^{t+1} := \boldsymbol{y}^t + \beta_t \mathcal{A} \left( \boldsymbol{X}^{t+1} + \theta_t \left( \boldsymbol{X}^{t+1} - \boldsymbol{X}^t \right) \right) - \beta_t \boldsymbol{b}, \tag{25}$$

Now we arrive at the same conclusion as Wang et al. (2024). Since our PDHG update Equation (10) is an instance of non-stationary DRS, convergence to the unique solution $(\boldsymbol{X}^*, \boldsymbol{y}^*)$ follows immediately from Lorenz & Tran-Dinh (2019). □

*Remark* C.12 (Notation and indexing). We clarify the indexing convention used in our derivation compared to Wang et al. (2024). While their Algorithm 1 performs the primal update $\boldsymbol{X}^t$ followed by the dual update $\boldsymbol{y}^t$, their theoretical proof presents the dual update first. Although these schemes are functionally equivalent due to the cyclic nature of alternating updates, we prioritize consistency between the derivation and the implementation. In the proof of Wang et al. (2024), a variable substitution to $\boldsymbol{y}^{t+1}$ is employed in their Equation 8 during the derivation of the dual step. We observe that by retaining the index $\boldsymbol{y}^t$ (i.e., avoiding this forward shift), the resulting fixed-point iteration aligns perfectly with the order of operations in their Algorithm 1. Consequently, our derivation adopts this convention in Equation (17), resulting in the PDHG algorithm in Equation (10) which updates the primal variable $\boldsymbol{X}^t$ first, thereby unifying the notation of the theory and the algorithm.

*Remark* C.13 (Strong convergence in finite dimensions). Standard convergence analyses for operator splitting methods, such as those in Lorenz & Tran-Dinh (2019), typically guarantee only weak convergence of the iterates in general Hilbert spaces. However, our problem setting is the space of symmetric matrices $\mathbb{S}^n$, which is a finite-dimensional space. It is a fundamental result that in finite-dimensional spaces, the weak topology coincides with the strong topology. Consequently, the weak convergence established in Proposition Appendix C.3 implies strong convergence in the Frobenius norm. That is, the sequence satisfies $\lim_{t\to\infty} \|\boldsymbol{X}^t - \boldsymbol{X}^*\|_F = 0$.

## C.4. Proof for VC-2-FWL expressivity

**Theorem C.14** (Restatement of Theorem 2.3). *Let $X^* \in \mathbb{S}_+^n$ be the primal optimal solution to a given SDP instance and given indices $(i, j), (p, q)$. If the stable colorings of* VC-2-FWL *satisfy $v_{ij}^\infty = v_{pq}^\infty$, then the solution values satisfy $X_{ij}^* = X_{pq}^*$.*

We begin by proving a critical lemma. While Fürer (1995; 2010) have proven that standard 2-FWL refines the spectral decomposition of the graph adjacency matrix, we contribute a similar result specifically for a multiset 2-FWL variant. Notably, we employ a novel proof technique to demonstrate that, despite its reduced expressivity, the multiset 2-FWL retains this crucial spectral refinement property.

**Lemma C.15.** *Let $M \in \mathbb{S}^n$ be a symmetric matrix with spectral decomposition*

$$M = \sum_{k=1}^m \lambda_k P_k$$

*where $\lambda_1 > \lambda_2 \cdots > \lambda_m$ are the non repeating eigenvalues, and $P_k$ are Frobenius covariants (Horn & Johnson, 1994, p. 437), i.e., the symmetric matrices describing the projection onto the eigenspace of eigenvalue $\lambda_k$.*

*The stable coloring $v^\infty$ are produced by the multiset* 2-FWL *algorithm:*

$$v_{ij}^0 = \mathsf{hash}\,(M_{ij}, \mathbb{I}_{i=j})$$
$$v_{ij}^t := \mathsf{hash}\left(v_{ij}^{t-1}, \{\!\{\{\!\{v_{uj}^{t-1}, v_{iu}^{t-1}\}\!\} \mid u \in [n]\}\!\}\right), \; \forall t > 0,$$

*where $\mathbb{I}_{i=j}$ is a diagonal indicator which takes value 1 if $i = j$ otherwise 0. The following result holds:*

$$v_{ij}^\infty = v_{pq}^\infty \implies (P_k)_{ij} = (P_k)_{pq}, \quad \forall k \in [m].$$

*Proof.* The proof is based on the fact that the spectral projectors $P_k$ are polynomials in $M$ via Sylvester's formula. We say a coloring $v$ refines a matrix $A$, defined as:

$$v_{ij} = v_{pq} \implies A_{ij} = A_{pq}.$$

We proceed by induction to demonstrate that the multiset aggregation effectively simulates matrix multiplication: if the 2-FWL coloring at an iteration refines the entries of $M^d$, the next iteration refines $M^{d+1}$. Consequently, the stable coloring refines any polynomial of $M$, and by extension, naturally refines all the specific polynomials defining the spectral projectors.

At initialization $t = 0$, the coloring of $v^0$ refines $M^1$ naturally, as they are injectively defined by $M$. Besides, $v^0$ also refines $M^0 = I$, because of the diagonal indicator $\mathbb{I}_{i=j}$.

Assume that at iteration $t$, the coloring $v^t$ refines the matrix power $M^d$:

$$v_{ij}^t = v_{pq}^t \implies \left(M^d\right)_{ij} = \left(M^d\right)_{pq}.$$

We show that the next iteration $v^{t+1}$ will refine $M^{d+1}$:

$$v_{ij}^{t+1} = v_{pq}^{t+1} \implies \left(M^{d+1}\right)_{ij} = \left(M^{d+1}\right)_{pq}.$$

Specifically, by definition of the multiset 2-FWL:

$$v_{ij}^{t+1} = v_{pq}^{t+1}$$
$$\implies \mathsf{hash}\left(v_{ij}^t, \{\!\{\{\!\{v_{uj}^t, v_{iu}^t\}\!\} \mid u \in [n]\}\!\}\right) = \mathsf{hash}\left(v_{pq}^t, \{\!\{\{\!\{v_{uq}^t, v_{pu}^t\}\!\} \mid u \in [n]\}\!\}\right)$$
$$\implies \mathsf{hash}\left(\{\!\{\{\!\{v_{uj}^t, v_{iu}^t\}\!\} \mid u \in [n]\}\!\}\right) = \mathsf{hash}\left(\{\!\{\{\!\{v_{uq}^t, v_{pu}^t\}\!\} \mid u \in [n]\}\!\}\right)$$
$$\implies f\left(\{\!\{g\left(v_{uj}^t, v_{iu}^t\right) \mid u \in [n]\}\!\}\right) = f\left(\{\!\{g\left(v_{uq}^t, v_{pu}^t\right) \mid u \in [n]\}\!\}\right)$$

The last implication holds for all choices of functions $f, g$ due to the injectivity of hash function, as long as $g$ is invariant to the order of inputs. In this specific case, we pick $f(\cdots)$ to be sum over a multiset. The choice for $g$ is trickier, we introduce

extra functions $h_1\left(\boldsymbol{v}_{ij}^t\right) = \left(\boldsymbol{M}^d\right)_{ij}$ and $h_2\left(\boldsymbol{v}_{ij}^t\right) = \left(\boldsymbol{M}^1\right)_{ij}$, and $g(a,b) = h_1(a)h_2(b) + h_2(a)h_1(b)$. The construction for function $h_1$ is valid because of the induction assumption $\boldsymbol{v}_{ij}^t = \boldsymbol{v}_{pq}^t \implies \left(\boldsymbol{M}^d\right)_{ij} = \left(\boldsymbol{M}^d\right)_{pq}$, we can find such a function $h_1$ that maps the colors to the matrix entries of $\boldsymbol{M}^d$. The construction for function $h_2$ is also valid because $\boldsymbol{v}^0$ already refines $\boldsymbol{M}$, and $\boldsymbol{v}^t$ refines $\boldsymbol{v}^0$. Besides, $g$ is invariant to the order of the inputs: $g(a,b) = g(b,a)$, compatible for the multiset.

Further, we have

$$
\begin{aligned}
& f\left(\left\{\!\left\{ g\left(\boldsymbol{v}_{uj}^t, \boldsymbol{v}_{iu}^t\right) \mid u \in [n] \right\}\!\right\}\right) \\
&= \sum_{u \in [n]} g\left(\boldsymbol{v}_{uj}^t, \boldsymbol{v}_{iu}^t\right) \\
&= \sum_{u \in [n]} h_1\left(\boldsymbol{v}_{uj}^t\right) h_2\left(\boldsymbol{v}_{iu}^t\right) + h_2\left(\boldsymbol{v}_{uj}^t\right) h_1\left(\boldsymbol{v}_{iu}^t\right) \\
&= \sum_{u \in [n]} \left(\boldsymbol{M}^d\right)_{uj} \left(\boldsymbol{M}^1\right)_{iu} + \left(\boldsymbol{M}^1\right)_{uj} \left(\boldsymbol{M}^d\right)_{iu} \\
&= \sum_{u \in [n]} \left(\boldsymbol{M}^1\right)_{iu} \left(\boldsymbol{M}^d\right)_{uj} + \sum_{u \in [n]} \left(\boldsymbol{M}^d\right)_{iu} \left(\boldsymbol{M}^1\right)_{uj} \\
&= \left(\boldsymbol{M}\boldsymbol{M}^d\right)_{ij} + \left(\boldsymbol{M}^d\boldsymbol{M}\right)_{ij} \\
&= 2\left(\boldsymbol{M}^{d+1}\right)_{ij}
\end{aligned}
$$

Likewise, we have $f\left(\left\{\!\left\{ g\left(\boldsymbol{v}_{uq}^t, \boldsymbol{v}_{pu}^t\right) \mid u \in [n] \right\}\!\right\}\right) = 2\left(\boldsymbol{M}^{d+1}\right)_{pq}$. Therefore, $\boldsymbol{v}_{ij}^{t+1} = \boldsymbol{v}_{pq}^{t+1} \implies \left(\boldsymbol{M}^{d+1}\right)_{ij} = \left(\boldsymbol{M}^{d+1}\right)_{pq}$.

By induction, the stable coloring $\boldsymbol{v}^\infty$ refines $\boldsymbol{M}^d$ for all $d \in \mathbb{N}$. Consequently, it also refines any matrix polynomial $p(\boldsymbol{M}) = \sum_{d=0}^\infty a_d \boldsymbol{M}^d$. This holds because the expressivity to distinguish the structural features of individual powers $\boldsymbol{M}^d$ implies the ability to distinguish their linear combinations. The real-valued matrix $\boldsymbol{M}$ is symmetric and diagonalizable, therefore we apply spectral decomposition $\boldsymbol{M} = \sum_{k=1}^m \lambda_k \boldsymbol{P}_k$, where matrices $\boldsymbol{P}_k$ are given by the Sylvester's formula (Horn & Johnson, 1994, p. 437):

$$
\boldsymbol{P}_k = \prod_{j \neq k} \frac{\boldsymbol{M} - \lambda_j \boldsymbol{I}}{\lambda_k - \lambda_j}.
$$

Because each $\boldsymbol{P}_k$ is a polynomial in $\boldsymbol{M}$, the stable coloring $\boldsymbol{v}^\infty$ refines each $\boldsymbol{P}_k$. Thus, it holds for all $k$ that $\boldsymbol{v}_{ij}^\infty = \boldsymbol{v}_{pq}^\infty \implies (\boldsymbol{P}_k)_{ij} = (\boldsymbol{P}_k)_{pq}$. □

*Remark* C.16. The lemma above analyzes multiset 2-FWL, a simplified VC-2-FWL algorithm acting solely on a symmetric matrix $\boldsymbol{M}$ without constraint nodes. Note that our full VC-2-FWL architecture Equation (4) subsumes this process. The VC-2-FWL variable update aggregates the same multiset of variable neighbors $\left\{\!\left\{ \left\{\!\left\{ \boldsymbol{v}_{uj}, \boldsymbol{v}_{iu} \right\}\!\right\} \mid u \in [n] \right\}\!\right\}$ together with constraint information. Therefore, VC-2-FWL is no less expressive than the simplified process, as it uses strictly more information for hashing, and any refinement made by the simplified process is guaranteed for VC-2-FWL as well. Thus, the spectral refinement properties proven above apply directly to the VC-2-FWL variable colors. Now with the lemma, we formally prove Theorem 2.3.

*Proof of Theorem 2.3.* We provide a constructive proof by simulating the primal-dual hybrid gradient (PDHG) algorithm (Chambolle & Pock, 2016; Wang et al., 2024). We show that the stable coloring of VC-2-FWL refines every update of PDHG algorithm, which is proven to converge.

As established in Appendix C.3, the PDHG updates for the Frobenius-regularized SDP are given by:

$$
\begin{aligned}
\boldsymbol{X}^{t+1} &:= \mathrm{Proj}_{\mathbb{S}_+^n}\left[ \frac{\boldsymbol{X}^t - \alpha_t \mathcal{A}^*(\boldsymbol{y}^t) - \alpha_t \boldsymbol{C}}{1 + \alpha_t \varepsilon} \right] \\
\boldsymbol{y}^{t+1} &:= \boldsymbol{y}^t + \beta_t \mathcal{A}\left( \boldsymbol{X}^{t+1} + \theta_t\left( \boldsymbol{X}^{t+1} - \boldsymbol{X}^t \right) \right) - \beta_t \boldsymbol{b}.
\end{aligned}
\tag{26}
$$

For any $\varepsilon > 0$, the SDP problem has a unique primal solution $\boldsymbol{X}^*$. And if $\varepsilon \to 0$, it reduces to normal linear SDP. We proceed by induction on $t$, initializing with $\boldsymbol{X}^0 = \boldsymbol{0}_{n \times n}$ and $\boldsymbol{y}^0 = \boldsymbol{0}_m$. Our goal is to show that for all $t \geq 0$, the stable VC-2-FWL coloring on variable and constraint nodes refines the intermediate primal-dual solutions of PDHG updates:

$$
\boldsymbol{v}_{ij}^\infty = \boldsymbol{v}_{pq}^\infty \implies X_{ij}^t = X_{pq}^t
$$
$$
\boldsymbol{c}_k^\infty = \boldsymbol{c}_l^\infty \implies y_k^t = y_l^t.
$$

Before the inductive step, we establish three key facts derived from the properties of the stable coloring.

**Fact 1** Since the stable coloring $\boldsymbol{v}^\infty$ refines the initial coloring $\boldsymbol{v}^0$, and $\boldsymbol{v}^0$ is defined by $\boldsymbol{C}$, it holds that $\boldsymbol{v}_{ij}^\infty = \boldsymbol{v}_{pq}^\infty \implies C_{ij} = C_{pq}$. Furthermore, by applying Lemma C.15 to $\boldsymbol{C}$, if its spectral decomposition is given by $\boldsymbol{C} = \sum_h \lambda_h \boldsymbol{P}_h$, then the projection matrices also respect the coloring: $\forall h \colon \boldsymbol{v}_{ij}^\infty = \boldsymbol{v}_{pq}^\infty \implies (\boldsymbol{P}_h)_{ij} = (\boldsymbol{P}_h)_{pq}$. Besides, we know $\boldsymbol{c}^\infty$ refines $\boldsymbol{b}$ by its initialization $\boldsymbol{c}_k^0 := \mathsf{hash}(b_k)$.

**Fact 2** Consider any two pairs $(i,j), (p,q) \in [n]^2$. According to VC-2-FWL definition Equation (4), if they have the same stable color, their hash function entries must be identical. Specifically, for all choices of function $f$:

$$
\boldsymbol{v}_{ij}^\infty = \boldsymbol{v}_{pq}^\infty
$$
$$
\implies \mathsf{hash}\Big(\boldsymbol{v}_{ij}^\infty, \{\!\{\, \{\!\{\boldsymbol{v}_{uj}^\infty, \boldsymbol{v}_{iu}^\infty\}\!\} \mid u \in [n]\}\!\}, \{\!\{(A_{k,ij}, \boldsymbol{c}_k^\infty) \mid k \in N(ij)\}\!\}\Big) =
$$
$$
\mathsf{hash}\Big(\boldsymbol{v}_{pq}^\infty, \{\!\{\, \{\!\{\boldsymbol{v}_{uq}^\infty, \boldsymbol{v}_{pu}^\infty\}\!\} \mid u \in [n]\}\!\}, \{\!\{(A_{k,pq}, \boldsymbol{c}_k^\infty) \mid k \in N(pq)\}\!\}\Big)
$$
$$
\implies \mathsf{hash}\Big(\{\!\{(A_{k,ij}, \boldsymbol{c}_k^\infty) \mid k \in N(ij)\}\!\}\Big) = \mathsf{hash}\Big(\{\!\{(A_{k,pq}, \boldsymbol{c}_k^\infty) \mid k \in N(pq)\}\!\}\Big)
$$
$$
\implies \sum_{k \in N(ij)} f(A_{k,ij}, \boldsymbol{c}_k^\infty) = \sum_{k \in N(pq)} f(A_{k,pq}, \boldsymbol{c}_k^\infty)
$$
$$
\implies \sum_{k \in N(ij)} A_{k,ij} f(\boldsymbol{c}_k^\infty) = \sum_{k \in N(pq)} A_{k,pq} f(\boldsymbol{c}_k^\infty), \text{(we overload } f \text{ for ease of notation)}
$$
$$
\implies \sum_{k \in [m]} A_{k,ij} f(\boldsymbol{c}_k^\infty) = \sum_{k \in [m]} A_{k,pq} f(\boldsymbol{c}_k^\infty)
$$

The last implication holds as $\forall k \notin N(ij) \colon A_{k,ij} = 0$, by definition of neighbors. Consequently, for any function $f$ mapping the $\boldsymbol{c}^\infty$ to real numbers, the summations over these multisets are identical. In the inductive steps, we pick function $f$ to map stable coloring to intermediate dual variables, $f(\boldsymbol{c}_k^\infty) = y_k^t$ for some suitable $t$, as we will show later.

**Fact 3** Similarly for constraint nodes, consider $k, l \in [m]$. According to Equation (4), if two constraint nodes have the same stable colors, we have, for any function $g$:

$$
\boldsymbol{c}_k^\infty = \boldsymbol{c}_l^\infty
$$
$$
\implies \mathsf{hash}\Big(\boldsymbol{c}_k^\infty, \{\!\{(A_{k,ij}, \boldsymbol{v}_{ij}^\infty) \mid (i,j) \in N(k)\}\!\}\Big) = \mathsf{hash}\Big(\boldsymbol{c}_l^\infty, \{\!\{(A_{l,ij}, \boldsymbol{v}_{ij}^\infty) \mid (i,j) \in N(l)\}\!\}\Big)
$$
$$
\implies \mathsf{hash}\Big(\{\!\{(A_{k,ij}, \boldsymbol{v}_{ij}^\infty) \mid (i,j) \in N(k)\}\!\}\Big) = \mathsf{hash}\Big(\{\!\{(A_{l,ij}, \boldsymbol{v}_{ij}^\infty) \mid (i,j) \in N(l)\}\!\}\Big)
$$
$$
\implies \sum_{(i,j) \in N(k)} g(A_{k,ij}, \boldsymbol{v}_{ij}^\infty) = \sum_{(i,j) \in N(l)} g(A_{l,ij}, \boldsymbol{v}_{ij}^\infty)
$$
$$
\implies \sum_{(i,j) \in N(k)} A_{k,ij} g(\boldsymbol{v}_{ij}^\infty) = \sum_{(i,j) \in N(l)} A_{l,ij} g(\boldsymbol{v}_{ij}^\infty), \text{(we overload } g \text{ for ease of notation)}
$$
$$
\implies \sum_{(i,j) \in [n]^2} A_{k,ij} g(\boldsymbol{v}_{ij}^\infty) = \sum_{(i,j) \in [n]^2} A_{l,ij} g(\boldsymbol{v}_{ij}^\infty)
$$

In the inductive steps, we will pick function $g$ to map stable coloring to intermediate primal variables, similar to the choice of $f$.

**Proof by induction** We now prove the theorem by induction on $t$. At initialization $t = 0$, we have $\boldsymbol{X}^0 = \boldsymbol{0}$ and $\boldsymbol{y}^0 = \boldsymbol{0}$. The implications $\boldsymbol{v}_{ij}^\infty = \boldsymbol{v}_{pq}^\infty \implies X_{ij}^0 = X_{pq}^0$ and $\boldsymbol{c}_k^\infty = \boldsymbol{c}_l^\infty \implies y_k^0 = y_l^0$ hold trivially.

Suppose the hypothesis holds for iteration $t \geq 0$. We show it holds for iteration $t + 1$.

**1. Primal Update.** In the primal update step, we must evaluate the term $\mathcal{A}^*(\boldsymbol{y}^t) = \sum_{k=1}^{m} y_k^t \boldsymbol{A}_k$. We define the function $f$ in Fact 2 as $f(\boldsymbol{c}_k^\infty) = y_k^t$. This function is well-defined because, by the inductive hypothesis, $y_k^t$ is refined by the color $\boldsymbol{c}_k^\infty$, i.e., $\boldsymbol{c}_k^\infty = \boldsymbol{c}_l^\infty \implies y_k^t = y_l^t$. Applying Fact 2 with this $f$, we obtain:

$$\boldsymbol{v}_{ij}^\infty = \boldsymbol{v}_{pq}^\infty \implies \sum_{k \in [m]} A_{k,ij} y_k^t = \sum_{k \in [m]} A_{k,pq} y_k^t.$$

The implication above shows that $\mathcal{A}^*(\boldsymbol{y}^t)$ is refined by the coloring $\boldsymbol{v}^\infty$. Now consider the full term inside the PSD projection in Equation (26):

$$\boldsymbol{Z}^t := \frac{\boldsymbol{X}^t - \alpha_t \mathcal{A}^*(\boldsymbol{y}^t) - \alpha_t \boldsymbol{C}}{1 + \alpha_t \varepsilon}.$$

Since $\boldsymbol{X}^t$ (by hypothesis), $\boldsymbol{C}$ (by Fact 1), and $\mathcal{A}^*(\boldsymbol{y}^t)$ (by derivation) are all refined by the coloring $\boldsymbol{v}^\infty$, their linear combination $\boldsymbol{Z}^t$ also is. Finally, the update is $\boldsymbol{X}^{t+1} = \text{Proj}_{\mathbb{S}_+^n}(\boldsymbol{Z}^t)$. The PSD projection depends only on the spectral decomposition as detailed in Algorithm 1. Specifically, spectral decomposition is performed on $\boldsymbol{Z}^t$, and the Frobenius covariants corresponding to negative eigenvalues are removed. By Lemma C.15, we know $\boldsymbol{v}_{ij}^\infty = \boldsymbol{v}_{pq}^\infty \implies X_{ij}^{t+1} = X_{pq}^{t+1}$, that is, $\boldsymbol{v}^\infty$ refines $\boldsymbol{X}^{t+1}$.

**2. Dual Update.** In the dual update step, we evaluate the term inside the operator $\mathcal{A}$:

$$\boldsymbol{W}^{t+1} := \boldsymbol{X}^{t+1} + \theta_t \left(\boldsymbol{X}^{t+1} - \boldsymbol{X}^t\right).$$

Since both $\boldsymbol{X}^{t+1}$ and $\boldsymbol{X}^t$ are refined by $\boldsymbol{v}^\infty$, so is $\boldsymbol{W}^{t+1}$. We define the function $g$ in Fact 3 as $g\left(\boldsymbol{v}_{ij}^\infty\right) = \left(\boldsymbol{W}^{t+1}\right)_{ij}$. This is well-defined because the value of $\left(\boldsymbol{W}^{t+1}\right)_{ij}$ is refined by the color $\boldsymbol{v}_{ij}^\infty$. Applying Fact 3 with this $g$, we obtain:

$$\boldsymbol{c}_k^\infty = \boldsymbol{c}_l^\infty \implies \left[\mathcal{A}(\boldsymbol{W}^{t+1})\right]_k = \left[\mathcal{A}(\boldsymbol{W}^{t+1})\right]_l.$$

Since $\boldsymbol{y}^t$ (by hypothesis) and $\boldsymbol{b}$ (by Fact 1) are both refined by the coloring $\boldsymbol{c}^\infty$, the sum $\boldsymbol{y}^{t+1}$ is refined by the coloring.

In summary, we have proven by induction that if the stable coloring $\boldsymbol{v}^\infty$ and $\boldsymbol{c}^\infty$ refine $\boldsymbol{X}^t$ and $\boldsymbol{y}^t$ in PDHG algorithm, they also refine $\boldsymbol{X}^{t+1}$ and $\boldsymbol{y}^{t+1}$.

Since PDHG is guaranteed to converge to $\boldsymbol{X}^*$ and $\boldsymbol{y}^*$, this completes the proof. $\square$

*Remark* C.17. Notably, this result extends to comparisons across distinct SDP instances. The pairs $(i, j)$ and $(p, q)$ do not need to belong to the same problem: provided consistent initialization, if their stable VC-2-FWL colorings are identical, their corresponding optimal values are guaranteed to be equal. This implies a form of universality, suggesting that the VC-2-FWL hierarchy captures fundamental structural invariants that generalize across the whole space of SDP problems.

### C.5. Proof for complexity analysis

**Lemma C.18** (Restatement of Theorem 12 of Berkholz et al. (2017))**.** *Let $G = (V, E, c)$ be an edge colored digraph with $N = |V|$ and $M = |E|$. In time $O((N + M) \log(N + M))$, a canonical coarsest edge-color stable coloring can be computed for $G$.*

**Proposition C.19** (Restatement of Proposition 2.4)**.** *Given an SDP of $n \times n$ variables and $m$ constraints, $nnz(\boldsymbol{A})$ denotes the number of non-zero entries in $\boldsymbol{A}$, then VC-2-FWL converges in time*

$$\mathcal{O}\left((n^3 + nnz(\boldsymbol{A})) \log n\right).$$

*Proof.* We analyze the computational complexity of VC-2-FWL by demonstrating that its update rule is algorithmically equivalent to standard 1-WL (color refinement) operating on a directed, edge-colored auxiliary graph. This equivalence allows us to invoke the tight complexity bounds established in Lemma C.18.

**Auxiliary graph construction** Let $G_{\text{aux}} = (V, E)$ be the implicit auxiliary graph defined by the update rules in Equation (4). The node set $V$ is the disjoint union of variable nodes and constraint nodes, and a set of higher-order nodes: $V = V_{\text{var}} \cup V_{\text{con}} \cup V_{\text{tri}}$.

1. $V_{\text{var}}$ consists of the $n^2$ variable nodes. These are initialized with colors $\boldsymbol{a}_{ij}^0 := \alpha_{\text{var}}(C_{ij})$, where $\alpha_{\text{var}}$ is a canonical mapping for variables.

2. $V_{\text{con}}$ consists of the $m$ constraint nodes. These are initialized with colors $\boldsymbol{b}_k^0 := \alpha_{\text{con}}(b_k)$, where $\alpha_{\text{con}}$ is distinct from $\alpha_{\text{var}}$.

3. We further construct a node set $V_{\text{tri}}$. For every variable node with index $(i, j)$, we construct $n$ nodes with index $(i, u, j)$ for $u \in [n]$. These are initialized with colors $\boldsymbol{c}_{iuj}^0 = \alpha_{\text{tri}}(\{\!\{C_{iu}, C_{uj}\}\!\})$.

Since the initialization functions $\alpha_{(\cdot)}$ have disjoint ranges, the color classes of three node sets remain disjoint throughout the execution. The total number of nodes is $N = n^2 + m + n^3$.

The edge set $E$ decomposes into the following:

1. $E_{\text{tri-v}} \subseteq V_{\text{tri}} \times V_{\text{var}}$: These edges connect nodes from $V_{\text{tri}}$ to $V_{\text{var}}$. Specifically, every variable node indexed $(i, j)$ gets links from $\{(i, u, j) \mid u \in [n]\}$. This requires $n^3$ edges. The edge colors can be a constant.

2. $E_{\text{v-tri}} \subseteq V_{\text{var}} \times V_{\text{tri}}$: This is the other direction of above. Every higher order node $(i, u, j)$ gets links from variable nodes $(i, u)$ and $(u, j)$, thus $2n^3$ edges. The edge colors can be a constant.

3. $E_{\text{v-c}} \subseteq V_{\text{var}} \times V_{\text{con}}$: These edges encode the $\mathbf{A}$ structure. An edge exists from $(i, j) \in V_{\text{var}}$ to $k \in V_{\text{con}}$ if and only if $A_{k,ij} \neq 0$. The edge colors are defined by the tensor entries: $e_{ij,k} = \alpha_A(A_{k,ij})$. The size of this set is $\text{nnz}(\mathbf{A})$.

4. $E_{\text{c-v}} \subseteq V_{\text{con}} \times V_{\text{var}}$: These edges are the opposite direction from $E_{\text{v-c}}$. The edge colors are the same: $e_{k,ij} = \alpha_A(A_{k,ij})$.

The total number of edges is $M = 3n^3 + 2\text{nnz}(\mathbf{A})$.

**Update equivalence** We now show that the 1-WL update on this edge-colored graph is equivalent to the VC-2-FWL update.

For a constraint node, it updates in the sense of 1-WL as:

$$\boldsymbol{b}_k^{t+1} := \mathsf{hash}\left(\boldsymbol{b}_k^t, \{\!\{\left(e_{ij,k}^t, \boldsymbol{a}_{ij}^t\right) \mid (i, j) \colon A_{k,ij} \neq 0\}\!\}\right)$$

which has exactly the same form as in Equation (4).

For a variable node, we first update the higher-order nodes:

$$\boldsymbol{c}_{iuj}^{t+1/2} := \mathsf{hash}\left(\boldsymbol{c}_{iuj}^t, \{\!\{\boldsymbol{v}_{iu}^t, \boldsymbol{v}_{uj}^t\}\!\}\right),$$

regardless of the edge colors. Then, we update the variable nodes:

$$\boldsymbol{a}_{ij}^{t+1} := \mathsf{hash}\left(\boldsymbol{a}_{ij}^t, \{\!\{\boldsymbol{c}_{i0j}^{t+1/2}, \cdots, \boldsymbol{c}_{inj}^{t+1/2}, (e_{k,ij}, \boldsymbol{b}_k^t)\cdots\}\!\}\right)$$

Since the node types $V_{\text{tri}}$ and $V_{\text{con}}$ are distinguished by disjoint color spaces, we can partition the multiset into two distinct multisets without ambiguity:

$$\boldsymbol{a}_{ij}^{t+1} := \mathsf{hash}\left(\boldsymbol{a}_{ij}^t, \{\!\{\boldsymbol{c}_{iuj}^{t+1/2} \mid u \in [n]\}\!\}, \{\!\{(e_{k,ij}, \boldsymbol{b}_k^t) \mid k \colon A_{k,ij} \neq 0\}\!\}\right).$$

This form matches the variable update in Equation (4). Therefore, the variable update in Equation (4) can be implemented with 2 steps of 1-WL.

Hence, we have shown the equivalence of our VC-2-FWL and standard 1-WL. Applying the bound in Lemma C.18 to our setting: $N = |V| = n^2 + m + n^3$ and $M = |E| = \mathcal{O}(n^3 + \text{nnz}(\mathbf{A}))$. Since $m$ is bounded by $\mathcal{O}(n^2)$ as we assume linearly independent constraints, $\mathcal{O}((N + M)) = \mathcal{O}(n^3 + \text{nnz}(\mathbf{A}))$, and $\mathcal{O}(\log(N + M)) = \mathcal{O}(\log n)$. Substituting these values yields the total time complexity: $\mathcal{O}\big((n^3 + \text{nnz}(\mathbf{A})) \log n\big)$. $\qquad \square$

### C.6. Proof for neuralization

*Remark* C.20. It is important to clarify the terminology regarding the hash function utilized in the WL test aggregation step. In standard computer science contexts, cryptographic or data-structure hash functions are inherently discrete and non-continuous. However, in the theoretical framework of GNN expressivity, a hash function simply denotes an injective mapping that assigns a unique, deterministic representation to any distinct multiset of neighbor features. Crucially, this

injective mapping is not required to be discrete. As established in the foundational GNN expressivity literature (Morris et al., 2019; Xu et al., 2019), there exist continuous, injective functions capable of perfectly resolving bounded multisets. Because these idealized WL aggregation functions can be chosen to be strictly continuous, they satisfy the necessary conditions to be arbitrarily approximated by MLPs via the universal approximation theorem.

**Proposition C.21** (Restatement of Proposition 2.5). *There exists a set of parameters for the functions* INIT, UPD, MSG, MAP, *such that* VC-2-FMPNN *has maximal expressivity equal to the* VC-2-FWL.

First, we state a lemma.

**Lemma C.22.** *Let $\mathcal{K} \subset \mathbb{R}^d$ be a compact set, and let $\mathcal{X}$ be the space of finite multisets of bounded maximum size with elements drawn from $\mathcal{K}$. There exists an integer $p \geq 1$ and a continuous, injective function*

$$f \colon \mathcal{X} \to \mathbb{R}^p$$

*such that for any two multisets $X_1, X_2 \in \mathcal{X}$,*

$$f(X_1) = f(X_2) \iff X_1 = X_2.$$

This is exactly a restatement of Lemma 4 of Xu et al. (2019), if the multiset space is discrete and countable; or of Proposition 1 of Maron et al. (2019a), if it is a multiset of real number space, by using the power-sum multi-symmetric polynomial technique. In our case, unlike discrete graphs, we assume that the problem coefficients $C, \mathbf{A}, \boldsymbol{b}$ as well as the intermediate outputs of the neural networks lie within a compact subset $\mathcal{K}$, i.e., real space where the coefficients don't go to infinity. This ensures that the inputs are bounded and allows for the application of the Stone-Weierstrass theorem for approximation. So we can reuse the proof technique in Maron et al. (2019a). Now let's prove Proposition 2.5.

*Proof for Proposition 2.5.* We prove the equality of expressivity by showing both directions of refinement.

VC-2-FWL $\sqsubseteq$ VC-2-FMPNN: First, VC-2-FMPNN is no more expressive than VC-2-FWL. It is easy to verify with the same logic as Theorem 1 of Morris et al. (2019). The VC-2-FMPNN updates are deterministic functions of the unrolling trees defined by the VC-2-FWL color refinement. Therefore, if VC-2-FWL produces identical colors for two variable pairs, VC-2-FMPNN must produce identical embeddings.

VC-2-FMPNN $\sqsubseteq$ VC-2-FWL: We show that VC-2-FMPNN can distinguish any two instances distinguishable by VC-2-FWL at every iteration $t$.

At initialization $t = 0$, the initial features are directly encoded from the problem data. Since we assume the input space $\mathcal{K}$ is compact, there exist MLPs by the universal approximation theorem that can implement the initialization functions injectively, preserving the distinction between different initial values of $C_{ij}$ or $b_k$. At iteration $t > 0$, we consider the update for constraint and variable nodes, respectively.

**Constraint node** Consider the update of a constraint node $k$. The update requires an injective mapping of the tuple containing the constraint node's previous feature $\boldsymbol{h}_k^{t-1}$ and the multiset of its variable neighbors' features together with the edge weights: $\left\{\!\!\left\{ \left( A_{k,ij}, \boldsymbol{v}_{ij}^{t-1} \right) \mid (i,j) \in N(k) \right\}\!\!\right\}$. According to Lemma C.22, there exists an injective function that uniquely encodes this multiset. As shown in the constructive proof of Maron et al. (2019a, Proposition 1), this injective function can be realized as a summation of power-sum polynomials. Consequently, the MLP in the term

$$\boldsymbol{m}_{k,\mathrm{v}\to\mathrm{c}}^t := \sum_{(i,j)\in N(k)} \mathsf{MSG}_{\mathrm{v}\to\mathrm{c}}^t \left( A_{k,ij}, \boldsymbol{h}_{ij}^{t-1} \right)$$

can be parameterized to approximate these continuous mappings, ensuring that the summation yields a unique representation of the multiset. Furthermore, the injective combination of the node's previous state $\boldsymbol{h}_k^{t-1}$ and this multiset embedding $\boldsymbol{m}_{k,\mathrm{v}\to\mathrm{c}}^t$ can be approximated arbitrarily well by concatenation of them and application of another MLP:

$$\boldsymbol{h}_k^t := \mathsf{UPD}_{\mathrm{c}}^t \left( \boldsymbol{h}_k^{t-1}, \boldsymbol{m}_{k,\mathrm{v}\to\mathrm{c}}^t \right).$$

Since the intermediate space $\mathcal{K}$ is compact, there exist MLPs that universally approximate the injective hash function in Equation (4).

**Variable node** Consider the VC-2-FWL update for a variable index $(i, j)$. VC-2-FWL updates the variable color based on the variable node's previous feature, the multiset of constraint neighbors and the multiset of variable neighbor pairs. For the multiset of constraint neighbors, it aggregates the multiset $\{\!\{(A_{k,ij}, c_k^{t-1}) \mid k \in N(ij)\}\!\}$, we apply Lemma C.22 once, and the term

$$m_{ij,c \to v}^t := \sum_{k \in N(ij)} \mathsf{MSG}_{c \to v}^t \left(A_{k,ij}, h_k^{t-1}\right)$$

implements the injective mapping from constraint neighbors to a vector embedding. The multiset of variable pairs are formed by $\{\!\{\{\!\{v_{uj}^{t-1}, v_{iu}^{t-1}\}\!\} \mid u \in [n]\}\!\}$. To complete this injectively, we invoke Lemma C.22 twice. First, for a specific $u$, the unordered pair $s_u = \{\!\{v_{uj}^{t-1}, v_{iu}^{t-1}\}\!\}$ constitutes a 2-multiset. By Lemma C.22, there exists an MLP capturing the injectivity of $s_u$. The term

$$\mathsf{MAP}^t \left(h_{uj}^{t-1}\right) + \mathsf{MAP}^t \left(h_{iu}^{t-1}\right)$$

explicitly implements this. Second, we must aggregate these encoded pairs over all $u \in [n]$. The target structure is the outer multiset $S_{ij} = \{\!\{s_u \mid u \in [n]\}\!\}$. Applying Lemma C.22 again, there exists an injective mapping with MLP. Thus, the term

$$m_{ij,v \to v}^t := \sum_{u \in [n]} \mathsf{MSG}_{v \to v}^t \left(\mathsf{MAP}^t \left(h_{uj}^{t-1}\right) + \mathsf{MAP}^t \left(h_{iu}^{t-1}\right)\right)$$

injectively encodes the nested multiset structure for the VC-2-FWL update. Finally, one can simply concatenate the embeddings $\left(h_{ij}^{t-1}, m_{ij,v \to v}^t, m_{ij,c \to v}^t\right)$ and update it with the MLP $\mathsf{UPD}_v^t$:

$$h_{ij}^t := \mathsf{UPD}_v^t \left(h_{ij}^{t-1}, m_{ij,v \to v}^t, m_{ij,c \to v}^t\right).$$

In summary, every component of VC-2-FWL in Equation (4) can be approximated by an MLP. By picking the output approximation error $\epsilon$ to be smaller than the separation margin of the distinct VC-2-FWL colors, VC-2-FMPNN maintains the injectivity of the update step.

Thus, VC-2-FMPNN is at least as expressive as VC-2-FWL, which completes the proof. $\qquad\square$

**VC-MPNN** Similar to VC-2-FMPNN, we can neuralize the VC-WL to VC-MPNN, whose initialization is identical to Equation (5), and the update steps:

$$
\begin{aligned}
m_{ij,c \to v}^t &:= \sum_{k \in N(ij)} \mathsf{MSG}_{c \to v}^t \left(A_{k,ij}, h_k^{t-1}\right) \\
m_{k,v \to c}^t &:= \sum_{(i,j) \in N(k)} \mathsf{MSG}_{v \to c}^t \left(A_{k,ij}, h_{ij}^{t-1}\right) \\
h_{ij}^t &:= \mathsf{UPD}_v^t \left(h_{ij}^{t-1}, m_{ij,c \to v}^t\right) \\
h_k^t &:= \mathsf{UPD}_c^t \left(h_k^{t-1}, m_{k,v \to c}^t\right)
\end{aligned}
\tag{27}
$$

**Proposition C.23.** *There exists a set of parameters for the functions* $\mathsf{INIT}, \mathsf{UPD}, \mathsf{MSG}$, *such that* VC-MPNN *has maximal expressivity equal to the* VC-WL.

*Proof.* We apply the same logic as the proof of Proposition 2.5, simplified for the reduced message passing structure.

VC-WL $\sqsubseteq$ VC-MPNN: Since the VC-MPNN update rules are deterministic functions of the VC-WL unrolling tree (which is actually a subset of the VC-2-FWL tree), the neural network cannot distinguish elements that the color refinement algorithm cannot.

VC-MPNN $\sqsubseteq$ VC-WL: We verify that the network components can implement the injective update steps of VC-WL. At initialization $t = 0$, similar to Proposition 2.5, there exist MLPs that perform this mapping injectively. For iteration $t > 0$:

**Constraint node** The constraint node update simply follows Proposition 2.5.

**Variable node** The VC-WL variable update aggregates only the constraint neighbors, omitting the variable node interactions. This is a simplified case of the variable update in Equation (6). The message term $m_{ij,c \to v}^t$ with an MLP is sufficient to

injectively encode the constraint neighbor multiset $\{\{(A_{k,ij}, c_k^{t-1}) \mid k \in N(ij)\}\}$. The $\text{UPD}_v^t$ then injectively maps the pair $(h_{ij}^{t-1}, m_{ij,c \to v}^t)$ to the next state.

Thus, VC-MPNN is capable of simulating VC-WL injectively. □

Building on the limitation of VC-WL established in Proposition 2.2, we demonstrate that the neuralized VC-MPNN architecture cannot universally approximate SDP solutions. We rely on the following homogeneity property of linear SDPs.

**Lemma C.24.** *Consider a standard form SDP with data $(C, \mathbf{A}, b)$ and a triplet of primal, dual and slack solution $(X^*, y^*, S^*)$. If the vector $b$ is scaled by a positive scalar $\alpha > 0$ to $\alpha b$, then $\alpha X^*$ is an optimal primal solution to the scaled problem.*

*Proof.* Optimality in linear SDPs is characterized by the Karush-Kuhn-Tucker (KKT) conditions. The original solution satisfies:

$$
\begin{aligned}
\mathcal{A}(X^*) &= b \\
C + \mathcal{A}^*(y^*) - S^* &= 0 \\
X^* &\succeq 0 \\
S^* &\succeq 0 \\
\langle X^*, S^* \rangle &= 0
\end{aligned}
\tag{28}
$$

It is not hard to find that the following also holds:

$$
\begin{aligned}
\mathcal{A}(\alpha X^*) &= \alpha \mathcal{A}(X^*) = \alpha b \\
C + \mathcal{A}^*(y^*) - S^* &= 0 \\
\alpha X^* &\succeq 0 \\
S^* &\succeq 0 \\
\langle \alpha X^*, S^* \rangle &= 0.
\end{aligned}
\tag{29}
$$

Therefore, $\alpha X^*$ is the solution to the scaled SDP problem; thus, it completes the proof. □

**Corollary C.25.** *The architecture* VC-MPNN *is not a universal approximator for SDPs. Specifically, for any $\epsilon > 0$, there exists an SDP instance such that for any parameters of a* VC-MPNN, *the error between the predicted solution $X'$ and the optimal solution $X^*$ satisfies*

$$
\|X^* - X'\|_\infty \geq \epsilon.
$$

*Proof.* According to Proposition C.23, the expressivity of VC-MPNN is bounded by the VC-WL coloring algorithm.

Consider the counterexample from the proof of Proposition 2.1. In this instance, variables $X_{11}$ and $X_{33}$ are structurally indistinguishable under the VC-WL algorithm, and thus VC-MPNN forces the network to predict identical values, i.e., $X_{11}' = X_{33}' = v$. However, in this case, the unique optimal solution satisfies $X_{11}^* \neq X_{33}^*$. Let $\delta = |X_{11}^* - X_{33}^*| > 0$. Now, scale the constraint vector to $\alpha b$ with $\alpha = 2\epsilon/\delta$. By Lemma C.24, the new optimal solution is $\alpha X^*$, and the gap between the entries becomes $|\alpha X_{11}^* - \alpha X_{33}^*| = \alpha\delta = 2\epsilon$. Since the scaling of $b$ preserves the structural symmetry of the graph, the unrolling trees for variable nodes $v_{11}$ and $v_{33}$ remain isomorphic, the VC-MPNN must still predict identical values for these entries, say $v'$. By the triangle inequality, the approximation error is bounded from below:

$$
\begin{aligned}
2\epsilon = |\alpha X_{11}^* - \alpha X_{33}^*| \\
\leq |\alpha X_{11}^* - v'| + |v' - \alpha X_{33}^*| \\
\leq 2\max\left(|\alpha X_{11}^* - v'|, |\alpha X_{33}^* - v'|\right) \\
\leq 2\|\alpha X^* - X'\|_\infty.
\end{aligned}
$$

Therefore, $\|\alpha X^* - X'\|_\infty \geq \epsilon$, proving that the error can be made arbitrarily large. □

**VC-2-MPNN** We neuralize the VC-2-WL and instantiate a neural network, which we name VC-2-MPNN. The initialization follows Equation (5), and the update is

$$
\begin{aligned}
\boldsymbol{m}_{ij,\text{row}}^t &:= \sum_{u \in [n]} \mathsf{MSG}_{\text{row}}^t(\boldsymbol{h}_{iu}^{t-1}) \\
\boldsymbol{m}_{ij,\text{col}}^t &:= \sum_{u \in [n]} \mathsf{MSG}_{\text{col}}^t(\boldsymbol{h}_{uj}^{t-1}) \\
\boldsymbol{m}_{ij,\text{c}\to\text{v}}^t &:= \sum_{k \in N(ij)} \mathsf{MSG}_{\text{c}\to\text{v}}^t\left(A_{k,ij}, \boldsymbol{h}_k^{t-1}\right) \\
\boldsymbol{m}_{k,\text{v}\to\text{c}}^t &:= \sum_{(i,j) \in N(k)} \mathsf{MSG}_{\text{v}\to\text{c}}^t\left(A_{k,ij}, \boldsymbol{h}_{ij}^{t-1}\right) \\
\boldsymbol{h}_{ij}^t &:= \mathsf{UPD}_\text{v}^t\left(\boldsymbol{h}_{ij}^{t-1}, \boldsymbol{m}_{ij,\text{col}}^t, \boldsymbol{m}_{ij,\text{row}}^t, \boldsymbol{m}_{ij,\text{c}\to\text{v}}^t\right) \\
\boldsymbol{h}_k^t &:= \mathsf{UPD}_\text{c}^t\left(\boldsymbol{h}_k^{t-1}, \boldsymbol{m}_{k,\text{v}\to\text{c}}^t\right)
\end{aligned}
\tag{30}
$$

Besides, following Equation (3), we force symmetry by $\boldsymbol{h}_{ji}^t = \boldsymbol{h}_{ji}^t$, if $i < j$.

**Proposition C.26.** *There exists a set of parameters for the functions* INIT, UPD, MSG, *such that* VC-2-MPNN *has maximal expressivity equal to the* VC-2-WL.

*Proof.* We apply the same logic as the proof of Proposition 2.5, tailored for the message passing structure of VC-2-MPNN.

VC-2-WL $\sqsubseteq$ VC-2-MPNN: Since the VC-2-MPNN update rules are deterministic functions of the VC-2-WL unrolling tree, the neural network cannot distinguish elements that the color refinement algorithm cannot.

VC-2-MPNN $\sqsubseteq$ VC-2-WL: We verify that the network components can implement the injective update steps of VC-2-WL. At initialization $t = 0$, similar to Proposition 2.5, there exist MLPs that perform this mapping injectively.

**Constraint node** The constraint node update simply follows Proposition 2.5.

**Variable node** The VC-2-WL variable update aggregates the constraint neighbors, as well as the row and column variable neighbors separately. The message from constraint neighbors $\boldsymbol{m}_{ij,\text{c}\to\text{v}}^t$ with an MLP is sufficient to injectively encode the constraint neighbor multiset $\{\!\{(A_{k,ij}, \boldsymbol{c}_k^{t-1}) \mid k \in N(ij)\}\!\}$. Similarly, the multisets of row and column neighbors $\boldsymbol{m}_{ij,\text{row}}^t$ and $\boldsymbol{m}_{ij,\text{col}}^t$ can be encoded injectively with MLPs respectively. Finally, the update function $\mathsf{UPD}_\text{v}^t$ then injectively maps the vectors $(\boldsymbol{h}_{ij}^{t-1}, \boldsymbol{m}_{ij,\text{col}}^t, \boldsymbol{m}_{ij,\text{row}}^t, \boldsymbol{m}_{ij,\text{c}\to\text{v}}^t)$ to the next state.

Thus, VC-2-MPNN is capable of simulating VC-2-WL injectively. $\qquad\square$

Next, we show a corollary that VC-2-MPNN is not a universal approximator to represent SDP.

**Corollary C.27.** VC-2-MPNN *is not a universal approximator for SDPs. Specifically, for any $\epsilon > 0$, there exists an SDP instance such that for any parameters of a* VC-2-MPNN*, the error between the predicted solution $\boldsymbol{X}'$ and the optimal solution $\boldsymbol{X}^*$ satisfies*

$$
\|\boldsymbol{X}^* - \boldsymbol{X}'\|_\infty \geq \epsilon.
$$

*Proof.* We follow exactly the proof logic in Corollary C.25. In this case, for VC-2-MPNN, we can pick the example in the proof of Proposition 2.2 where VC-2-WL fails. $\qquad\square$

## D. SDP application

### D.1. Relaxation of CO

In reality, SDP serves as the continuous relaxation of several graph CO problems.

**Max-Cut** (Goemans & Williams, 1995). Let $G$ be a graph with nodes $V$ and edges $E$ with edge weights $w_{ij} \geq 0$, $\{i, j\} \in E$. The goal of the max-cut problem is to partition the nodes into two disjoint sets such that the sum of weights of edges

crossing the cut is maximized. The problem formulation is

$$\max_{\boldsymbol{x}\in\{-1,1\}^n} \quad \frac{1}{2} \sum_{\{i,j\}\in E} w_{ij}(1 - x_i x_j), \tag{31}$$

where 1 and $-1$ denote different partitions. Instead of integer domain $\{-1,1\}$, we relax each variable $x_i$ to a unit vector $\|\boldsymbol{v}_i\| = 1$, which defines $X_{ij} = \boldsymbol{v}_i^\mathsf{T} \boldsymbol{v}_j$ (Yau et al., 2024). Thus, SDP relaxation is

$$\max_{\boldsymbol{X}\in\mathbb{S}_+^n} \quad \frac{1}{2} \sum_{\{i,j\}\in E} w_{ij}(1 - X_{ij})$$
$$s.t. \quad X_{ii} = 1, \forall i \in [n] \tag{32}$$

**Max clique problem** Let $G$ be a graph with nodes $V$ and edges $E$, let $x_i \in \{0,1\}$ denote whether $x_i$ is included in a clique. The max clique problem can be formulated as

$$\max_{\boldsymbol{x}\in\{0,1\}^n} \quad \sum_i x_i$$
$$s.t. \quad x_i x_j = 0, \forall\{i,j\} \notin E. \tag{33}$$

We know that the Lovász function (Galli & Letchford, 2017) is defined as the optimal value of the SDP relaxation of max-clique problem:

$$\max_{\boldsymbol{X}\in\mathbb{S}_+^n} \quad \langle \boldsymbol{J}, \boldsymbol{X} \rangle$$
$$s.t. \quad \mathrm{Tr}(\boldsymbol{X}) = 1$$
$$X_{ij} = 0, \forall\{i,j\} \notin E, \tag{34}$$

where $\boldsymbol{J}$ is a ones matrix.

**Max independent set** The max independent set problem is the complement of the max clique problem. Formally,

$$\max_{\boldsymbol{x}\in\{0,1\}^n} \quad \sum_i x_i$$
$$s.t. \quad x_i x_j = 0, \forall\{i,j\} \in E. \tag{35}$$

and its relaxation:

$$\max_{\boldsymbol{X}\in\mathbb{S}_+^n} \quad \langle \boldsymbol{J}, \boldsymbol{X} \rangle$$
$$s.t. \quad \mathrm{Tr}(\boldsymbol{X}) = 1$$
$$X_{ij} = 0, \forall\{i,j\} \in E. \tag{36}$$

**Graph coloring** The goal of the graph coloring problem is to assign colors to nodes such that no adjacent nodes share the same color. Let $y_k \in \{0,1\}, k \in [K]$ indicate whether the color $k$ is used, where $K$ is an upper bound, e.g., $K = |V|$. And let $x_{ik} \in \{0,1\}$ indicate whether the node $i$ is assigned to color $k$. The graph coloring problem in ILP form is defined as:

$$\min \quad \sum_k y_k$$
$$s.t. \quad \sum_k x_{ik} = 1$$
$$x_{ik} + x_{jk} \le y_k, \forall\{i,j\} \in E \tag{37}$$
$$x_{ik} \le y_k.$$

We have the SDP relaxation (Charikar, 2002):

$$\min_{\boldsymbol{X}\in\mathbb{S}_+^n} \quad k$$
$$s.t. \quad X_{ij} = -\frac{1}{k-1}, \forall\{i,j\} \in E \tag{38}$$
$$X_{ii} = 1.$$

**Min vertex cover** A standard formulation of the minimum vertex cover problem as a quadratic integer program is as following:

$$\min_{\boldsymbol{x} \in \{-1,1\}^n} \quad \sum_i \frac{1}{2}(1 + x_i) \tag{39}$$
$$s.t. \quad (1 - x_i)(1 - x_j) = 0, \forall \{i, j\} \in E$$

The SDP relaxation is as following (Kleinberg & Goemans, 1998; Hatami et al., 2009), where we introduce an extra variable:

$$\min_{\boldsymbol{X} \in \mathbb{S}_+^{n+1}} \quad \sum_i \frac{1}{2}(1 + X_{0i}) \tag{40}$$
$$s.t. \quad X_{ij} + X_{00} - X_{0i} - X_{0j} = 0, \forall \{i, j\} \in E$$
$$X_{ii} = 1, i \in \{0\} \cup V.$$

**Max 2-SAT** A max SAT problem is given a number of variables $n$, that takes on the values $\{-1, 1\}^n$, and a collection of clauses $\{C_1, C_2, \cdots, C_k\}$, each clause is a disjunction (logical OR) of a subset of literals. A literal is either a variable or its negation. Each clause may be assigned a weight, and the goal of Max SAT is to find an assignment of variables that maximizes the total weight of satisfied clauses. A max $k$-SAT problem is one in which each clause has at most $k$ literals.

An instance with $n$ variables and $k$ clauses can be represented by a matrix $\boldsymbol{A} \in \{-1, 0, 1\}^{k \times n}$, where $A_{ij} = 1$ if clause $i$ includes the positive literal of variable $j$, $A_{ij} = -1$ if it includes the negated literal, and $A_{ij} = 0$ otherwise. We consider the unweighted setting, where all clause weights are equal, and the objective is to maximize the number of satisfied clauses. The quadratic assignment formulation of the max 2-SAT problem is (de Klerk & Warners, 2002)

$$\min_{\boldsymbol{x} \in \{-1,1\}^n} \quad \frac{1}{8} \left( \boldsymbol{x}^\mathsf{T} \boldsymbol{A}^\mathsf{T} \boldsymbol{A} \boldsymbol{x} - 2 \boldsymbol{1}^\mathsf{T} \boldsymbol{A} \boldsymbol{x} \right) \tag{41}$$

whose SDP relaxation is

$$\min_{\boldsymbol{X} \in \mathbb{S}_+^{n+1}} \quad \frac{1}{8} \left\langle \begin{bmatrix} \boldsymbol{A}^\mathsf{T} \boldsymbol{A} - \text{diag}(\boldsymbol{A}^\mathsf{T} \boldsymbol{A}) & -\boldsymbol{A}^\mathsf{T} \boldsymbol{1} \\ -\boldsymbol{1}^\mathsf{T} \boldsymbol{A} & 0 \end{bmatrix}, \boldsymbol{X} \right\rangle \tag{42}$$
$$s.t. \quad X_{ii} = 1, \forall i \in [n].$$

### D.2. Quadratic assignment problem

Besides, SDP is also a common relaxation for quadratic assignment problems (QAP).

**Traveling salesman problem** It is already known that the symmetric traveling salesman problem (TSP) is a special case of QAP. We first introduce the notation of circulant matrices on $n$ vertices $\mathcal{C}^n$. A circulant matrix in $\mathbb{R}^{n \times n}$ has the form

$$\begin{bmatrix} c_0 & c_1 & c_2 & c_3 & \cdots & c_{n-1} \\ c_{n-1} & c_0 & c_1 & c_2 & \cdots & c_{n-2} \\ c_{n-2} & c_{n-1} & c_0 & c_1 & \cdots & c_{n-3} \\ \vdots & \vdots & \vdots & \vdots & \ddots & \vdots \\ c_1 & c_2 & c_3 & c_4 & \cdots & c_0 \end{bmatrix}$$

We have a basis for symmetric circulant matrices consisting of $\{\boldsymbol{C}_1^n, \boldsymbol{C}_2^n, \cdots, \boldsymbol{C}_d^n\}$, where $\boldsymbol{C}_i^n$ is a matrix with $c_i = c_{n-i} = 1$ and 0 elsewhere. Besides, we have a symmetric matrix $\boldsymbol{D}$ where $D_{ii} = 0$ and $D_{ij} = D_{ji} > 0$ is the cost per edge. The objective of TSP is

$$\min_{\boldsymbol{X} \in \mathbb{P}^n} \frac{1}{2} \text{Tr}(\boldsymbol{D} \boldsymbol{X} \boldsymbol{C}_1^n \boldsymbol{X}^\mathsf{T}), \tag{43}$$

where $\mathbb{P}$ is the permutation matrix group. $\boldsymbol{X} \boldsymbol{C}_1^n \boldsymbol{X}$ can be interpreted as the adjacency matrix of the tour.

**(Sub)graph isomorphism problem** Given adjacency matrices $\boldsymbol{A}, \boldsymbol{B}$, supposed to be the same shape, we seek the largest isomorphic subgraphs by

$$\min_{\boldsymbol{X} \in \mathbb{P}^n} \|\boldsymbol{X} \boldsymbol{A} - \boldsymbol{B} \boldsymbol{X}\|_F. \tag{44}$$

With some transformations, we have $\|\boldsymbol{X}\boldsymbol{A} - \boldsymbol{B}\boldsymbol{X}\|_F = \|\boldsymbol{A}\|_F - 2\operatorname{Tr}(\boldsymbol{B}\boldsymbol{X}\boldsymbol{A}\boldsymbol{X}^\mathsf{T}) + \|\boldsymbol{B}\|_F$, thus, the problem is equivalent to

$$\max_{\boldsymbol{X}\in\mathbb{P}^n} \operatorname{Tr}(\boldsymbol{B}\boldsymbol{X}\boldsymbol{A}\boldsymbol{X}^\mathsf{T}). \tag{45}$$

The TSP and GIP are special cases of QAP. The relaxation of SDP for QAP has been well-studied (Povh & Rendl, 2009; Oliveira et al., 2018).

### D.3. Subsumption of convex optimization

SDP subsumes a lot of convex optimization problems, including second-order conic programming (SOCP), quadratically-constrained quadratic programming (QCQP), linearly-constrained quadratic programming (LCQP), and linear programming (LP) (Dattorro, 2010, p. 220).

**LP** It is straightforward to show that LP is a special case of SDP. Given an LP instance

$$\begin{aligned} \min_{\boldsymbol{x}\in\mathbb{R}^n_{\geq 0}} \quad & \boldsymbol{c}^\mathsf{T}\boldsymbol{x} \\ \text{s.t.} \quad & \boldsymbol{A}\boldsymbol{x} = \boldsymbol{b}, \end{aligned} \tag{46}$$

we can turn it into an SDP instance by constructing $\boldsymbol{C} = \operatorname{diag}(\boldsymbol{c})$ and $\boldsymbol{A}_i = \operatorname{diag}(\boldsymbol{A}_i)$, and extra constraints $X_{ij} = 0, i \neq j$. The positive semidefiniteness of SDP variables perfectly captures the positivity of the LP variables, as the SDP variables are forced to form a diagonal matrix.

**SOCP** The formulation of an SOCP is

$$\begin{aligned} \min \quad & \boldsymbol{c}^\mathsf{T}\boldsymbol{x} \\ \text{s.t.} \quad & \|\boldsymbol{A}_{(i)}\boldsymbol{x} + \boldsymbol{b}_{(i)}\|_2 \leq \boldsymbol{c}_{(i)}^\mathsf{T}\boldsymbol{x} + d_{(i)}, \forall i \in [m], \end{aligned} \tag{47}$$

where $\boldsymbol{A}_{(i)} \in \mathbb{R}^{k_i \times n}, \boldsymbol{b}_{(i)} \in \mathbb{R}^{k_i}, \boldsymbol{c}_{(i)} \in \mathbb{R}^n, d_{(i)} \in \mathbb{R}$. The positivity constraints of variables can also be captured by such forms. Each constraint is in a second order cone $\mathcal{L}^{k+1} := \{(\boldsymbol{x}, t) \in \mathbb{R}^{k+1} | \|\boldsymbol{x}\|_2 \leq t\}$, hence the name. Using Schur decomposition, we turn the second order cone $\|\boldsymbol{x}\|_2 \leq t$ into a PSD cone $\begin{bmatrix} t & \boldsymbol{x}^\mathsf{T} \\ \boldsymbol{x} & t\boldsymbol{I} \end{bmatrix} \succeq 0$. So we can turn the SOCP problem into an SDP problem (Lobo et al., 1998)

$$\begin{aligned} \min \quad & \boldsymbol{c}^\mathsf{T}\boldsymbol{x} \\ \text{s.t.} \quad & \begin{bmatrix} \boldsymbol{c}_{(i)}^\mathsf{T}\boldsymbol{x} + d_{(i)} & \left(\boldsymbol{A}_{(i)}\boldsymbol{x} + \boldsymbol{b}_{(i)}\right)^\mathsf{T} \\ \boldsymbol{A}_{(i)}\boldsymbol{x} + \boldsymbol{b}_{(i)} & \left(\boldsymbol{c}_{(i)}^\mathsf{T}\boldsymbol{x} + d_{(i)}\right)\boldsymbol{I} \end{bmatrix} \succeq 0, \forall i \in [m]. \end{aligned} \tag{48}$$

Note that it is not the standard primal form of Equation (1), but a dual form

$$\begin{aligned} \min \quad & \boldsymbol{b}^\mathsf{T}\boldsymbol{y} \\ \text{s.t.} \quad & \boldsymbol{C} + \sum_{i=1}^m y_i \boldsymbol{A}_i \succeq 0. \end{aligned} \tag{49}$$

**QP** To show that LCQPs and QCQPs are also special cases of SDP, we just need to show they are included in the SOCP family. Given a QCQP instance

$$\begin{aligned} \min \quad & \boldsymbol{x}^\mathsf{T}\boldsymbol{Q}_{(0)}\boldsymbol{x} + 2\boldsymbol{q}_{(0)}^\mathsf{T}\boldsymbol{x} + q_{(0)} \\ \text{s.t.} \quad & \boldsymbol{x}^\mathsf{T}\boldsymbol{Q}_{(i)}\boldsymbol{x} + 2\boldsymbol{q}_{(i)}^\mathsf{T}\boldsymbol{x} + q_{(i)} \leq 0, \forall i \in [m], \end{aligned} \tag{50}$$

for the problem to be convex, we assume $\boldsymbol{Q}_{(i)}$ to be PSD matrices. This allows us to transform the QCQP in the following form (Lobo et al., 1998):

$$\begin{aligned} \min \quad & \|\boldsymbol{Q}_{(0)}^{1/2}\boldsymbol{x} + \boldsymbol{Q}_{(0)}^{-1/2}\boldsymbol{q}_{(0)}\|_2^2 + c_{(0)} - \boldsymbol{q}_{(0)}^\mathsf{T}\boldsymbol{Q}_{(0)}^{-1}\boldsymbol{q}_{(0)} \\ \text{s.t.} \quad & \|\boldsymbol{Q}_{(i)}^{1/2}\boldsymbol{x} + \boldsymbol{Q}_{(i)}^{-1/2}\boldsymbol{q}_{(i)}\|_2^2 + c_{(i)} - \boldsymbol{q}_{(i)}^\mathsf{T}\boldsymbol{Q}_{(i)}^{-1}\boldsymbol{q}_{(i)} \leq 0, \forall i \in [m], \end{aligned} \tag{51}$$

and further in SOCP form

$$\min \quad t$$
$$\text{s.t.} \ \|\boldsymbol{Q}_{(i)}^{1/2}\boldsymbol{x} + \boldsymbol{Q}_{(i)}^{-1/2}\boldsymbol{q}_{(i)}\|_2 \leq \left(\boldsymbol{q}_{(i)}^{\mathsf{T}}\boldsymbol{Q}_{(i)}^{-1}\boldsymbol{q}_{(i)} - c_{(i)}\right)^{1/2}, \forall i \in [m] \tag{52}$$
$$\|\boldsymbol{Q}_{(0)}^{1/2}\boldsymbol{x} + \boldsymbol{Q}_{(0)}^{-1/2}\boldsymbol{q}_{(0)}\|_2 \leq t.$$

And LCQP is a special case of QCQP.

### D.4. Control theory

**Linear Matrix Inequality** Lyapunov stability analysis is a fundamental problem in control theory used to verify if a dynamical system $\dot{z} = \boldsymbol{A}_{sys}z$ is stable. A necessary and sufficient condition for stability is the existence of a quadratic Lyapunov function $V(z) = z^{\mathsf{T}}\boldsymbol{P}z$ with $\boldsymbol{P} \succ 0$ such that the derivative $\dot{V}(z) < 0$. Formally, this is the Linear Matrix Inequality (LMI):

$$\begin{aligned} \text{Find} \quad & \boldsymbol{P} \in \mathbb{S}_+^n \\ \text{s.t.} \quad & \boldsymbol{A}_{sys}^{\mathsf{T}}\boldsymbol{P} + \boldsymbol{P}\boldsymbol{A}_{sys} \prec 0. \end{aligned} \tag{53}$$

To optimize numerical properties or check for specific performance bounds, this is cast as a Semidefinite Program. We adopt the formulation from Boyd et al. (1994, p. 9), enforcing a trace constraint to bound the solution scale and discretizing the matrix inequality into $m$ scalar linear constraints:

$$\begin{aligned} \min_{\boldsymbol{P} \in \mathbb{S}_+^n} \quad & \langle \boldsymbol{C}, \boldsymbol{P} \rangle \\ \text{s.t.} \quad & \text{Tr}(\boldsymbol{P}) = 1 \\ & \mathcal{A}(\boldsymbol{P}) = \boldsymbol{b}. \end{aligned} \tag{54}$$

where each constraint matrix $\boldsymbol{A}_k$ represents the projection of the Lyapunov operator onto a random direction vector $v_k$:

$$\boldsymbol{A}_k = \frac{1}{2}(\boldsymbol{A}_{sys}v_k v_k^{\mathsf{T}} + v_k v_k^{\mathsf{T}}\boldsymbol{A}_{sys}^{\mathsf{T}}).$$

We employ an inverse strategy to generate feasible problem instances with known stability certificates. We first sample a random positive definite matrix $\boldsymbol{P}_{true}$ and normalize it such that $\text{Tr}(\boldsymbol{P}_{true}) = 1$. Instead of sampling a random system $\boldsymbol{A}_{sys}$ (which might be unstable), we construct a stable system compatible with our certificate. We solve the inverse Lyapunov equation $\boldsymbol{A}_{sys}^{\mathsf{T}}\boldsymbol{P}_{true} + \boldsymbol{P}_{true}\boldsymbol{A}_{sys} = -\epsilon \boldsymbol{I}$ to recover $\boldsymbol{A}_{sys}$. We sample $m$ random sparse direction vectors $v_k$. We compute the RHS scalars $\langle \boldsymbol{A}_k, \boldsymbol{P}_{true} = \boldsymbol{b} \rangle$. This ensures that the ground truth $\boldsymbol{P}_{true}$ lies exactly on the boundary of the feasible region defined by the hyperplanes.

## E. Other architectures

### E.1. Ablation of VC-2-FWL

**VC-2-FWL+** We define a 2-tuple based 2-FWL-like architecture, which we name VC-2-FWL+. The color initialization follows Equation (2), and the update is as defined below:

$$\begin{aligned} \boldsymbol{v}_{ij}^t &:= \mathsf{hash}\left(\boldsymbol{v}_{ij}^{t-1}, \{\!\{(\boldsymbol{v}_{uj}^{t-1}, \boldsymbol{v}_{iu}^{t-1}) \mid u \in [n]\}\!\}, \{\!\{(A_{k,ij}, \boldsymbol{c}_k^{t-1}) \mid k \in N(ij)\}\!\}\right) \\ \boldsymbol{c}_k^t &:= \mathsf{hash}\left(\boldsymbol{c}_k^{t-1}, \{\!\{(A_{k,ij}, \boldsymbol{v}_{ij}^{t-1}) \mid (i,j) \in N(k)\}\!\}\right). \end{aligned} \tag{55}$$

Similar to VC-2-WL, it requires symmetrization after the update $\boldsymbol{c}_{ji}^t \leftarrow \boldsymbol{c}_{ij}^t, \forall i < j$. See Proposition C.7 for the proof of the expressive power between VC-2-FWL and VC-2-FWL+.

Besides, we discuss some interesting yet weaker versions of VC-2-FWL.

**Multiset encoding across constraint**   We can encode the problem instance $I := (\boldsymbol{C}, \mathbf{A}, \boldsymbol{b})$ with an injective function on a multiset, in the form of

$$\boldsymbol{v}_{ij}^0 := \mathsf{hash}\left(C_{ij}, \{\!\{(A_{k,ij}, b_k) \mid k \in [m]\}\!\}\right),$$

such that the constraint elements on entry $(i, j)$ are aggregated in an injective and permutation invariant way, then merged with the objective coefficient $C_{ij}$. Then we execute the following multiset-based 2-FWL:

$$\boldsymbol{v}_{ij}^t := \mathsf{hash}\left(\boldsymbol{v}_{ij}^{t-1}, \{\!\{\{\!\{\boldsymbol{v}_{uj}^{t-1}, \boldsymbol{v}_{iu}^{t-1}\}\!\} \mid u \in [n]\}\!\}\right)$$

Unfortunately, this aggregation is not expressive enough; we can construct a problem:

$$\boldsymbol{C} = \begin{bmatrix} 1 & 0 & 0 \\ 0 & 1 & 0 \\ 0 & 0 & 1 \end{bmatrix}, \quad \boldsymbol{A}_1 = \begin{bmatrix} 1 & 0 & 0 \\ 0 & 1 & 0 \\ 0 & 0 & 0 \end{bmatrix}, \quad \boldsymbol{A}_2 = \begin{bmatrix} 0 & 0 & 0 \\ 0 & 0 & 0 \\ 0 & 0 & 1 \end{bmatrix}, \quad \boldsymbol{b} = \begin{bmatrix} 1 & 1 \end{bmatrix}$$

whose optimal solution has the form $\boldsymbol{X}^* = \begin{bmatrix} 1-\lambda & 0 & 0 \\ 0 & \lambda & 0 \\ 0 & 0 & 1 \end{bmatrix}$, where $\lambda \in [0, 1]$. The solution with minimum Frobenius

norm is $\begin{bmatrix} 0.5 & 0 & 0 \\ 0 & 0.5 & 0 \\ 0 & 0 & 1 \end{bmatrix}$.

However, the encoding

$$\boldsymbol{v}_{22}^0 = \mathsf{hash}\left(1, \{\!\{(1, 1), (0, 1)\}\!\}\right)$$
$$\boldsymbol{v}_{33}^0 = \mathsf{hash}\left(1, \{\!\{(0, 1), (1, 1)\}\!\}\right)$$

are the same. Later, one can easily verify that the converged colors $\boldsymbol{v}_{22}^\infty = \boldsymbol{v}_{33}^\infty$.

The failing of such a combination lies in that the encoding cannot distinguish some structural difference, for example, above, $b_1$ is related with 2 elements in $\boldsymbol{A}_1$, while $b_2$ associates with only one in $\boldsymbol{A}_2$. The encoding with lack of expressivity will falsely encode variables with different structural roles as the same.

**VC-WL then 2-FWL**   Another interesting ablation of Equation (4) is, can we run VC-WL on the V-C bipartite graph until convergence, then run multiset-based 2-FWL until convergence, and obtain the same expressivity as Equation (4)? Therefore, it effectively splits the aggregation of VC-2-FWL. The answer is no. We give a counterexample.

$$\boldsymbol{C} = \mathbf{1}_{5\times 5}$$

$$\boldsymbol{A}_1 = \begin{bmatrix} 1 & 2 & 1 & 1 & 1 \\ 2 & 0 & 0 & 0 & 0 \\ 1 & 0 & 0 & 0 & 0 \\ 1 & 0 & 0 & 0 & 0 \\ 1 & 0 & 0 & 0 & 0 \end{bmatrix}, \quad \boldsymbol{A}_2 = \begin{bmatrix} 0 & 0 & 0 & 0 & 0 \\ 0 & 1 & 1 & 1 & 0 \\ 0 & 1 & 0 & 0 & 0 \\ 0 & 1 & 0 & 0 & 0 \\ 0 & 0 & 0 & 0 & 0 \end{bmatrix}, \quad \boldsymbol{A}_3 = \begin{bmatrix} 0 & 0 & 0 & 0 & 0 \\ 0 & 0 & 0 & 0 & 1 \\ 0 & 0 & 1 & 1 & 0 \\ 0 & 0 & 1 & 0 & 0 \\ 0 & 1 & 0 & 0 & 0 \end{bmatrix}$$

$$\boldsymbol{A}_4 = \begin{bmatrix} 0 & 0 & 0 & 0 & 0 \\ 0 & 0 & 0 & 0 & 0 \\ 0 & 0 & 0 & 0 & 1 \\ 0 & 0 & 0 & 1 & 1 \\ 0 & 0 & 1 & 1 & 0 \end{bmatrix}, \quad \boldsymbol{A}_5 = \begin{bmatrix} 0 & 0 & 0 & 0 & 0 \\ 0 & 0 & 0 & 0 & 0 \\ 0 & 0 & 0 & 0 & 0 \\ 0 & 0 & 0 & 0 & 0 \\ 0 & 0 & 0 & 0 & 1 \end{bmatrix}$$

$$\boldsymbol{b} = \begin{bmatrix} 1 & 1 & 1 & 1 & 1 \end{bmatrix}.$$

(56)

In this example, after the convergence of VC-WL, the converged colors $\boldsymbol{v}_{ij}^\infty$ will be $\begin{bmatrix} a & b & e & e & e \\ b & d & c & c & c \\ e & c & d & c & c \\ e & c & c & d & c \\ e & c & c & c & f \end{bmatrix}$, where $\{a, b, c, \cdots\}$

are from an alphabet to represent colors.

After the convergence of multiset-based 2-FWL, the converged colors will be $\begin{bmatrix} a & b & c & c & d \\ b & e & f & f & g \\ c & f & k & h & i \\ c & f & h & k & i \\ d & g & i & i & j \end{bmatrix}$. The solution is

approximately $\begin{bmatrix} 4.682 & 1.821 & -4.889 & 0.000 & -0.594 \\ 1.821 & 4.327 & -1.664 & 0.000 & -2.060 \\ -4.889 & -1.664 & 5.121 & 0.000 & 0.500 \\ 0.000 & 0.000 & 0.000 & 0.000 & 0.000 \\ -0.594 & -2.060 & 0.500 & 0.000 & 1.000 \end{bmatrix}$. We can see the pair e.g., $X_{13}$ and $X_{14}$ have respectively different values in their solution, yet cannot be distinguished.

The limitation of such a design lies in the message passing cannot distinguish some variables in different constraints, and these variables may have non-equivalent positions in the matrix and play different roles in the matrix.

Therefore, one round of sequential execution of VC-WL and multiset-based 2-FWL does not suffice. Empirically, multiset-based 2-FWL might break the previous VC-WL symmetry. One has to execute both in turn until they both converge. However, this might introduce more steps until convergence.

### E.2. Local VC-2-WL

In addition to our VC-2-WL stemmed from standard 2-WL, we evaluate a stronger variant named $\delta$-VC-2-WL based on the $\delta$-$k$-WL framework of Morris et al. (2020), adapted here for $k = 2$. The color initialization follows Equation (2), and the constraint node update follows Equation (3). But the refinement step of variable nodes is augmented to explicitly incorporate local connectivity:

$$\boldsymbol{v}_{ij}^t := \mathsf{hash}\left(\boldsymbol{v}_{ij}^{t-1}, \{\!\!\{ (\boldsymbol{v}_{uj}^{t-1}, \mathsf{adj}(C_{ui})) \mid u \in [n] \}\!\!\}, \{\!\!\{ (\boldsymbol{v}_{iu}^{t-1}, \mathsf{adj}(C_{uj})) \mid u \in [n] \}\!\!\}, \{\!\!\{ (A_{k,ij}, \boldsymbol{c}_k^{t-1}) \mid k \in N(ij) \}\!\!\}\right)$$

where $\mathsf{adj}(C_{ij})$ is an indicator function representing the sparsity pattern of the cost matrix $C$, i.e., taking the value 1 if $C_{ij} \neq 0$, and 0 otherwise.

In Morris et al. (2020), it is established that $\delta$-$k$-WL is strictly more expressive than standard $k$-WL for graph isomorphism testing. In our SDP context, we prove the same result.

**Proposition E.1.** *The $\delta$-VC-2-WL algorithm strictly refines* VC-2-WL*, denoted as*

$$\delta\text{-VC-2-WL} \sqsubset \text{VC-2-WL}.$$

*Proof.* By definition, $\delta$-VC-2-WL initialization and constraint node update are identical to VC-2-WL, therefore, we focus on variable node update. By observation, the information aggregated by $\delta$-VC-2-WL is a superset of VC-2-WL. Therefore, we immediately have $\delta$-VC-2-WL $\sqsubseteq$ VC-2-WL, because the variables indistinguishable to $\delta$-VC-2-WL must also be the same under VC-2-WL.

To show the strict refinement, we construct a sparse example where the adjacency information is non-trivial. Consider an SDP instance defined by

$$C = \begin{bmatrix} 0 & 1 & 0 & 0 & 1 & 1 \\ 1 & 0 & 0 & 1 & 1 & 0 \\ 0 & 0 & 0 & 1 & 1 & 1 \\ 0 & 1 & 1 & 0 & 0 & 1 \\ 1 & 1 & 1 & 0 & 0 & 0 \\ 1 & 0 & 1 & 1 & 0 & 0 \end{bmatrix}, \quad A_1 = I_6, \quad \boldsymbol{b} = [1]. \tag{57}$$

The $C$ is constructed as the adjacency matrix of a regular graph. Because all the row and column multisets are identical, the VC-2-WL converges to the color configuration $\begin{bmatrix} a & b & c & c & b & b \\ b & a & c & b & b & c \\ c & c & a & b & b & b \\ c & b & b & a & c & b \\ b & b & b & c & a & c \\ b & c & b & b & c & a \end{bmatrix}$ immediately after initialization, where $\{a, b, \cdots\}$

are colors from an alphabet. Thus, the variables $(1, 6)$ and $(2, 5)$ are not distinguished by VC-2-WL.

For the update of $\delta$-VC-2-WL, we have at initialization $\boldsymbol{v}_{16}^0 = \boldsymbol{v}_{25}^0 = b$, while

$$\boldsymbol{v}_{16}^1 := \mathsf{hash}\bigg( \boldsymbol{v}_{16}^0, \{\{(\boldsymbol{v}_{16}^0, \mathsf{adj}(C_{11})), (\boldsymbol{v}_{26}^0, \mathsf{adj}(C_{12})), (\boldsymbol{v}_{36}^0, \mathsf{adj}(C_{13})), (\boldsymbol{v}_{46}^0, \mathsf{adj}(C_{14})), (\boldsymbol{v}_{56}^0, \mathsf{adj}(C_{15}))(\boldsymbol{v}_{66}^0, \mathsf{adj}(C_{16}))\}\},$$

$$\{\{(\boldsymbol{v}_{11}^0, \mathsf{adj}(C_{16})), (\boldsymbol{v}_{12}^0, \mathsf{adj}(C_{26})), (\boldsymbol{v}_{13}^0, \mathsf{adj}(C_{36})), (\boldsymbol{v}_{14}^0, \mathsf{adj}(C_{46})), (\boldsymbol{v}_{15}^0, \mathsf{adj}(C_{56}))(\boldsymbol{v}_{16}^0, \mathsf{adj}(C_{66}))\}\} \bigg)$$

$$= \bigg( b, \{\{(b, 0), (c, 1), (b, 0), (b, 0), (c, 1)(a, 1)\}\}, \{\{(a, 1), (b, 0), (c, 1), (c, 1), (b, 0)(b, 0)\}\} \bigg)$$

and

$$\boldsymbol{v}_{25}^1 := \mathsf{hash}\bigg( \boldsymbol{v}_{25}^0, \{\{(\boldsymbol{v}_{15}^0, \mathsf{adj}(C_{21})), (\boldsymbol{v}_{25}^0, \mathsf{adj}(C_{22})), (\boldsymbol{v}_{35}^0, \mathsf{adj}(C_{23})), (\boldsymbol{v}_{45}^0, \mathsf{adj}(C_{24})), (\boldsymbol{v}_{55}^0, \mathsf{adj}(C_{25}))(\boldsymbol{v}_{65}^0, \mathsf{adj}(C_{26}))\}\},$$

$$\{\{(\boldsymbol{v}_{21}^0, \mathsf{adj}(C_{15})), (\boldsymbol{v}_{22}^0, \mathsf{adj}(C_{25})), (\boldsymbol{v}_{23}^0, \mathsf{adj}(C_{35})), (\boldsymbol{v}_{24}^0, \mathsf{adj}(C_{45})), (\boldsymbol{v}_{25}^0, \mathsf{adj}(C_{55}))(\boldsymbol{v}_{26}^0, \mathsf{adj}(C_{56}))\}\} \bigg)$$

$$= \bigg( b, \{\{(b, 1), (b, 0), (b, 0), (c, 1), (a, 1), (c, 0)\}\}, \{\{(b, 1), (a, 1), (c, 1), (b, 0), (b, 0), (c, 0), \}\} \bigg)$$

We notice that in the first multiset, $\boldsymbol{v}_{25}^1$ has a $(b, 1)$ that $\boldsymbol{v}_{16}^1$ doesn't, so $(1, 6)$ and $(2, 5)$ are distinguished by $\delta$-VC-2-WL.

Therefore, there are variables where $\delta$-VC-2-WL can distinguish while VC-2-WL cannot. That completes the proof that $\delta$-VC-2-WL $\sqsubset$ VC-2-WL. $\qquad\square$

However, we demonstrate that this increased expressivity remains insufficient for approximating SDP solutions.

**Proposition E.2.** *The $\delta$-VC-2-WL fails to represent linear SDPs. That is, there exist instances where the stable colors under $\delta$-VC-2-WL satisfy $\boldsymbol{v}_{ij}^\infty = \boldsymbol{v}_{pq}^\infty$ for distinct indices $(i, j)$ and $(p, q)$, yet the unique optimal solution entries differ $X_{ij}^* \neq X_{pq}^*$.*

*Proof.* We prove by construction with a counterexample.

Consider the one established in the proof of Proposition 2.2. If the cost matrix $C$ in that instance is fully dense, the indicator term adj outputs a constant value and provides no extra information. Consequently, the $\delta$-VC-2-WL update rule degenerates to the standard VC-2-WL rule, showing the same inability to distinguish the optimal solution entries. $\qquad\square$

Similar to the analysis between VC-2-FWL and VC-2-WL, we show that VC-2-FWL and $\delta$-VC-2-WL are incomparable.

**Proposition E.3.** $\delta$-VC-2-WL *and* VC-2-FWL *are incomparable in expressive power, denoted as*

$$\delta\text{-VC-2-WL} \not\equiv \text{VC-2-FWL}.$$

*Proof.* The proof is straightforward with the same logic as Proposition C.8. First, we can reuse the same instance in Proposition C.8, where VC-2-WL can distinguish a pair of variables and VC-2-FWL cannot. Since $\delta$-VC-2-WL strictly refines VC-2-WL, it is also stronger than VC-2-FWL on that example. Second, we reuse the example in Proposition 2.2, where $\delta$-VC-2-WL fails while VC-2-FWL can effectively distinguish all the variables except symmetric ones.

Therefore, $\delta$-VC-2-WL and VC-2-FWL are incomparable in expressive power. $\qquad\square$

Similar to VC-2-MPNN, we can neuralize $\delta$-VC-2-WL into $\delta$-VC-2-MPNN:

$$\boldsymbol{m}_{ij,\text{row}}^t \coloneqq \sum_{u \in [n]} \mathsf{MSG}_{\text{row}}^t \left( \boldsymbol{h}_{iu}^{t-1}, \mathsf{adj}(C_{uj}) \right)$$

$$\boldsymbol{m}_{ij,\text{col}}^t \coloneqq \sum_{u \in [n]} \mathsf{MSG}_{\text{col}}^t \left( \boldsymbol{h}_{uj}^{t-1}, \mathsf{adj}(C_{iu}) \right)$$

$$\boldsymbol{m}_{ij,\text{c}\to\text{v}}^t \coloneqq \sum_{k \in N(ij)} \mathsf{MSG}_{\text{c}\to\text{v}}^t \left( A_{k,ij}, \boldsymbol{h}_k^{t-1} \right)$$

$$\boldsymbol{m}_{k,\text{v}\to\text{c}}^t \coloneqq \sum_{(i,j) \in N(k)} \mathsf{MSG}_{\text{v}\to\text{c}}^t \left( A_{k,ij}, \boldsymbol{h}_{ij}^{t-1} \right)$$

$$\boldsymbol{h}_{ij}^t \coloneqq \mathsf{UPD}_{\text{v}}^t \left( \boldsymbol{h}_{ij}^{t-1}, \boldsymbol{m}_{ij,\text{col}}^t, \boldsymbol{m}_{ij,\text{row}}^t, \boldsymbol{m}_{ij,\text{c}\to\text{v}}^t \right)$$

$$\boldsymbol{h}_k^t \coloneqq \mathsf{UPD}_{\text{c}}^t \left( \boldsymbol{h}_k^{t-1}, \boldsymbol{m}_{k,\text{v}\to\text{c}}^t \right)$$

which, similar to Proposition C.26, is not universal approximating.

### E.3. VC-2-IGN

As an alternative instantiation of the variable update mechanism, we consider the 2-order Invariant Graph Network (2-IGN), based on the theoretical characterization of permutation equivariant layers by Maron et al. (2019b).

The 2-IGN operates on second-order tensors. For a general linear layer mapping $\mathbb{R}^{n \times n} \to \mathbb{R}^{n \times n}$ (feature dimension omitted for simplicity), the space of permutation equivariant operations is spanned by a basis of size $b(2+2) = 15$, where $b(\cdot)$ denotes the Bell number. However, our application to SDPs imposes a strict geometric constraint: both the input feature matrix and the output updated features must lie in the symmetric space $\mathbb{S}^n$. This symmetry constraint reduces the dimension of the equivariant basis from 15 to 9. The reduction occurs through two mechanisms:

1. **Input symmetry**: Since $\boldsymbol{X}_{ij} = \boldsymbol{X}_{ji}$, operations that distinguish rows from columns become redundant, e.g., summing over rows yields the same vector as summing over columns.

2. **Output symmetry**: To guarantee the output is symmetric, the corresponding basis operations must share weights. For instance, broadcasting a vector to rows must be paired with broadcasting to columns.

Formally, let $\mathbf{1} \in \mathbb{R}^n$ be the all-ones vector, $\boldsymbol{I}$ be the identity matrix, $\boldsymbol{d} = \text{diag}(\boldsymbol{X}) \in \mathbb{R}^n$ be the vector of diagonal entries, and $\boldsymbol{s} = \boldsymbol{X}\mathbf{1} \in \mathbb{R}^n$ be the vector of row sums. We identify the following 9 basis operations $\mathcal{B} = \{\mathsf{Op}_1, \dots, \mathsf{Op}_9\}$ that map a symmetric $\boldsymbol{X}$ to a symmetric output:

1. $\mathsf{Op}_1(\boldsymbol{X}) = \boldsymbol{X}$. Identity.

2. $\mathsf{Op}_2(\boldsymbol{X}) = \text{Tr}(\boldsymbol{X})\mathbf{1}\mathbf{1}^\intercal$. Global trace broadcast.

3. $\mathsf{Op}_3(\boldsymbol{X}) = (\mathbf{1}^\intercal \boldsymbol{X} \mathbf{1})\mathbf{1}\mathbf{1}^\intercal$. Global sum broadcast.

4. $\mathsf{Op}_4(\boldsymbol{X}) = \text{diag}(\boldsymbol{d})$. Copy diagonal.

5. $\mathsf{Op}_5(\boldsymbol{X}) = \text{diag}(\boldsymbol{s})$. Row sums to diagonal.

6. $\mathsf{Op}_6(\boldsymbol{X}) = \text{Tr}(\boldsymbol{X})\boldsymbol{I}$. Trace to diagonal.

7. $\mathsf{Op}_7(\boldsymbol{X}) = (\mathbf{1}^\intercal \boldsymbol{X} \mathbf{1})\boldsymbol{I}$. Global sum to diagonal.

8. $\mathsf{Op}_8(\boldsymbol{X}) = \boldsymbol{s}\mathbf{1}^\intercal + \mathbf{1}\boldsymbol{s}^\intercal$. Row/Col sums broadcast.

9. $\mathsf{Op}_9(\boldsymbol{X}) = \boldsymbol{d}\mathbf{1}^\intercal + \mathbf{1}\boldsymbol{d}^\intercal$. Diagonal entries broadcast.

In the context of deep learning, we extend these operations to multi-dimensional inputs $\boldsymbol{X} \in \mathbb{R}^{n \times n \times f}$. For each basis operation $\mathsf{Op}_b \in \mathcal{B}$, we assign a learnable weight matrix $\boldsymbol{W}_b \in \mathbb{R}^{f \times f'}$. The 2-IGN layer update is defined as the aggregation of these transformed bases:

$$\boldsymbol{X}' = \sum_{b=1}^9 \mathsf{Op}_b(\boldsymbol{X})\boldsymbol{W}_b \quad \in \mathbb{R}^{n \times n \times f'}.$$

This formulation ensures that the neural network remains strictly equivariant and preserves the symmetric matrix structure required by the SDP relaxation.

We wrap this up as a neural network IGN and plug it into our V-C bipartite graph message passing, and we have the following form of VC-2-IGN:

$$
\begin{aligned}
\boldsymbol{m}_{ij,\text{IGN}}^t &:= \text{IGN}^t(\boldsymbol{H}^{t-1})_{ij} \\
\boldsymbol{m}_{ij,\text{c}\to\text{v}}^t &:= \sum_{k\in N(ij)} \text{MSG}_{\text{c}\to\text{v}}^t\left(A_{k,ij}, \boldsymbol{h}_k^{t-1}\right) \\
\boldsymbol{m}_{k,\text{v}\to\text{c}}^t &:= \sum_{(i,j)\in N(k)} \text{MSG}_{\text{v}\to\text{c}}^t\left(A_{k,ij}, \boldsymbol{h}_{ij}^{t-1}\right) \\
\boldsymbol{h}_{ij}^t &:= \text{UPD}_{\text{v}}^t\left(\boldsymbol{h}_{ij}^{t-1}, \boldsymbol{m}_{ij,\text{IGN}}^t, \boldsymbol{m}_{ij,\text{c}\to\text{v}}^t\right) \\
\boldsymbol{h}_k^t &:= \text{UPD}_{\text{c}}^t\left(\boldsymbol{h}_k^{t-1}, \boldsymbol{m}_{k,\text{v}\to\text{c}}^t\right)
\end{aligned}
$$

where $\boldsymbol{H}^{t-1} \in \mathbb{R}^{n\times n\times d}$ is the tensor of variable features $\boldsymbol{h}_{ij}^{t-1}$.

**Proposition E.4.** VC-2-IGN *is not a universal approximator for SDPs. Given* $\forall\epsilon > 0$*, there always exists an SDP problem, such that for all* VC-2-IGN*, the error between the predicted solution* $\boldsymbol{X}'$ *and the optimal solution* $\boldsymbol{X}^*$ *satisfies*

$$
\|\boldsymbol{X}^* - \boldsymbol{X}'\|_\infty > \epsilon.
$$

We can easily prove in the same way as Proposition 2.2. VC-2-IGN fails on that example too, because the row and column multisets are identical $\{1, 2, 3, 4, 5, 6\}$, and the diagonal elements are the same. We omit the proof here.

While the 2-IGN architecture is defined via continuous linear operations on tensors, comparing its expressivity directly to the discrete VC-2-WL requires a discrete analogue. To this end, we define the VC-2-IGNWL algorithm. This variant adapts some of the 2-IGN basis operations, specifically the row/column aggregations and diagonal broadcasting, because the other operations do not contribute to expressivity.

$$
\begin{aligned}
\boldsymbol{v}_{ij}^t &:= \text{hash}\Big(\boldsymbol{v}_{ij}^{t-1}, \big\{\!\!\big\{\boldsymbol{v}_{uj}^{t-1} \mid u\in[n]\big\}\!\!\big\}, \big\{\!\!\big\{\boldsymbol{v}_{iu}^{t-1} \mid u\in[n]\big\}\!\!\big\}, \\
&\qquad\qquad \big\{\!\!\big\{(A_{k,ij}, \boldsymbol{c}_k^{t-1}) \mid k\in N(ij)\big\}\!\!\big\}, \boldsymbol{v}_{ii}^t, \boldsymbol{v}_{jj}^t\Big) \\
\boldsymbol{c}_k^t &:= \text{hash}\Big(\boldsymbol{c}_k^{t-1}, \big\{\!\!\big\{(A_{k,ij}, \boldsymbol{v}_{ij}^{t-1}) \mid (i,j)\in N(k)\big\}\!\!\big\}\Big).
\end{aligned}
\tag{58}
$$

Notably, we replace the sum over rows and columns with hash of the multisets $\big\{\!\!\big\{\boldsymbol{v}_{uj}^{t-1} \mid u\in[n]\big\}\!\!\big\}, \big\{\!\!\big\{\boldsymbol{v}_{iu}^{t-1} \mid u\in[n]\big\}\!\!\big\}$, which boosts the expressivity because of the injectivity of hash function. The constraint update is the same as Equation (3).

We now establish the expressivity hierarchy between this variant and the VC-2-WL.

**Proposition E.5.** VC-*2*-WL *has expressivity power equivalent to* VC-*2*-IGNWL*, denoted as*

$$
\text{VC-}2\text{-WL} \equiv \text{VC-}2\text{-IGNWL}.
$$

*Proof.* First, we show VC-2-IGNWL $\sqsubseteq$ VC-2-WL. For $t = 0$, since their initialization is the same, VC-2-IGNWL trivially refines VC-2-WL. For any $t > 0$, if the coloring of VC-2-IGNWL cannot distinguish 2 variable, then neither can VC-2-WL, because the color information used by VC-2-IGNWL includes that used by the VC-2-WL.

Then, we show VC-2-WL $\sqsubseteq$ VC-2-IGNWL. For $t = 0$, since their initialization is the same, it trivially holds. For $t > 0$, if

the color refinement under VC-2-WL cannot distinguish 2 variable $(i, j)$ and $(p, q)$, we have:

$$\boldsymbol{v}_{ij}^t = \boldsymbol{v}_{pq}^t$$

$$\implies \mathsf{hash}\Big(\boldsymbol{v}_{ij}^{t-1}, \{\!\{\boldsymbol{v}_{uj}^{t-1} \mid u \in [n]\}\!\}, \{\!\{\boldsymbol{v}_{iu}^{t-1} \mid u \in [n]\}\!\},$$

$$\{\!\{(A_{k,ij}, \boldsymbol{c}_k^{t-1}) \mid k \in N(ij)\}\!\}\Big)$$

$$= \mathsf{hash}\Big(\boldsymbol{v}_{pq}^{t-1}, \{\!\{\boldsymbol{v}_{uq}^{t-1} \mid u \in [n]\}\!\}, \{\!\{\boldsymbol{v}_{pu}^{t-1} \mid u \in [n]\}\!\},$$

$$\{\!\{(A_{k,pq}, \boldsymbol{c}_k^{t-1}) \mid k \in N(pq)\}\!\}\Big)$$

$$\implies \boldsymbol{v}_{ij}^{t-1} = \boldsymbol{v}_{pq}^{t-1}$$

$$\wedge \{\!\{\boldsymbol{v}_{uj}^{t-1} \mid u \in [n]\}\!\} = \{\!\{\boldsymbol{v}_{uq}^{t-1} \mid u \in [n]\}\!\}$$

$$\wedge \{\!\{\boldsymbol{v}_{iu}^{t-1} \mid u \in [n]\}\!\} = \{\!\{\boldsymbol{v}_{pu}^{t-1} \mid u \in [n]\}\!\}$$

$$\wedge \{\!\{(A_{k,ij}, \boldsymbol{c}_k^{t-1}) \mid k \in N(ij)\}\!\} = \{\!\{(A_{k,pq}, \boldsymbol{c}_k^{t-1}) \mid k \in N(pq)\}\!\}$$

Recall in Equation (2) that diagonal elements are specially labeled, they receive unique labels that distinguish them from non-diagonal elements. That means the diagonal elements within e.g. $\{\!\{\boldsymbol{v}_{uj}^{t-1} \mid u \in [n]\}\!\}$ and $\{\!\{\boldsymbol{v}_{uq}^{t-1} \mid u \in [n]\}\!\}$ are unique, and they must be the same as well. So we can continue the implication above:

$$\implies \boldsymbol{v}_{ij}^{t-1} = \boldsymbol{v}_{pq}^{t-1}$$

$$\wedge \{\!\{\boldsymbol{v}_{uj}^{t-1} \mid u \in [n]\}\!\} = \{\!\{\boldsymbol{v}_{uq}^{t-1} \mid u \in [n]\}\!\} \wedge \boldsymbol{v}_{jj}^{t-1} = \boldsymbol{v}_{qq}^{t-1}$$

$$\wedge \{\!\{\boldsymbol{v}_{iu}^{t-1} \mid u \in [n]\}\!\} = \{\!\{\boldsymbol{v}_{pu}^{t-1} \mid u \in [n]\}\!\} \wedge \boldsymbol{v}_{ii}^{t-1} = \boldsymbol{v}_{pp}^{t-1}$$

$$\wedge \{\!\{(A_{k,ij}, \boldsymbol{c}_k^{t-1}) \mid k \in N(ij)\}\!\} = \{\!\{(A_{k,pq}, \boldsymbol{c}_k^{t-1}) \mid k \in N(pq)\}\!\}$$

$$\implies \mathsf{hash}\Big(\boldsymbol{v}_{ij}^{t-1}, \{\!\{\boldsymbol{v}_{uj}^{t-1} \mid u \in [n]\}\!\}, \{\!\{\boldsymbol{v}_{iu}^{t-1} \mid u \in [n]\}\!\},$$

$$\{\!\{(A_{k,ij}, \boldsymbol{c}_k^{t-1}) \mid k \in N(ij)\}\!\}, \boldsymbol{v}_{ii}^{t-1}, \boldsymbol{v}_{jj}^{t-1}\Big)$$

$$= \mathsf{hash}\Big(\boldsymbol{v}_{pq}^{t-1}, \{\!\{\boldsymbol{v}_{uq}^{t-1} \mid u \in [n]\}\!\}, \{\!\{\boldsymbol{v}_{pu}^{t-1} \mid u \in [n]\}\!\},$$

$$\{\!\{(A_{k,pq}, \boldsymbol{c}_k^{t-1}) \mid k \in N(pq)\}\!\}, \boldsymbol{v}_{pp}^{t-1}, \boldsymbol{v}_{qq}^{t-1}\Big)$$

the last step corresponds to the update of VC-2-IGNWL, showing VC-2-IGNWL also cannot distinguish them.

Therefore, VC-2-WL and VC-2-IGNWL have the same expressive power. $\qquad\square$

Following Proposition C.8, we also conclude that VC-2-IGNWL and VC-2-FWL are incomparable in expressive power.

### E.4. Edge Transformer

We adapt the Edge Transformer (ET) module introduced by Müller et al. (2024). The ET architecture is structurally analogous to standard 2-FWL. While the original ET operates on graph node pairs, our framework operates on variables in a matrix, sharing the same pairwise form. Furthermore, the ET has been rigorously proven to possess the expressive power of the 2-FWL algorithm. These properties make it a perfect candidate to compare with our VC-2-FMPNN.

The ET operates on a second-order tensor $\boldsymbol{X} \in \mathbb{R}^{n \times n \times d}$, where $d$ is the embedding dimension and $\boldsymbol{X}_{ij}$ denotes the feature vector of the variable indexed at $(i, j)$. Formally, the layer update is defined as:

$$\boldsymbol{X}_{ij}^t \coloneqq \mathsf{FFN}^t\big(\boldsymbol{X}_{ij}^{t-1} + \mathsf{TriAttn}^t\big(\mathsf{LN}^t\big(\boldsymbol{X}_{ij}^{t-1}\big)\big)\big),$$

where FFN is a feed-forward neural network, LN denotes layer normalization (Ba et al., 2016) and TriAttn is defined as

$$\mathsf{TriAttn}(\boldsymbol{X}_{ij}) \coloneqq \sum_{l=1}^n \alpha_{ilj} \boldsymbol{V}_{ilj}. \tag{59}$$

This operation computes a linear combination of a *value vector* $\boldsymbol{V}_{ilj}$, with weights given by scalar attention scores $\alpha ilj$, over the summation index $l$. The attention scores are computed via a scaled dot-product mechanism:

$$\alpha_{ilj} := \underset{l \in [n]}{\mathsf{softmax}} \left( \frac{1}{\sqrt{d}} \boldsymbol{X}_{il} \boldsymbol{W}^Q \left( \boldsymbol{X}_{lj} \boldsymbol{W}^K \right)^\top \right) \in \mathbb{R} \tag{60}$$

which captures the relevance of the triplet $(i, l, j)$, and the value vector is computed via the *value fusion* of the intermediate representations

$$\boldsymbol{V}_{ilj} := \boldsymbol{X}_{il} \boldsymbol{W}^{V_1} \odot \boldsymbol{X}_{lj} \boldsymbol{W}^{V_2}, \tag{61}$$

where $\odot$ denotes element-wise multiplication, and $\boldsymbol{W}^Q, \boldsymbol{W}^K, \boldsymbol{W}^{V_1}, \boldsymbol{W}^{V_2} \in \mathbb{R}^{d \times d}$ are learnable projection matrices.

Integrating this module into our framework yields the VC-ET architecture. The initialization follows Equation (5). For the update step, we substitute the variable-to-variable message passing with the ET layer. Let $\boldsymbol{H}^{t-1} \in \mathbb{R}^{n \times n \times d}$ denote the tensor of variable features $\boldsymbol{h}_{ij}^{t-1}$. The update rules are:

$$
\begin{aligned}
\boldsymbol{m}_{ij,\mathrm{ET}}^t &:= \mathsf{ET}^t \left( \boldsymbol{H}^{t-1} \right)_{ij} \\
\boldsymbol{m}_{ij,\mathrm{c \to v}}^t &:= \sum_{k \in N(ij)} \mathsf{MSG}_{\mathrm{c \to v}}^t \left( A_{k,ij}, \boldsymbol{h}_k^{t-1} \right) \\
\boldsymbol{m}_{k,\mathrm{v \to c}}^t &:= \sum_{(i,j) \in N(k)} \mathsf{MSG}_{\mathrm{v \to c}}^t \left( A_{k,ij}, \boldsymbol{h}_{ij}^{t-1} \right) \\
\boldsymbol{h}_{ij}^t &:= \mathsf{UPD}_{\mathrm{v}}^t \left( \boldsymbol{h}_{ij}^{t-1}, \boldsymbol{m}_{ij,\mathrm{ET}}^t, \boldsymbol{m}_{ij,\mathrm{c \to v}}^t \right) \\
\boldsymbol{h}_k^t &:= \mathsf{UPD}_{\mathrm{c}}^t \left( \boldsymbol{h}_k^{t-1}, \boldsymbol{m}_{k,\mathrm{v \to c}}^t \right).
\end{aligned}
\tag{62}
$$

Thus, VC-ET effectively replaces the explicit 2-FWL-based aggregation with the learnable, triangular attention of the Edge Transformer.

## F. More experiment details

**Neural architecture and hyperparameters** For the VC-MPNN, the initialization follows Equation (5), implemented via two 2-layer MLPs with ReLU activation. The message passing MSG and update UPD functions, defined in Equation (27), are parameterized by a single linear layer followed by ReLU activation and GraphNorm (Cai et al., 2021). The final prediction head is a 3-layer MLP with ReLU activation, mapping the latent features to a scalar prediction per variable.

Similarly, for VC-2-MPNN and VC-2-FMPNN, all intermediate neural functions are implemented as single-layer MLPs. For VC-2-FMPNN, the aggregation function for $\mathsf{MAP}^t \left( \boldsymbol{h}_{uj}^{t-1} \right)$ and $\mathsf{MAP}^t \left( \boldsymbol{h}_{iu}^{t-1} \right)$ are be replaced by multiplication, therefore it can be implemented in a way similar to Maron et al. (2019a). The $\delta$-VC-2-MPNN differs from the VC-2-MPNN architecture, in that a matrix multiplication is performed between the embedding matrix and a matrix comprised by adj indicators, effectively aggregating the neighborhood indicator. For VC-2-IGN, we employ the basis formulation described in Appendix E.3, parameterizing each basis operation with a single linear layer. For VC-ET, the feed-forward network FFN consists of a single linear layer, and the TriAttn module of 1 attention head is implemented with learnable weight matrices. For all neural architectures, we insert a LayerNorm (Ba et al., 2016) between the GNN convolutional layers.

To ensure fair comparison, we maintain consistent hidden dimensions and layer counts across models for each dataset. We fix the hidden dimension to 96 for all architectures and all datasets. Regarding number of GNN layers, we employ 6 layers for the Max Clique, MIS, and LMI datasets; and 10 layers for the max-cut (both ER and regular graphs, and SDPLIB), Vertex Cover, and max 2-SAT datasets.

**Results on LMI problems** Since the optimal objective value for LMI problems is 0, we report the predicted objective values directly in Table 6. As shown, performance varies minimally across different neural architectures. This is likely because the LMI instance coefficients are randomly generated, lacking the complex structural symmetries found in graph-based problems that typically expose the limitations of weaker methods.

**Compare to CO solution** To directly illustrate the comparison regarding how well our predictions translate to discrete CO solutions, we evaluated relative objective gap for Max-Cut, MIS and Vertex Cover.

*Table 6.* Loss and objective of LMI problems on test set, repeated with 5 seeds. The best results across all are highlighted.

| Model | Test loss | Obj. | Proj. obj. |
|---|---|---|---|
| VC-MPNN | $4.520e-5_{\pm 0.000}$ | $-0.091_{\pm 0.275}$ | $0.124_{\pm 0.126}$ |
| VC-2-MPNN | $4.516e-5_{\pm 0.000}$ | $0.031_{\pm 0.044}$ | $0.074_{\pm 0.028}$ |
| $\delta$-VC-2-MPNN | $4.563e-5_{\pm 0.000}$ | $-0.163_{\pm 0.066}$ | $\mathbf{0.021_{\pm 0.001}}$ |
| VC-2-IGN | $4.506e-5_{\pm 0.000}$ | $\mathbf{0.017_{\pm 0.103}}$ | $0.090_{\pm 0.053}$ |
| VC-2-FMPNN | $\mathbf{4.305e-5_{\pm 0.000}}$ | $-0.063_{\pm 0.042}$ | $0.066_{\pm 0.012}$ |
| VC-ET | $4.357e-5_{\pm 0.000}$ | $-0.036_{\pm 0.000}$ | $0.122_{\pm 0.097}$ |

*Table 7.* The relative objective gap (%) between our predicted SDP solution and CO solution.

| Candidate | Max-Cut | MIS | Vertex Cover |
|---|---|---|---|
| SCS - CO | $4.547_{\pm 0.858}$ | $199.229_{\pm 2.465}$ | $44.622_{\pm 3.699}$ |
| VC-2-FMPNN - CO | $4.610_{\pm 0.855}$ | $199.358_{\pm 3.426}$ | $44.955_{\pm 3.897}$ |

As Table 7 shows, the downstream CO gaps produced by our neural solver perfectly mirror the inherent gaps of the exact SDP relaxations. For Max-Cut, the SDP solution of our GNN performs almost identically to exact SDP solver outputs, with a negligible 0.06% difference. The observations have validated our core claim: the neural approach successfully learns the underlying continuous SDP geometry, rather than directly learning the discrete integer solutions.

However, the neural solver's high quality solution and fast inference time enable massive potential for CO solvers. Modern exact branch-and-bound methods for Max-Cut (Hrga & Povh, 2021) require solving thousands of intermediate SDPs, which is the primary computational bottleneck. Replacing these expensive exact solvers with our cheap neural proxy to guide branching or pruning decisions, similar to the seminal work Gasse et al. (2019) for LPs, is a highly promising future direction.

**Compare to OptGNN** We adapted OptGNN (Yau et al., 2024) to our evaluation framework for comparison in relative objective gap on Max-Cut SDP, and the other datasets are incompatible due to formulation differences or unavailable source code.

*Table 8.* Comparison to OptGNN on Max-Cut problems.

| Model | Target | Max-Cut ER | Max-Cut Regular |
|---|---|---|---|
| OptGNN | obj. gap | $1.304_{\pm 0.014}$ | $2.030_{\pm 0.019}$ |
| OptGNN | cons. vio. | 0.0 | 0.0 |
| VC-2-FMPNN | obj. gap | $0.111_{\pm 0.017}$ | $0.092_{\pm 0.004}$ |
| VC-2-FMPNN | cons. vio. | $0.003_{\pm 0.001}$ | $0.003_{\pm 0.001}$ |

OptGNN achieves zero constraint violations because it optimizes a low-rank latent embedding where the Max-Cut constraint is trivially satisfied via row-normalization. While we could also adapt VC-2-FMPNN to predict the latent to achieve feasibility, we stick to our formulation in the paper to ensure consistency with general SDP. As shown in Table 8, our architecture achieves an obj. gap one order of magnitude tighter. However, we still highlight our different scopes: OptGNN targets discrete CO problems using SDP relaxations as a tool, whereas our framework serves as a continuous surrogate for solving the general linear SDP.

**More SDPLIB results** We exhibit the statistics of SDPLIB datasets in Table 9.

We show the complete approximation performance of VC-MPNN, VC-2-MPNN and VC-2-FMPNN on SDPLIB, in Table 10.

**Constraint violation** As demonstrated in Table 11, our VC-2-FMPNN achieves remarkably low constraint violation residues consistently across all tasks, on the order of $10^{-3}$ to $10^{-5}$. While not perfectly feasible, these violations are very low and highly competitive for practical applications.

*Table 9.* Statistics of selected SDPLIB problems.

| Name | #vars | #cons. |
|------|-------|--------|
| MCP100 | 10000 | 100 |
| MCP124-{1-4} | 15376 | 124 |
| MCP250-{1-4} | 62500 | 250 |
| MCP500-{1-4} | 250000 | 500 |

*Table 10.* Loss and objective gap (%) on SDPLIB instances, repeated with 5 seeds. The best results are highlighted.

| Name | Loss | | | Obj gap (%) | | | Proj. obj gap (%) | | |
|------|---------|-----------|------------|---------|-----------|------------|---------|-----------|------------|
| | VC-MPNN | VC-2-MPNN | VC-2-FMPNN | VC-MPNN | VC-2-MPNN | VC-2-FMPNN | VC-MPNN | VC-2-MPNN | VC-2-FMPNN |
| MCP100 | $0.203\pm_{0.000}$ | $0.199\pm_{0.004}$ | $\mathbf{1.974e\text{-}4}\pm_{\mathbf{0.000}}$ | $0.132\pm_{0.083}$ | $0.145\pm_{0.052}$ | $\mathbf{0.002}\pm_{\mathbf{0.000}}$ | $15.336\pm_{0.030}$ | $16.332\pm_{0.438}$ | $\mathbf{1.839}\pm_{\mathbf{0.050}}$ |
| MCP124-1 | $0.225\pm_{0.000}$ | $0.224\pm_{0.000}$ | $\mathbf{1.546e\text{-}4}\pm_{\mathbf{0.000}}$ | $0.298\pm_{0.199}$ | $0.377\pm_{0.023}$ | $\mathbf{0.039}\pm_{\mathbf{0.001}}$ | $12.255\pm_{0.157}$ | $12.770\pm_{0.162}$ | $\mathbf{1.802}\pm_{\mathbf{0.605}}$ |
| MCP124-2 | $0.220\pm_{0.000}$ | $0.218\pm_{0.002}$ | $\mathbf{2.303e\text{-}4}\pm_{\mathbf{0.000}}$ | $0.166\pm_{0.105}$ | $0.267\pm_{0.023}$ | $\mathbf{0.047}\pm_{\mathbf{0.005}}$ | $15.184\pm_{0.213}$ | $15.998\pm_{0.324}$ | $\mathbf{1.976}\pm_{\mathbf{0.785}}$ |
| MCP124-3 | $0.215\pm_{0.000}$ | $0.214\pm_{0.002}$ | $\mathbf{3.076e\text{-}4}\pm_{\mathbf{0.000}}$ | $\mathbf{0.019}\pm_{\mathbf{0.012}}$ | $0.325\pm_{0.176}$ | $0.053\pm_{0.017}$ | $17.424\pm_{0.094}$ | $17.959\pm_{0.542}$ | $\mathbf{2.092}\pm_{\mathbf{0.452}}$ |
| MCP124-4 | $0.270\pm_{0.000}$ | $0.272\pm_{0.007}$ | $\mathbf{3.374e\text{-}4}\pm_{\mathbf{0.000}}$ | $0.139\pm_{0.067}$ | $0.174\pm_{0.012}$ | $\mathbf{0.069}\pm_{\mathbf{0.005}}$ | $20.246\pm_{0.013}$ | $20.966\pm_{0.828}$ | $\mathbf{2.967}\pm_{\mathbf{0.881}}$ |
| MCP250-1 | $0.215\pm_{0.000}$ | $0.215\pm_{0.000}$ | $\mathbf{3.034e\text{-}4}\pm_{\mathbf{0.000}}$ | $0.284\pm_{0.022}$ | $0.232\pm_{0.032}$ | $\mathbf{0.018}\pm_{\mathbf{0.001}}$ | $13.171\pm_{0.188}$ | $12.794\pm_{0.045}$ | $\mathbf{2.393}\pm_{\mathbf{1.103}}$ |
| MCP250-2 | $0.158\pm_{0.015}$ | $0.160\pm_{0.007}$ | $\mathbf{4.176e\text{-}4}\pm_{\mathbf{0.000}}$ | $0.113\pm_{0.017}$ | $0.274\pm_{0.134}$ | $\mathbf{0.013}\pm_{\mathbf{0.008}}$ | $15.320\pm_{0.129}$ | $15.269\pm_{0.533}$ | $\mathbf{2.525}\pm_{\mathbf{0.778}}$ |
| MCP250-3 | $0.155\pm_{0.000}$ | $0.156\pm_{0.000}$ | $\mathbf{6.266e\text{-}4}\pm_{\mathbf{0.000}}$ | $\mathbf{0.111}\pm_{\mathbf{0.049}}$ | $0.523\pm_{0.176}$ | $0.391\pm_{0.076}$ | $18.913\pm_{0.087}$ | $17.242\pm_{0.129}$ | $\mathbf{2.892}\pm_{\mathbf{0.948}}$ |
| MCP250-4 | $0.182\pm_{0.013}$ | $0.186\pm_{0.004}$ | $\mathbf{5.892e\text{-}4}\pm_{\mathbf{0.000}}$ | $0.137\pm_{0.017}$ | $0.166\pm_{0.088}$ | $\mathbf{0.041}\pm_{\mathbf{0.008}}$ | $20.134\pm_{0.144}$ | $20.405\pm_{0.592}$ | $\mathbf{3.428}\pm_{\mathbf{1.321}}$ |
| MCP500-1 | $0.118\pm_{0.000}$ | $0.118\pm_{0.001}$ | $\mathbf{4.536e\text{-}4}\pm_{\mathbf{0.000}}$ | $0.310\pm_{0.022}$ | $\mathbf{0.237}\pm_{\mathbf{0.020}}$ | $0.247\pm_{0.011}$ | $13.054\pm_{0.176}$ | $12.842\pm_{0.175}$ | $\mathbf{3.265}\pm_{\mathbf{0.912}}$ |
| MCP500-2 | $0.123\pm_{0.000}$ | $0.124\pm_{0.001}$ | $\mathbf{4.948e\text{-}4}\pm_{\mathbf{0.000}}$ | $\mathbf{0.337}\pm_{\mathbf{0.103}}$ | $0.431\pm_{0.027}$ | $0.345\pm_{0.117}$ | $16.303\pm_{0.121}$ | $16.307\pm_{0.671}$ | $\mathbf{3.240}\pm_{\mathbf{0.431}}$ |
| MCP500-3 | $0.121\pm_{0.000}$ | $0.122\pm_{0.000}$ | $\mathbf{6.745e\text{-}4}\pm_{\mathbf{0.000}}$ | $\mathbf{0.060}\pm_{\mathbf{0.059}}$ | $0.157\pm_{0.011}$ | $0.322\pm_{0.034}$ | $18.428\pm_{0.114}$ | $18.136\pm_{0.090}$ | $\mathbf{4.143}\pm_{\mathbf{1.323}}$ |
| MCP500-4 | $0.126\pm_{0.000}$ | $0.130\pm_{0.003}$ | $\mathbf{9.761e\text{-}4}\pm_{\mathbf{0.000}}$ | $0.437\pm_{0.032}$ | $\mathbf{0.074}\pm_{\mathbf{0.058}}$ | $0.407\pm_{0.010}$ | $21.048\pm_{0.154}$ | $20.610\pm_{0.051}$ | $\mathbf{4.505}\pm_{\mathbf{1.201}}$ |

*Table 11.* Mean absolute residuals on constraints of different CO problems on test set, repeated with 5 seeds.

| Candidate | Target | Problems | | | | | |
|-----------|--------|----------|---------------|------------|-----|--------------|-----------|
| | | Max-Cut | Max-Cut (reg) | Max-Clique | MIS | Vertex Cover | Max 2-SAT |
| SCS | Time (sec.) | $0.386\pm_{0.028}$ | $0.385\pm_{0.190}$ | $0.642\pm_{0.308}$ | $0.672\pm_{0.340}$ | $0.089\pm_{0.004}$ | $0.266\pm_{0.081}$ |
| VC-2-FMPNN | Time (sec.) | $0.010\pm_{0.000}$ | $0.010\pm_{0.000}$ | $0.019\pm_{0.001}$ | $0.019\pm_{0.001}$ | $0.019\pm_{0.001}$ | $0.011\pm_{0.001}$ |
| | Proj. Obj. gap (%) | $0.111\pm_{0.017}$ | $0.092\pm_{0.004}$ | $0.436\pm_{0.035}$ | $0.411\pm_{0.025}$ | $0.469\pm_{0.051}$ | $0.862\pm_{0.211}$ |
| | Proj. cons. vio | $0.003\pm_{0.001}$ | $0.003\pm_{0.001}$ | $3.082e\text{-}5\pm_{0.000}$ | $3.067e\text{-}5\pm_{0.000}$ | $0.009\pm_{0.001}$ | $0.008\pm_{0.001}$ |

**SDPLR quality**  Burer-Monteiro method is not guaranteed to solve the SDP exactly. In Table 12, SDPLR yields near-zero objective gaps compared to SCS, except for 0.02% on Max-Clique/MIS, and almost zero constraint violation up to numerical precisions. While it is highly competitive on Max-Cut problems, it fails to converge on Vertex Cover, and scales poorly in runtime on MIS.

*Table 12.* Mean absolute residuals on constraints of different CO problems on test set, repeated with 5 seeds.

| Candidate | Target | Problems | | | | | |
|-----------|--------|----------|---------------|------------|-----|--------------|-----------|
| | | Max-Cut | Max-Cut (reg) | Max-Clique | MIS | Vertex Cover | Max 2-SAT |
| SCS | Time (sec.) | $0.386\pm_{0.028}$ | $0.385\pm_{0.190}$ | $0.642\pm_{0.308}$ | $0.672\pm_{0.340}$ | $0.089\pm_{0.004}$ | $0.266\pm_{0.081}$ |
| SDPLR | Time (sec.) | $0.195\pm_{0.041}$ | $0.235\pm_{0.052}$ | $1.803\pm_{0.184}$ | $3.301\pm_{0.285}$ | Not converging | $0.391\pm_{0.145}$ |
| | Proj. Obj. gap (%) | $3.672e\text{-}6\pm_{0.000}$ | $4.434e\text{-}6\pm_{0.000}$ | $0.020\pm_{0.001}$ | $0.022\pm_{0.002}$ | – | $2.341e\text{-}5\pm_{0.000}$ |
| | Proj. cons. vio | $1.059e\text{-}6\pm_{0.000}$ | $1.193e\text{-}6\pm_{0.000}$ | $4.269e\text{-}7\pm_{0.000}$ | $2.696e\text{-}7\pm_{0.000}$ | – | $8.223e\text{-}7\pm_{0.000}$ |

**Effect of depth**  To directly observe the effect of depth of VC-2-FMPNN, we conducted a sensitivity analysis on smaller Max-Cut SDPs (ER graphs, 50 nodes, $p = 0.1$) with varying network depths.

Remarkably, we observe no performance drop even at 30 layers. Performance consistently improves with depth, though the gains are marginal after 10 layers, which motivated our default setting with 10 layers in the paper.

*Table 13.* The loss, relative objective gap and constraint violation on test set, with different number of layers.

| Layers | Loss | Obj. gap | Proj. obj. gap | Cons. vio. | Proj. cons. vio. |
|---|---|---|---|---|---|
| 5 | 3.103e-3±0.000 | 0.276±0.001 | 0.254±0.001 | 0.004±0.001 | 0.039±0.002 |
| 10 | 1.088e-4±0.000 | 0.076±0.005 | 0.073±0.005 | 0.001±0.000 | 0.004±0.001 |
| 15 | 3.448e-5±0.000 | 0.054±0.002 | 0.077±0.005 | 0.001±0.000 | 0.002±0.001 |
| 20 | 2.594e-5±0.000 | 0.049±0.005 | 0.051+0.005 | 0.001±0.000 | 0.002±0.000 |
| 30 | 1.141e-5±0.000 | 0.047±0.005 | 0.054±0.017 | 0.001±0.000 | 0.001±0.000 |

