# OpenReview forum: "On the Expressive Power of GNNs to Solve Linear SDPs"
_ICML.cc/2026/Conference — ICML 2026 regular_

### Official Review · Reviewer_bC7w · 2026-03-11

**Soundness:** 3
**Presentation:** 3
**Significance:** 3
**Originality:** 4
**Overall Recommendation:** 5
**Confidence:** 4

**Summary:**

The paper investigates the expressive power required by GNNs to act as computational surrogates for solving linear Semidefinite Programs (SDPs). The authors establish negative results showing that standard variable-constraint message passing (VC-WL) and higher-order 2-WL architectures (VC-2-WL) fail to uniquely represent optimal SDP solutions. To resolve this, they prove that an architecture equivalent to the 2-dimensional folklore Weisfeiler-Leman test (VC-2-FWL) is sufficiently expressive to emulate the updates of the first-order PDHG solver. Empirically, the proposed VC-2-FMPNN architecture achieves significantly lower objective gaps on synthetic combinatorial relaxations and SDPLIB instances compared to theoretically weaker baselines, and successfully accelerates the SCS solver by up to 80% via warm-starting.

**Compliance With Llm Reviewing Policy:**

Affirmed.

**Final Justification:**

In my view, the paper's exploration of the "expressive power of GNNs for SDPs" is already sufficiently innovative and forms a solid standalone contribution. The issues regarding generalization and oversmoothing, while important, are essentially "icing on the cake" for a paper focused on establishing fundamental expressivity.

The explanations and additional empirical data provided in the rebuttal regarding these two aspects, while perhaps not completely resolving the issues theoretically, are more than sufficient to alleviate my practical concerns. They do not detract from the main merits of the paper.

Overall, this is a strong paper with a clear and novel contribution. I am satisfied with the rebuttal and maintain my recommendation for acceptance.

**Key Questions For Authors:**

1. The paper proves expressivity for a given graph size, but GNNs in L2O are most valuable when they can be trained on small problems and deployed on larger ones. Have you conducted experiments where the model trained on $n=50$ nodes is tested on $n=500$ nodes?
2. As higher-order GNNs are typically more challenging to train, what specific techniques or strategies did you employ to ensure the stability of the training process and the consistency of the results?
3. There is a typo on line 2146:  'omite' should be corrected to 'omit'.

**Limitations:**

Yes

**Strengths And Weaknesses:**

**Strengths:**
1. Theoretical rigor: Instead of simply testing a model and reporting performance, the authors establish a clear theoretical hierarchy based on the WL framework. By formally proving the expressive power of VC-WL, VC-2-WL, VC-2-FWL, and VC-2-FWL+, the paper maps out exactly where standard GNNs fail and where higher-order GNNs become necessary for SDPs.
2. Connection to classical optimization algorithms: The paper rigorously proves that the stable coloring of VC-2-FWL refines the iterations of the PDHG algorithm.
3. Empirical results: The practical implications are clear and well-supported by empirical data.

**Weaknesses:**
1. Lack of generalization guarantees: While this paper focuses primarily on the expressive power of the model, it lacks a thorough discussion on generalization capability. There is no formal proof ensuring that a model trained on small SDP instances will maintain numerical stability or optimality when applied to significantly larger instances. Moreover, L2O models are often sensitive to the distribution of problem instances. The paper does not theoretically address how the "stable coloring" adapts to structural shifts in the constraint matrices between training and testing.
2. Over-smoothing" in high-order GNNs: Higher-order GNNs are notoriously prone to over-smoothing, and gradient instability. The paper provides no sensitivity analysis on how many layers or what hidden dimensions are required to maintain the 2-FWL power without collapsing into numerical noise.

---

> ### Author Rebuttal · Authors · 2026-03-27
>
> We thank the reviewer for recognizing the theoretical and empirical significance of our work. We address the concerns below:
>
> # 1. Generalization
> While we did not provide generalization bounds, we argue that establishing the exact expressive power required for a problem is a fundamental prerequisite for such analysis, which is exactly the scope of this paper.
> The reviewer’s concern about stable coloring sheds critical light on generalization. In fact, __if two elements share the same stable coloring, they are guaranteed to yield the same solution even across different SDP instances__. To prove this, one can concatenate them into a larger SDP of two block diagonals, and reuse Theorem 2.3. This provides a strong structural foundation for generalization, making formal bounds on generalization a highly promising direction for future work.
>
> # 2. Oversmoothing
>
> To directly observe whether our architecture suffers from oversmoothing, we conducted a sensitivity analysis on smaller Max-Cut SDPs (ER graphs, 50 nodes, $p=0.1$) across varying network depths:
> | Layers | Loss | Obj. gap    | Proj. obj. gap |  Cons. vio.    | Proj. cons. vio. |
> | --- | --- |-------------|----------------|-------------|----------------|
> | 5 | 3.103e-3±0.000 | 0.276±0.001 | 0.254±0.001    | 0.004±0.001 | 0.039±0.002    |
> | 10 | 1.088e-4±0.000 | 0.076±0.005 | 0.073±0.005    | 0.001±0.000 | 0.004±0.001    |
> | 15 | 3.448e-5±0.000 | 0.054±0.002 | 0.077±0.005    | 0.001±0.000 | 0.002±0.001    |
> | 20 | 2.594e-5±0.000 | 0.049±0.005 | 0.051+0.005    | 0.001±0.000 | 0.002±0.000    |
> | 30 | 1.141e-5±0.000 | 0.047±0.005 | 0.054±0.017    | 0.001±0.000 | 0.001±0.000    |
>
> Remarkably, we observe no oversmoothing or gradient collapse even at 30 layers. Performance consistently improves with depth, though the gains are marginal after 10 layers, which motivated our default setting in the paper. Regarding Q2, to ensure training stability, we apply LayerNorm between the VC-2-FMPNN layers, which effectively controls the variance of the higher-order tensor representations.
>
> # 3. Size generalization
> Generalizing from small-scale supervised targets to massive, unseen graphs is notoriously difficult. As shown below, our supervised VC-2-FMPNN struggles when scaling from Max-Cut of 100 to 500 nodes:
>
> | Size    | Loss           | Proj. obj. gap | Proj. cons. vio. | Time (sec.) |
> |---|---|---|---|---|
> | 100   | 5.140e-5±0.000 | 0.092±0.004  | 0.003±0.001    | 0.014±0.001 |
> | 200   | 0.003±0.000  | 9.255±1.065    | 0.311±0.039    | 0.016±0.001 |
> | 300   | 0.007±0.001  | 17.193±0.885   | 0.526±0.067    | 0.034±0.001 |
> | 500   | 0.021±0.006  | 30.186±2.506   | 1.027±0.312    | 0.138±0.001 |
>
> __To determine whether this gap stems from VC-2-FMPNN or potentially the limitations of supervised training dynamics__, e.g., memorizing spurious features, we experimented by using self-supervised learning (SSL) for Max-Cut SDP. Because Max-Cut SDP constraints $X_{ii} = 1$ are easily satisfied, it serves as an ideal proof-of-concept. Using the **exact same VC-2-FMPNN architecture from our submission**, we only changed the prediction head to map outputs.
>
> - Primal SSL: The network outputs a latent embedding $\mathbf{V} \in \mathbb{R}^{n \times d}$. We train the network by directly minimizing $< \mathbf{C}, \mathbf{V} \mathbf{V}^T >$. Row-normalizing $\mathbf{V}$ satisfies the $X_{ii} = 1$.
> | Size    | obj. gap | Time (sec.) |
> |---|---|---|
> | 100   | 4.683±0.476  | 0.009±0.002 |
> | 200   | 5.404±0.032  | 0.010±0.002 |
> | 300   | 7.621±0.905  | 0.015±0.001 |
> | 500   | 15.863±3.787  | 0.065±0.001 |
>
> - Dual SSL: The network predicts the dual variables $\mathbf{y} \in \mathbb{R}^n$. To enforce the positive semidefinite (PSD) constraint on the slack variable $\mathbf{S} = \mathbf{C} - \text{diag}(\mathbf{y})$, we compute its most negative eigenvalue $\lambda < 0$ and apply a shift $\mathbf{y} \leftarrow \mathbf{y} + \lambda \mathbf{1}$, $\mathbf{S} \leftarrow \mathbf{S} - \lambda \mathbf{I}$, so that we obtain __feasible__ dual solution. The network is then trained to minimize the dual objective $-\sum y_i$.
> | Size    | obj. gap | Time (sec.) |
> |---|---|---|
> | 100   | 0.067±0.000 | 0.010±0.001 |
> | 200   | 0.389±0.022 | 0.011±0.002 |
> | 300   | 0.501±0.030 | 0.022±0.002 |
> | 500   | 0.775±0.161 | 0.077±0.001 |
>
> As shown in the tables, __when freed from rigid supervised labels, VC-2-FMPNN (especially with dual SSL) scales exceptionally well to larger graphs__. Meanwhile, although stable, primal SSL may need more training tricks to unleash its power.
>
> We conclude:
> 1. Supervised results already validates the expressivity of VC-2-FMPNN over baselines.
> 2. This SSL ablation further consolidates our work, empirically demonstrating that VC-2-FMPNN inherently captures the continuous optimization landscape, enabling more robust scaling.
>
> __We hope these new empirical results address your concerns, and we kindly ask you to consider raising your score. We are happy to answer further questions.__

---

> > ### Author Rebuttal · Reviewer_bC7w · 2026-04-01
> >
> > Thanks to the authors for their thorough response.
> >
> > In my view, the paper's exploration of the "expressive power of GNNs for SDPs" is already sufficiently innovative and forms a solid standalone contribution. The issues regarding generalization and oversmoothing, while important, are essentially "icing on the cake" for a paper focused on establishing fundamental expressivity.
> >
> > The explanations and additional empirical data provided in the rebuttal regarding these two aspects, while perhaps not completely resolving the issues theoretically, are more than sufficient to alleviate my practical concerns. They do not detract from the main merits of the paper.
> >
> > Overall, this is a strong paper with a clear and novel contribution. I am satisfied with the rebuttal and maintain my recommendation for acceptance.

---

> > > ### Author Response · Authors · 2026-04-01
> > >
> > > We sincerely thank the reviewer for the highly encouraging feedback and the recommendation for acceptance. We are very glad that the additional empirical data and discussion effectively alleviated your practical concerns. Building upon this solid theoretical foundation, we will explore the theoretical generalization performance as a promising direction for future work.

---

### Official Review · Reviewer_cBxK · 2026-03-11

**Soundness:** 3
**Presentation:** 3
**Significance:** 3
**Originality:** 3
**Overall Recommendation:** 4
**Confidence:** 4

**Summary:**

This paper investigates the expressive power of Graph Neural Networks (GNNs) in the context of solving linear Semidefinite Programs (SDPs). Given the high computational cost of traditional SDP solvers, the authors explore using GNNs as fast surrogates to predict optimal solutions. The work provides theoretical results showing that standard GNNs are insufficient to recover optimal SDP solutions due to their inability to capture the required matrix-level interactions. To address this, the authors propose a more expressive architecture—VC-2-FMPNN (Variable-Constraint 2-order Folded Message Passing Neural Network)—which can emulate the updates of standard first-order solvers. Empirically, the model demonstrates superior performance on synthetic and SDPLIB benchmarks, and when used to warm-start first-order solvers, it achieves speedups of up to 80%.

**Compliance With Llm Reviewing Policy:**

Affirmed.

**Final Justification:**

This paper successfully bridges the gap between GNN expressive power and convex optimization. The authors’ responses have addressed most of my concerns. Although I still have a minor reservation regarding scalability, I maintain that the work is solid and of high quality.

**Key Questions For Authors:**

1. Efficiency: What is the specific inference time for the VC-2-FMPNN on the largest SDPLIB instances compared to a single iteration of a standard first-order solver? Does the higher-order nature of the GNN lead to scalability issues for extremely large graphs?
﻿
2. Generalization: How well does the model generalize to SDP classes not seen during training?  For example, if trained on Max-Cut instances, can it provide useful warm-starts for Graph Coloring SDPs?
﻿
3. Scalability: Could the model scale effectively to ultra-large-scale graph problems? What are the theoretical and practical memory or time bottlenecks when the number of variables ($n$) becomes extremely large, and how might these be mitigated?

**Limitations:**

Yes.

**Strengths And Weaknesses:**

- Technical novelty and innovation: The paper bridges the gap between GNN expressive power and convex optimization. The identification of specific failure modes of standard MPNNs in representing SDP solutions and the proposal of a higher-order GNN (VC-2-FMPNN) to emulate first-order algorithms (like ADMM) is a significant conceptual contribution.
- Significance of contributions: SDPs are ubiquitous in optimization and machine learning. Providing a theoretically grounded GNN framework that can significantly speed up SDP solving through warm-starting has high practical value for large-scale combinatorial optimization.
---
- Technical limitations or concerns: While the VC-2-FMPNN shows higher expressive power, the paper could further clarify the trade-off between the increased computational complexity of higher-order message passing and the actual wall-clock time saved during the warm-start process.
- Experimental gaps or methodological issues: Although the speedups are impressive, the ``80% speedup'' claim needs more context regarding the specific overhead of the neural network inference itself compared to the total solver time across different problem scales.
- Clarity or presentation issues: Some of the mathematical notation for the higher-order GNN (VC-2-FMPNN) is dense. More intuitive visualizations of how the message passing aligns with ADMM steps would benefit the reader.
- Missing related work or comparisons: While first-order solvers are mentioned, more explicit comparisons with other recent ML-based optimization surrogates (e.g., those using Transformers or different graph-based architectures for LPs/MIPs) would strengthen the positioning.

---

> ### Author Rebuttal · Authors · 2026-03-27
>
> We sincerely thank the reviewer for recognizing our novelty and significance. We address the concerns below:
>
> # Presentation weaknesses
> We greatly appreciate the reviewer’s constructive feedback regarding the clarity and contextualization of our claims, which we will fully incorporate into the camera-ready version.
> - Efficiency & Speedup: As acknowledged in our limitations (Appendix A), VC-2-FMPNN has high theoretical computational costs. However, as demonstrated by the runtime comparisons in Table 3, our neural proxy does not suffer from the exponential runtime observed with classical solvers. Because our higher-order models leverage GPU parallelism, their theoretical complexity does not translate into prohibitive actual execution time. We will also explicitly clarify the "80% speedup" and the inherent expressivity-efficiency trade-off in the revision.
> - Visualization: To make the dense mathematical alignment in Theorem 2.3 more intuitive for a broader audience, we will add a clear illustration that shows how our nodes and messages emulate PDHG updates.
> - Related work/baselines: We acknowledge the graph-based LP/MILP methods mentioned by the reviewer. We will keep an eye on the latest neural optimizer work and update our related work section. For baselines, __we have indeed extended graph-based LP solving to SDP, as represented in our paper by the VC-WL__. It empirically fails due to insufficient expressivity. Furthermore, __we have included Edge Transformer__ in Table 1 and Appendix E.4, which has the same VC-2-FWL expressivity but incurs slower inference time due to global attention overhead.
> __If there are other strong neural baselines we may have overlooked, please let us know and we would be happy to include them in our experiments__.
>
> # Q1 Regarding efficiency
> The hardest problem of SDPLIB is `MCP500-1`. The SCS solver requires approximately `328 seconds` to converge over `~1400 iterations` (about `0.238 seconds per step`). Our GNN runtime with `0.147 seconds` is faster than even a single classical iteration.
> Our paper exhibits $\mathcal{O}((n^3 + \text{nnz}(A)) \log n)$ complexity for converged VC-2-FWL, which is likely to be inefficient on very large instances. However, we argue that cubic complexity is inevitable, as classical solvers for general SDPs require a Frobenius projection onto the PSD cone at each iteration (see Algorithm 1 in Appendix C.3), which requires eigenvalue decomposition with cubic complexity.
>
> # Q2 Regarding generalization
> Generalizing across fundamentally different classes is known as out-of-distribution (OOD) generalization and is notoriously difficult due to various distributions of the problems. We take the VC-2-FMPNN pretrained on Max-Clique and evaluate its OOD generalization:
> - Tested on Maximum Independent Set (MIS): `101.354±0.283%` objective gap.
> - Tested on Max-Cut: `9.592±3.001%` objective gap.
>
> The severe gap in MIS is likely due to Max-Clique and MIS operating on complementary graph topologies, pushing the network far out of distribution. However, inspired by recent successes in graph foundation models [1][2], a highly promising future direction is constructing a massive, mixed SDP dataset to train a unified model capable of broad generalization.
>
> # Q3 Regarding scalability
> In our opinion, scaling to ultra-large instances faces two fundamental bottlenecks: 1) the $\mathcal{O}(n^3)$ complexity and 2) the prohibitive cost of generating ground-truth labels using classical solvers. Regarding time and memory, the cubic complexity is mathematically inevitable for _general_ linear SDPs, even validating the positive semi-definiteness $\mathbf{X} \succeq 0$ requires an eigenvalue decomposition. However, we remain positive about designing more efficient neural architectures for _specific_ types of SDPs. Furthermore, to directly overcome the second bottleneck, we experimented a self-supervised learning (SSL) framework for Max-Cut SDPs. As requested by __Reviewer bC7w__, we evaluated this approach and found that by optimizing dual SSL loss directly, our VC-2-FMPNN generalizes well to graphs $5\times$ larger than the training set, maintaining a tight objective gap and strict feasibility. We kindly direct the reviewer to our detailed response and tables provided to __Reviewer bC7w__, which demonstrate the scalability of our work.
>
> __We hope these clarifications and empirical insights address your concerns and strengthen your confidence, and we kindly ask you to consider raising your score. We are happy to answer further questions__.
>
>
> [1] https://arxiv.org/abs/2310.04292
> [2] https://arxiv.org/abs/2405.20445

---

> > ### Author Rebuttal · Reviewer_cBxK · 2026-04-01
> >
> > Thank you for the authors' response. Although most of my concerns have been addressed, two key points remain:
> > ﻿
> > 1. It seems that the question regarding "ultra-large dataset scalability" has not been directly addressed. It is not entirely clear whether a task involving only 100 nodes (line 302) is sufficient to demonstrate the method's effectiveness on the described bottlenecks. Have you ever conducted experiments on larger scales, and if so, what were the results? If this is already detailed in the text, please point me to the specific section.
> > ﻿
> > 2. The clarification regarding the "80% speedup" remains insufficient. I would like to confirm if this is derived from line 404 ("yielding runtime reductions up to 80.7%"). Furthermore, is this efficiency gain consistent across different scenarios, or is it an isolated, best-case result? If it is the latter, I suggest that the authors use a range or a median value in the abstract to provide a more representative summary.

---

> > > ### Author Response · Authors · 2026-04-04
> > >
> > > We sincerely thank the reviewer for the continued engagement and helping us refine the clarity and precision of our paper. We address your two remaining points below:
> > >
> > > # Ultra-Large Dataset Scalability
> > >
> > > Scaling neural SDP solvers to ultra-large instances faces two fundamental bottlenecks: 1) the mathematical $\mathcal{O}(n^3)$ complexity inherent to PSD nature, and 2) the computational cost of generating ground-truth training labels. We had not included ultra-large scale experiments in the initial submission due to the second limitation. While mitigating the $\mathcal{O}(n^3)$ complexity requires fundamentally different architectures (which we leave for future work), we successfully overcame the second bottleneck, by utilizing __self-supervised learning (SSL)__ and __test-time size generalization__.
> > >
> > > As also suggested by __Reviewer b7Cw__, we conducted size generalization experiments on Max-Cut SDP problems, as the constraints are simply $X_{ii} = 1$ and easy to satisfy.
> > > Importantly, our VC-2-FMPNN can be equipped with separate prediction heads for primal and dual variable prediction. To bypass the calculation of the dense primal matrix $\mathbf{X}$, we utilized the network's dual prediction head to predict the dual variables $\mathbf{y} \in \mathbb{R}^n$.
> > > To strictly enforce the PSD constraint on the slack variable matrix $\mathbf{S} = \mathbf{C} - \text{diag}(\mathbf{y})$, we compute its most negative eigenvalue $\lambda < 0$ and apply a shift: $\mathbf{y} \leftarrow \mathbf{y} + \lambda \mathbf{1}$ and $\mathbf{S} \leftarrow \mathbf{S} - \lambda \mathbf{I}$. This guarantees a feasible dual solution, allowing us to train the network by minimizing the dual objective $-\sum y_i$.
> > >
> > > - Train on 100 nodes, test up to 500 nodes
> > >
> > > First, we trained using SSL on 100-node Max-Cut problems and tested on larger graphs. The model exhibits strong size generalization with feasibility. See also our reply to Reviewer bC7w:
> > >
> > > | Size    | obj. gap (%) | cons. vio. | Time (sec.) |
> > > |---|---|---|---|
> > > | 100   | 0.067±0.000    | 0.0            | 0.010±0.001 |
> > > | 200   | 0.389±0.022    | 0.0            | 0.011±0.002 |
> > > | 300   | 0.501±0.030    | 0.0            | 0.022±0.002 |
> > > | 500   | 0.775±0.161    | 0.0            | 0.077±0.001 |
> > >
> > > - Train on 500 nodes, test up to 3000 nodes (9 million variables)
> > >
> > > To directly answer the question regarding the upper limits of our architecture, we created an unlabeled dataset of 10000 Max-Cut SDP problems, which can be generated efficiently without computing the ground truth label. To calculate the objective gaps for evaluation, we ran the SDPLR solver[1], which we found faster on Max-Cut problems compared with SCS and MOSEK in our paper, exclusively on the test sets to obtain the ground-truth optimal values. Using the same hyper-parameters from the paper, and batchsize 16, with 50 epochs (training takes about 5 mins/epoch on 4 NVIDIA L40S GPUs), we trained the models via SSL, repeated with random seeds and evaluated its performance on test sets of graphs up to 3000 nodes (9 million variables in the SDP).
> > >
> > > | Size | obj. gap (%)   | Time (sec)   | SDPLR(sec) |GNN VRAM (GB) |
> > > |---|---|---|---|---|
> > > | 500  | 0.077±0.012 | 0.070±0.005 | 5        | 0.6     |
> > > | 1000 | 0.120±0.013 | 0.319±0.023 | 25       | 2.3     |
> > > | 1500 | 0.168±0.012 | 0.750±0.017 | 75       | 5.3     |
> > > | 2000 | 0.211±0.020 | 1.368±0.021 | 360      | 9.4  |
> > > | 3000 | 0.397±0.039 | 3.335±0.101 | 2150 | 25.1 |
> > >
> > > The pretrained VC-2-FMPNN can easily extend to 3000 node Max-Cut problems with a very low objective gap (<0.4%). Besides, VC-2-FMPNN efficiently handles 9 million variables at test time in just 3.33 seconds. In contrast, the SDPLR solver, requires about 35 minutes on average. This demonstrates that VC-2-FMPNN achieves an extraordinary $\sim 645\times$ inference speedup at this scale.
> > >
> > > While the 25 GB VRAM at this scale is substantial, we emphasize that our current implementation does not utilize any memory-saving optimizations. Applying chunking techniques to the dense matrix multiplications, analogous to FlashAttention [2], will be a highly promising path to completely mitigate this memory bottleneck in future work.
> > >
> > > We will add a "Scalability Limits" section to the appendix detailing these results.
> > >
> > > # Clarification on the "80% Speedup" Claim
> > >
> > > We appreciate the reviewer pointing this out. The "up to 80%" was obtained as the maximal improvement across the SDPLIB benchmark. We completely agree with your suggestion that a range and average are much more scientifically representative. In the revision paper, we will revise the abstract to state: "yields practical speedups ranging from 11% to 80% (averaging ~50%) across real-world benchmarks" to ensure the summary is entirely transparent.
> > >
> > > __If these experiments address your remaining concerns, we would be incredibly grateful if you might consider reflecting this in your final justification and score__.
> > >
> > > [1] https://github.com/sburer/sdplr
> > > [2] https://arxiv.org/abs/2205.14135

---

### Official Review · Reviewer_rHPh · 2026-03-12

**Soundness:** 3
**Presentation:** 2
**Significance:** 2
**Originality:** 3
**Overall Recommendation:** 4
**Confidence:** 3

**Summary:**

The paper studies the problem of solving large-scale semidefinite programs (SDPs), for which classical optimization solvers can become computationally expensive. The authors propose several GNN-based architectures intended to approximate SDP solutions.

The paper first introduces a set of architectures aimed at predicting SDP solutions. However, the authors argue that these initial models are insufficient to recover optimal SDP solutions due to limited expressive power. To address this limitation, they propose a more expressive architecture, VC-2-FWL, which is designed to better capture the key structure of SDPs.

In the experimental evaluation, the authors demonstrate that the earlier architectures indeed fail to recover satisfactory solutions, while the proposed VC-2-FWL architecture achieves improved performance relative to these insufficient baselines. However, the experiments do not include comparisons with established SDP solvers from the prior literature. Additionally, although the motivation of the work is to address large-scale SDPs, the empirical evaluation does not include experiments on large problem instances.

Finally, the authors show that the solutions predicted by their GNN-based model can be used to warm-start a first-order SDP solver, which can lead to practical speedups.

**Compliance With Llm Reviewing Policy:**

Affirmed.

**Final Justification:**

The reviewers have satisfactorily addressed my concerns via additional experimental results and comparisons with SDPLR. As a result, I am raising my score.

**Key Questions For Authors:**

Q1: Why both VC-2-WL and VC-2-FWL satisfy the three design principles? Is this clearly explained in the paper?

Q2: Additionally, there exist specialised algorithms for exploiting structured sparsity in SDPs (see reference below). Can the authors comment on the viability of their approach for such SDP instances?

Vandenberghe, Lieven, and Martin S. Andersen. "Chordal graphs and semidefinite optimization." Foundations and Trends in Optimization 1, no. 4 (2015): 241-433.

**Limitations:**

yes

**Strengths And Weaknesses:**

Strengths:

S1: The authors present experiments that suggest that the proposed method can provide significant speedups when used to warm-start an SDP solver.

S2: The work studies the expressive power required for GNN architectures to represent SDP solutions and motivates the need for more expressive models, leading to the proposed VC-2-FWL architecture.

S3: The paper provides both theoretical discussion and empirical evidence highlighting the limitations of simpler architectures.

Weaknesses:

W1: The presentation does not meet expectations. While the formal proofs are placed in the appendix, which is standard practice, the main paper provides limited intuition or high-level explanation of the key claims and theoretical results. As a result, it is difficult to understand the key underlying ideas and implications without carefully studying the appendix. For example, the proof of Theorem 2.3 relies on establishing a link between PDHG algorithm and the stable colorings of VC-2-FWL, but the insights underpinning the technique are completely relegated to the appendix.

W2: Regarding the complexity claims of the paper. The paper argues that the complexity of VC-2-FWL, O((n^3 + nnz(A)) log n) is comparable to the complexity of state-of-the-art SDP solvers such as interior-point methods, O((n^3.5) log(1/\epsilon)) of the fastest implementation of IPM for SDPs, or O(n^3) per-iteration cost of some first-order methods. However, several aspects of this comparison remain unclear. First, the proposed method does not compute exact optimal solutions. Second, the paper does not provide a detailed evaluation of how close the predicted solutions are to those obtained by standard SDP solvers, in terms of loss, and the residual constraint violations.

W3: The practical implications of the results are not clear. Although the paper uses the architectures to solve SDP relaxations of hard CO problems, no attempt is made to verify how good the resulting solutions are. One could point to other alternatives for solving such SDP relaxations, such as the class of Burer-Monteiro factorizations (see references below). The omission of such a comparison limits the significance of the results.

Burer, Samuel, and Renato DC Monteiro. "A nonlinear programming algorithm for solving semidefinite programs via low-rank factorization." Mathematical programming 95, no. 2 (2003): 329-357.

Boumal, Nicolas, Vladislav Voroninski, and Afonso S. Bandeira. "Deterministic Guarantees for Burer‐Monteiro Factorizations of Smooth Semidefinite Programs." Communications on Pure and Applied Mathematics 73, no. 3 (2020): 581-608.

Bhojanapalli, Srinadh, Anastasios Kyrillidis, and Sujay Sanghavi. "Dropping convexity for faster semi-definite optimization." In Conference on Learning Theory, pp. 530-582. PMLR, 2016.

W4: Although the paper is motivated by the challenge of solving large-scale SDPs, the empirical evaluation does not include experiments on large problem instances. This makes it difficult to assess whether the proposed approach achieves its intended goal, despite the warm-start experiments.

---

> ### Author Rebuttal · Authors · 2026-03-27
>
> We sincerely thank the reviewer for the constructive feedback and insightful recommendations on the literature. We address the specific weaknesses and questions below.
> # Presentation (W1)
> We agree that high-level intuition is crucial for readability. In our final version, using the extra page, we will update the main paper to include a proof sketch, highlighting the geometric intuition between PDHG algorithm and the stable colorings of VC-2-FWL, ensuring the core ideas are easily understandable.
>
> # Complexity (W2)
> The reviewer correctly notes that our method does not compute exact optimal solutions. Indeed, our theoretical analysis focuses on the complexity of the converged VC-2-FWL, which is guaranteed to refine each update of PDHG. In fact, establishing rigorous theoretical neural network complexity bound up to $\epsilon$-approximation is non-trivial, and our work takes the fundamental first step by establishing the strict separation power required for such approximations.
>
> # Solution quality (W2 & W3)
> The reviewer __repeated in W2 and W3__ that our paper lacks evaluation of the solution quality. __We have indeed provided extensive detail on how close our predictions are to standard solvers in terms of test loss, objective gap, and residual constraint violations__ in Tables 1 & 2 and Tables 7 & 8 (Appendix). Furthermore, the high practical quality of these neural approximations is directly evidenced by their __ability to warm-start the classical SCS solver__, thereby reducing overall solving time by up to 80%, as shown in Tables 3 & 4. We politely refer the reviewer to these tables and we are happy to conduct further evaluation if required.
>
> # Burer-Monteiro method (W3)
> We thank the reviewer for pointing out Burer-Monteiro (BM) factorization as an alternative. We must emphasize, however, that SDPLR is a classical optimization algorithm rather than a neural baseline. A neural solver is not comparable to a mathematical optimizer in terms of exact solution quality. We extended our Table 3 comparing our inference time with that of the SDPLR solver [1]:
>
> | Solver | 50 | 100 | 150 |
> |--- | --- | --- | --- |
> | SDPLR         | 0.111±0.00 | 0.195±0.041 | 0.285±0.062 |
> |VC-2-FMPNN | 0.009±0.000 | 0.010±0.000 | 0.013±0.000 |
>
> While SDPLR is highly efficient on large problems, our VC-2-FMPNN remains faster because it can leverage massive GPU parallelism (and even for batched inference). Most importantly, the core significance of our neural approach lies in the neural architecture's capability to learn underlying data distributions and generalize to unseen instances, a powerful data-driven advantage that classical solvers lack.
>
> # Scalability (W4)
> Deploying neural SDP solvers at an industrial scale remains a long-term goal for the community. We emphasize that the primary value of our work lies in the theoretical background. Our empirical results are designed to validate the theoretical foundation.
> That said, we recognize that scaling is bottlenecked by the intractable cost of generating ground-truth labels for supervised training. To overcome this, we experimented with a self-supervised learning framework that optimizes the objective directly. We kindly refer to our response to __Reviewer bC7w__, it allowed us to train VC-2-FMPNN on small instances and evaluate on graphs $5\times$ larger. The architecture successfully maintained tight objective gaps and strict feasibility at scale.
>
> # Invariance & Equivariance (Q1)
> In the graph learning community, it is known that MPNN (including $k$-(F)WL) is permutation-equivariant, and multiset-based function is permutation-invariant [2][3]. Wrt our design principles:
> 1. it is structurally guaranteed by design.
> 2. In VC-2-WL, the row/column are aggregated as order-invariant multisets. Any permutation of the problem $C$ and $A$ also permutes the update on variables, ensuring equivariance. VC-2-FWL maintains this exact equivariance because its outer multiset aggregates over the row/column.
> 3. Both architectures are invariant to the ordering of the constraints. Because Information from the constraint nodes is aggregated into the variables via a multiset.
> # Sparsity (Q2)
> We thank the reviewer for highlighting Vandenberghe et al.'s work. Adapting VC-2-FWL to exploit sparsity with chordal patterns is __highly viable__. One possibility would be to decompose a sparse matrix with chordal patterns into smaller ones, following the paper’s theory in Chapter 9. By performing VC-2-FWL on these small patterns, the complexity would drop significantly. Another possibility would be a topology-specific WL hierarchy on the sparse chordal patterns directly. Both are exciting and promising.
>
> __We hope these clarifications address all the concerns, and we kindly ask the reviewer to consider raising the score if our defenses are valid. We are happy to answer any further questions__.
>
> [1] https://github.com/sburer/sdplr
> [2] https://arxiv.org/abs/1703.06114
> [3] https://arxiv.org/abs/1812.09902

---

> > ### Author Rebuttal · Reviewer_rHPh · 2026-04-02
> >
> > I thank the authors for their detailed rebuttal, which has cleared up most of my concerns. However, I still have 1-2 points remaining.
> >
> > (1) Regarding the authors' response to W2-W3: indeed a thorough examination of the objective loss for the SDP is conducted, but analysis of constraint violations are limited to the max-cut example (Table 8), whereas the paper conducts experiments across 5 additional SDP relaxations, but does not report the results.  Additionally, the point I wanted to make in W3 is that the SDP solution is typically used to construct approximate solutions of the CO problem (e.g. via Gaussian randomization, randomized rounding). How well does the proposed approach fare for the respective CO problems, has not been addressed. Despite the solid theoretical foundations, it is not clear whether the resulting approach can serve as a breakthrough in terms of changing how we solve SDP relaxations of CO problems. I request the authors to clarify this point, as this has not been satisfactorily addressed.
> >
> > (2) The comparison to SDPLR is incomplete and inconclusive. The BM approach is not guaranteed to solve the SDP exactly in general, like the neural solvers. Without including the solution quality, I can't make a judgement. Since the authors show that the neural solvers are much faster, they may justify their approach in terms of a trade-off between solution quality and time complexity.

---

> > > ### Author Response · Authors · 2026-04-05
> > >
> > > We sincerely thank the reviewer for pushing us to provide more comprehensive empirical studies to strengthen our paper.
> > >
> > > # Constraint violations
> > > We have expanded our evaluation in Table 1 to include the (PSD projected) constraint violations, objective gaps, and runtimes across all 6 datasets for our VC-2-FMPNN, SDPLR, and the exact SCS solver. Objective gaps are all calculated using the exact SCS solution as the ground truth.
> > >
> > > | Candidate  | Target       | Max-Cut (ER)   | Max-Cut (Reg)  | Max-Clique     | MIS            | Vertex Cover   | Max-2-SAT       |
> > > |---|---|---|---|---|---|---|---|
> > > | VC-2-FMPNN | Time (sec)   | 0.010±0.000    | 0.010±0.000    | 0.019±0.001 | 0.019±0.001 | 0.019±0.001 | 0.011±0.001     |
> > > |            | Obj. gap (%) | 0.111±0.017    | 0.092±0.004    | 0.436±0.035    | 0.411±0.025    | 0.469±0.051    | 0.862±0.211     |
> > > |            | Cons. vio.   | 0.003±0.001    | 0.003±0.001    | 3.082e-5±0.000 | 3.067e-5±0.000 |0.009±0.001    |0.008±0.001    |
> > > | SDPLR      | Time (sec)   | 0.195±0.041    | 0.235±0.052    | 1.803±0.184    | 3.301±0.285    | Not converging | 0.391±0.145     |
> > > |            | Obj. gap (%)     | 3.672e-06±0.000    | 4.434e-6±0.000 | 0.020±0.001    | 0.022±0.002    | --               | 2.341e-05±0.000 |
> > > |            | Cons. vio.   | 1.059e-6±0.000 | 1.193e-6±0.000 | 4.269e-7±0.000 | 2.696e-7±0.000 | --             | 8.223e-07±0.000 |
> > > | SCS        | Time (sec)   | 0.386±0.028    | 0.385±0.190    | 0.642±0.308    | 0.672±0.340    | 0.089±0.004    | 0.266±0.081     |
> > >
> > > As demonstrated in the table, our neural approach achieves remarkably low constraint violation residues consistently across all tasks, on the order of $10^{-3}$ to $10^{-5}$. While not perfectly feasible, these violations are very low and highly competitive for practical applications.
> > >
> > > # Evaluation on CO problems
> > > To be mathematically precise about our scope: our method is designed to act as an efficient proxy for the SDP relaxation itself. Consequently, __it intrinsically inherits the theoretical integrality gap of SDP formulations__.
> > >
> > > To directly answer the question regarding how well our predictions translate to discrete CO solutions, we evaluated both the _constructive rounded gap_ (using Gaussian randomization) for Max-Cut, and the _theoretical integrality gap_ for MIS and Vertex Cover. Due to the limited rebuttal window, we provide three representative tasks here, but commit to including downstream CO evaluations for all tasks in the final appendix.
> > >
> > > | Candidate | Max-Cut (%)  | MIS (%) | Vertex Cover (%) |
> > > | ---| ---| ---| ---|
> > > | SDP - CO | 4.547 ± 0.858 | 199.229±2.465 | 44.622±3.699 |
> > > | VC-2-FMPNN - CO | 4.610 ± 0.855 | 199.358±3.426 | 44.955±3.897 |
> > >
> > > As the table shows, the downstream CO gaps produced by our neural solver perfectly mirror the inherent gaps of the exact SDP relaxations. For Max-Cut, randomized rounding on our GNN outputs performs almost identically to rounding exact solver outputs, with a negligible ~0.06% difference. The observations have validated our core claim: the neural approach successfully learns the underlying continuous SDP geometry, rather than directly learning the discrete integer solutions.
> > >
> > > However, __the neural solver’s high quality solution and fast inference time enable massive potential for CO solvers__. As stated in our Appendix A, modern exact branch-and-bound methods for Max-Cut [1] require solving thousands of intermediate SDPs, which is the primary computational bottleneck. Replacing these expensive exact solvers with our cheap neural proxy to guide branching or pruning decisions, similar to the seminal work of Gasse et al. [2] for LPs, is a highly promising future direction.
> > >
> > > # Regarding BM method and SDPLR
> > > As the reviewer pointed out, BM methods are not guaranteed to solve the SDP exactly. Our extended table now fully addresses this. SDPLR yields near-zero objective gaps compared to SCS, except for 0.02% on Max-Clique/MIS; and almost zero constraint violation up to numerical precisions. However, it fails to converge on Vertex Cover, and scales poorly in runtime on MIS.
> > >
> > > We would like to emphasize that our neural approach is incomparable to SDPLR as a solver in terms of solution quality. We completely agree with the reviewer's characterization that our neural solver presents a necessary trade-off. By approximating the complex geometry of SDPs with a constant number of neural layers, we sacrifice exactness (with <1% objective gap) to achieve orders-of-magnitude speedups, generalizable inference on large sizes, and the ability to leverage GPU acceleration especially on batched problems.
> > >
> > > We hope these new empirical evaluations and CO analyses fully address your remaining points. __If our response and clarified scopes satisfactorily resolve your concerns, we would be incredibly grateful if you considered reflecting this in your final score__.
> > >
> > > [1] https://arxiv.org/abs/2010.07839
> > > [2] https://arxiv.org/abs/1906.01629

---

### Official Review · Reviewer_iP1Y · 2026-03-13

**Soundness:** 3
**Presentation:** 3
**Significance:** 2
**Originality:** 2
**Overall Recommendation:** 4
**Confidence:** 3

**Summary:**

This paper studies the expressivity of Graph Neural Networks (GNNs) for solving Semidefinite Programming (SDP).
The authors first show that existing GNN structures cannot recover linear SDP solutions via a negative example.
They then propose a more expressive structure and prove its universality to solve SDPs by emulating the updates of a standard first-order solver.
Empirical results show that the proposed structure consistently achieves lower prediction error and objective gap than baselines.

**Compliance With Llm Reviewing Policy:**

Affirmed.

**Key Questions For Authors:**

Minor Concerns:
- Typo in line 213, NN(c,A,Qb)=NN(c,A,b) should be NN(c,QA,Qb)=NN(c,A,b)?
- If I understand right, in Lemma C.20, the range of f should be in a different space instead of $\mathbb{R}^d$, align with u(x)’s definition in [1]
- The Hash function may not be continuous in general, so using MLP to approximate the hash function may not be convincing in Section C.6.
- In equality (8) in page 20, to apply the sum rule of subdifferential, it’s better to add f is proper beyond convex and closed, so that the relative interior of dom(f) is non-empty.


[1]: Maron, Haggai, et al. "Provably powerful graph networks." Advances in neural information processing systems 32 (2019).

**Limitations:**

Yes

**Strengths And Weaknesses:**

Strength:
- This paper establishes a rigorous theoretical framework bridging GNN expressivity (via the Weisfeiler-Leman hierarchy) and continuous optimization (specifically, a modified PDHG). The findings are particularly compelling: a VC-2-FWL-based GNN can "refine" the PDHG algorithm for SDP problem with Frobenius norm regularization.
- The experimental setup across a wide range of SDPs supports the theoretical claims. By carefully selecting baselines across the expressivity spectrum, the authors empirically validate the strict expressivity hierarchy.
- The paper is well-written and easy to read.

Weakness.
- The authors state that OptGNN was specifically designed for discrete problems (e.g., Max-CSPs) and therefore excluded this baseline. However, both frameworks (VC-2-FMPNN and OptGNN) fundamentally require solving the SDP problem first, albeit using different algorithm unrolling approaches in their proofs (PDHG vs. low-rank primal-dual methods). Including this recent baseline would provide valuable insights into the comparative advantages and limitations of both frameworks, at least in the Max-Cut and Max-SAT problems, as the authors also test their VC-2-FMPNN in these domains (so did OptGNN).

- As the design of VC-2-FWL relies on the permutation action, it is not guaranteed to produce the same outputs when exchanging two arbitrary elements in the lower triangular block (while preserving matrix symmetry). Consequently, if two (linear) SDPs yield the same output multiset, their corresponding true solutions might not be equivalent up to permutations. This raises concerns about whether VC-2-FWL has sufficient separation power for (linear) SDPs.

- Beyond separation power, the approximation capability is equally crucial for providing downstream application guarantees. Is there a formal way to establish a connection between the model's theoretical expressivity and its continuous approximation abilities? (e.g., Can VC-2-FMPNN simulate the PDHG algorithm for SDPs?)
- Regarding computational complexity, it is inappropriate to directly compare the per-iteration cost of the WL-test with that of Interior Point Methods (IPMs). These two approaches serve fundamentally different purposes and operate under entirely distinct paradigms. A fairer comparison would be to evaluate the total iteration/GNN complexity given the same convergence threshold for SDP solutions.
- Regarding the negative results, a single counterexample is insufficient to disprove universal approximation. From a measure-theoretic perspective, if the failure occurs on a set of measure zero, the model may still maintain universal approximation capabilities almost everywhere (a.e.).

---

> ### Author Rebuttal · Authors · 2026-03-27
>
> We sincerely thank the reviewer for the extremely careful reading (including our Appendix) and the rigorous, mathematically grounded feedback, which will help us better shape the paper. We address the concerns in detail below:
>
> # 1. OptGNN
> We adapted OptGNN to our evaluation framework for comparison in obj. gap on Max-Cut SDP (other datasets are incompatible due to formulation differences or unavailable source code).
>
> | Model | Target | Max-Cut-ER | Max-Cut-Regular |
> | --- | --- |--- | --- |
> |OptGNN  | obj. gap | 1.304±0.014  |  2.030±0.019 |
> |OptGNN | cons. vio. |  0.0 | 0.0  |
> |VC-2-FMPNN | obj. gap |  0.111±0.017 |  0.092±0.004 |
> |VC-2-FMPNN | cons. vio. | 0.003±0.001  |  0.003±0.001 |
>
> OptGNN achieves 0 constraint violations because it optimizes a low-rank latent embedding $\mathbf{V} \in \mathbb{R}^{n \times d}$ where the Max-Cut constraint $X_{ii} = 1$ is trivially satisfied via row-normalization. While we could also adapt VC-2-FMPNN to predict the latent $\mathbf{V}$ to achieve feasibility, __we stick to our formulation in the submission__ to ensure consistency with general SDP. As shown, our architecture achieves an obj. gap one order of magnitude tighter. However, we still highlight our different scopes: OptGNN targets discrete CO problems using SDP relaxations as a tool, whereas our framework serves as a continuous surrogate for solving the general linear SDP.
>
> # 2. Matrix symmetry
> The reviewer’s question regarding swapping arbitrary elements touches on the fundamental geometry of SDPs. In fact, the concerned case will not happen with out VC-2-FWL. If one arbitrarily swaps two elements $X_{ij}$ and $X_{pq}$ without the corresponding row/column swaps, the underlying PSD property can be completely broken. In this case, the network should distinguish them. In our case, VC-2-FWL aggregates nested multisets (i.e., the $\{\{v_{uj}, v_{iu}\}\}$ pair over all nodes $u$), an arbitrary swap of $X_{ij}$ and $X_{pq}$ will change the inner and outer multisets.
> The structural symmetry of an SDP is strictly defined by equivariance under simultaneous row and column permutations. VC-2-FWL satisfies this. Actually, __Reviewer rHPh__ asked for explanation why our neural network satisfies the invariance and equivariance mentioned in our design principles in Section 2.2, we kindly refer to our detailed answer in that reply.
>
> # 3. Approximation ability
> There is a formal, rigorous connection between our discrete expressivity and continuous approximation. In Theorem 2.3, we prove that the converged VC-2-FWL colors formally refine every single continuous iteration of the PDHG algorithm. We state in our Proposition 2.5, that there exists VC-2-FMPNN that has maximal expressivity of VC-2-FWL power, with the proof technique of power-sum polynomials in [1]. Thus, VC-2-FMPNN can unroll and simulate PDHG. It is possible to formalize it similarly to [2][3], or a more detailed constructive proof as in [4].
> # 4. Computational complexity
> Our analysis in Proposition 2.4 is demonstrating that the __total complexity of the converged discrete VC-2-FWL algorithm__ is $\mathcal{O}((n^3 + \text{nnz}(\mathcal{A})) \log n)$, which compares to the convergence of fast IPMs. We have clarified this distinction in our paper. Furthermore, establishing an end-to-end $\epsilon$-approximation complexity bound of a GNN is non-trivial, and is definitely a great suggestion for future work.
> # 5. Universal approximation
> In the context of SDP from CO problems, inputs are not uniformly distributed across $\mathbb{R}^{n \times n}$. Instead, CO problems are concentrated on discrete structures. Furthermore, our failure cases are not isolated points. By Lemma C.22, scaling a failing instance by any $\alpha > 0$ generates an infinite set of failures. Note that this is only one of the many possible examples to construct failure cases. The empirical collapse of weaker models in Table 1 proves this is an expressivity limitation, not just a mathematical anomaly in measure space.
> # 6. Minor points
> - The typo in line 213 indeed has $\mathbf{Q}$ missing.
> - We agree the range of $f$ in Lemma C.20 should be modified to precisely align with [1].
> - The hash function in programming language is encrypted and discrete. However, in GNN expressivity, a hash function is an injective mapping that assigns unique representations to multisets or sequences, and it need not be discrete. In [5][6], there are various ways to prove such approximations to hash functions.
> - We have stated under Eq (7) that $f$ is closed and convex, but we will emphasize this.
>
> __We hope these clarifications address all the concerns, and we kindly ask the reviewer to consider raising the score if our defenses are valid. We are happy to answer any further questions__.
>
> [1] https://arxiv.org/abs/1905.11136
> [2] https://arxiv.org/abs/2406.01908
> [3] https://arxiv.org/abs/2310.10603
> [4] https://arxiv.org/abs/2502.02446
> [5] https://arxiv.org/abs/1810.00826
> [6] https://arxiv.org/abs/1810.02244

---

> > ### Author Rebuttal · Reviewer_iP1Y · 2026-04-03
> >
> > Thanks for the rebuttal. Most of my concerns have been addressed.
> >
> > - Regarding permutation symmetries, consider two SDP instances that produce identical output multisets under GNNs or the 2-FWL test. Since nodes of the same color may appear in different places across these two instances, one might need to permute the input matrix of one instance to align with the other, thereby verifying that their solutions are indeed equivalent. In the LP case, this can be easily done as there are no additional structure requirements. However, the required permutation is not necessarily a similarity transformation (i.e., of the form `PMPᵀ`) in the SDP case and thus may not be a "valid" operation, as it could break the PSD constraint of the solution matrix. This presents a challenge: if we define the "separation power" based on whether it produces the same output multiset for two inputs, we run into trouble that two instances that have the same output multiset may not have the same minimal $\ell_2$ norm solution (up to permutations), as demonstrated above. Therefore, I argue that the concept of separation of powers is crucial and requires a more rigorous definition than merely "producing the same output multiset." Establishing such a definition would be a meaningful contribution to the community. And this may lead to some new Stone-Weierstrass-based proof for universality.
> >
> > - Similar in computational complexity, the universal approximation, and the approximation of the hash functions. The original contribution is very good, but can be improved by demonstrating greater rigor in both the main text and the appendix. I agree that the techniques in your listing papers make sense for solving the problem, and it would be valuable to revise these details carefully for future readers.
> >
> > - For the sum rule of subdifferential, we know that it’s valid when the two functions’ domains ’ relative interior’s intersection isn’t empty. Since one function is real-valued and the relative interior of a non-empty convex set is non-empty, we just need the other function to be proper.

---

> > > ### Author Response · Authors · 2026-04-04
> > >
> > > We are glad to hear that our initial rebuttal addressed most of the reviewer concerns, and we sincerely thank the reviewer for this insightful follow-up discussion. We deeply appreciate your constructive feedback, which has provided a clear roadmap to strengthen our paper's theoretical foundations.
> > > # Permutation symmetries
> > > We agree that in the SDP case, an __arbitrary permutation in the multiset__ of the entries may violate the PSD constraint unless it takes the form of $PMP^T$. However, we believe the discrepancy between this sharp observation and our results stems from the choice of matrix norm evaluated. In Proposition 1.1 and throughout the paper, our theoretical framework focuses on the optimal solution with the least __Frobenius norm__, rather than the __spectral (L2) norm__. Crucially, the Frobenius norm $\lVert X \rVert_F^2 = \sum_{i,j} X_{ij}^2$ depends exclusively on the multiset of matrix entries, whereas the L2 norm defined by the largest eigenvalue depends on the matrix structure.
> > >
> > > We fully agree that even if VC-2-FWL outputs 2 identical multisets for 2 SDP problems, it does not mean that the SDPs are equivariant up to a permutation. To illustrate this, consider a pair of non-isomorphic graphs that cannot be distinguished by 2-FWL, as $G$ and $H$. If we construct a graph CO problem (e.g., Max-Cut) using their adjacency matrices as the objective coefficient matrix $C$, we obtain 2 SDP relaxations, $I_G$ and $I_H$. When VC-2-FWL is applied, it cannot distinguish the entries of $I_G$ from $I_H$. Because $G$ and $H$ are non-isomorphic, there exists no permutation matrix $P$ to transform $I_G$ into $I_H$. However, in SDP formulation, VC-2-FWL non-distinguishable entries yield the exact same solution values. Consequently, the solution multisets for both instances are identical. Therefore, their Frobenius norms are mathematically identical, meaning they share the same minimal Frobenius norm solution. __The reviewer is absolutely correct that they may not share the same minimal L2 norm solution due to the lack of a valid $PMP^T$ transformation, but our claims are valid wrt the Frobenius norm__.
> > >
> > > Furthermore, as stated in Theorem 2.3, the same converged VC-2-FWL color implies the same solution value on each entries $(i,j)$ and $(p,q)$. When defining separation power, our framework treats __each individual entry in the SDP problem__ as the atomic target, rather than the multiset of variables. This also leads to a corollary: if VC-2-FWL outputs the same multisets for two SDP problems, the multisets of their solutions must also be identical, with the same Frobenius norm.
> > >
> > > Interestingly, we do not assume that the entries indexed at $(i,j)$ and $(p, q)$ must originate from the same SDP instance. Even if they are drawn from different SDP problems, $X_{ij}^* = X_{pq}^*$ still holds. One can prove this by concatenating 2 separate SDP instances into one with 2 block diagonals, and simply applying Theorem 2.3. We believe this universal property addresses the concern for a more rigorous definition of separation power and sheds critical light on the generalization capabilities. Furthermore, to answer the reviewer's concern, establishing this entry-level separation power provides the exact mathematical foundation needed to apply the Stone-Weierstrass theorem for our universal approximation proofs in Appendix C.6.
> > >
> > > In summary, the reviewer’s concern regarding the SDP symmetry is mathematically sound and highly appreciated, but because our analysis is scoped to the __Frobenius norm__ and __entry-level separation__, our core results remain valid. We will explicitly clarify this distinction between the norms in our final revision.
> > > # Details of computational complexity, the universal approximation, and hash function
> > > We entirely agree that demonstrating greater mathematical rigor in these areas will benefit future readers. In the revision, we will expand the main text and appendix to explicitly detail these theoretical foundations. Specifically, we will make the comparison between VC-2-FWL's complexity and interior point method more rigorous, and better define the function $f$ in Lemma C.20, and expand on how power-sum multi-symmetric polynomials are constructed to guarantee the universal approximation of our injective hash functions.
> > > # Subdifferential
> > > We thank the reviewer for pointing out this technical nuance regarding the intersection of the relative interiors. In the revised paper, we will explicitly state that the function $f$ in Equation 8 is convex, closed, and __proper__, thereby formally satisfying the conditions required for the subdifferential sum rule to hold.
> > >
> > > __If these clarifications and our commitment to improve the theoretical details address your remaining concerns, we would be incredibly grateful if you might consider reflecting this in your final justification and score__.

---

### Decision · Program_Chairs · 2026-04-30

**Decision:**

Accept (regular)

**Comment:**

the paper considers the expressive power of GNN architectures to solve SDPs. It establishes negative results for standard architectures but shows that higher-order GNNs are sufficiently expressive. the rebuttal phase and ensuing discussion was good with some scores raised. the theoretical contributions are novel and the empirical validation looks good.